# Rethinking Out-of-Distribution Detection and Generalization with Collective Behavior Dynamics

**Zhenbin Wang[1], Lei Zhang[1,2]\*, Wei Huang[1], Zhao Zhang[1], Zizhou Wang[3]**

[1]Machine Intelligence Laboratory, Sichuan University, Chengdu, China
[2]Tianfu Jincheng Laboratory, Chengdu, China
[3]Institute of High Performance Computing, A*STAR, Singapore
`wangzhenbin@stu.scu.edu.cn`, `leizhang@scu.edu.cn`

## Abstract

Out-of-distribution (OOD) problems commonly occur when models process data with a distribution significantly deviates from the in-distribution (InD) training data. In this paper, we hypothesize that a *field* or *potential* more essential than features exists, and features are not the ultimate essence of the data but rather manifestations of them during training. With this in mind, we first treat the output of the feature extractor as charged particles and investigate their collective behavior dynamics within a self-consistent electric field. Then, to characterize the relationship between OOD problems and dynamical equations, we introduce the *basin of attraction* and prove that its boundary can be represented as the zero level set of a differentiable function of the potential, *i.e.*, the spatial integral of field. We further demonstrate that: *i)* InD and OOD inputs can be effectively separated based on whether they are steady state solutions for specific field conditions, enabling robust OOD detection and outperforming prior methods over three benchmarks. *ii)* the generalization capability correlates positively with the basin of attraction. By analyzing the dynamics of perturbations, we propose that the potential is well-characterized by a Fourier-domain form of the Poisson equation. Evaluated on six benchmark datasets, our method rivals the SoTA approaches for OOD generalization and can be seamlessly integrated with them to deliver additional gains. The code is available at https://github.com/wongzbb/CBD.

## 1 Introduction

Machine learning systems [15, 25, 70] have achieved significant success under the i.i.d. assumption, where training and test data are drawn from the same distribution, known as in-distribution (InD). However, this assumption is often violated in real-world scenarios [89, 95], as test data frequently originate from different distributions, referred to as out-of-distribution (OOD) data, which causes distribution shift and substantial performance degradation [20, 61, 21, 87]. Prior work on addressing OOD challenges primarily follows two directions: 1) OOD detection [19, 27, 49], which aims to identify test inputs that deviate from the training distribution to enhance model robustness and safety on unknown or anomalous data; 2) OOD generalization [8, 57, 1], which seeks to improve model performance on OOD data to enhance model transferability and adaptability. Despite extensive prior works and diverse methodologies, recent study [72, 36] in the community inspired by other disciplines is expected to provide new perspectives and insights into OOD problems.

Recent advancements witness the success of physics-inspired deep generative models, such as the Poisson flow model derived from electrostatics [92, 93] and the model based on the stochastic reversed heat equation [63]. The former abstracts data as charges and generates samples by evolving them

---

*The corresponding author.

39th Conference on Neural Information Processing Systems (NeurIPS 2025).

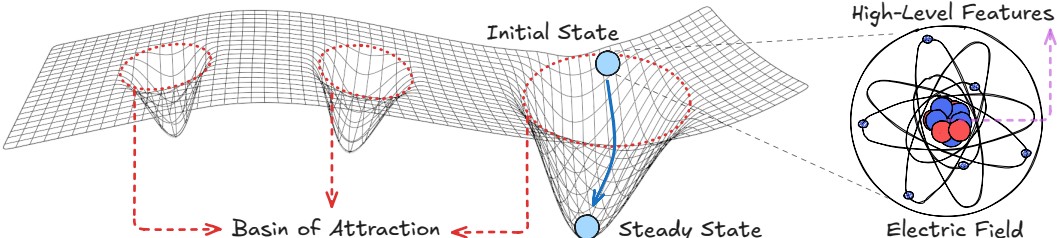

Figure 1: Conceptual illustration of our collective behavior dynamics insight for addressing OOD problems. High-level features are viewed as interacting particles within basin of attraction, transitioning from initial to steady states under the influence of a self-consistent electric field.

along electric field lines in an augmented space, while the latter interprets the solution of the forward heat equation with added constant noise as a variational approximation to a diffusion-based latent variable model. Overall, these approaches focus on learning the field of data using physical partial differential equations (PDE), rather than directly modeling the data distribution. These impressive achievements make us realize the fundamental nature of field or potential, with data are merely external manifestations or inherent properties of these deeper underlying concepts. This motivates us to model InD data using physical PDE. However, directly transferring prior work to OOD problems is infeasible, as generative models primarily focus on *the behavior of individual particles* within a field. In contrast, OOD problem center on determining field or potential boundaries, which requires analyzing *collective behavior* rather than individual behaviors. This discrepancy motivates us to first pose the following question:

> *Can we find a concise, theoretically grounded yet tractable PDE to model the collective behavior of the high-level features and measure the boundary of the basin of attraction?*

In pursuit of this, we introduce the Vlasov-Poisson system [67, 18], a dynamic physics model that describes the evolution of collective particle behavior under self-consistent electric field. This system captures the *feedback loop* where particle positions induce changes in the field, which, in turn, influence particle positions, driving the system towards a **steady state**[1]. In this sence, we abstract high-level features from the feature extractor as charged particles, with their values corresponding to initial positions and assigned uniform initial velocities. Under the assumption that all InD features correspond to a steady-state solution of the Vlasov-Poisson system, their dynamics will naturally evolve toward that equilibrium. The region formed by such initial states is referred to as the **basin of attraction** (as depicted in Figure 1). For OOD detection, we provide two principles: *i)* Vlasov-Poisson systems modeled by collective particles with distinct initial positions but identical initial velocities, typically yield *unique* steady-state solutions (Theorem 2.1), and *ii)* minor perturbations outside the basin of attraction amplify during system evolution, *sharpening gradients* at the boundary (Corollary 2.3). Thus, the basin of attraction is ideal for distinguishing InD and OOD features. Since this process operates on pretrained high-level features, it is therefore a post-hoc detection method. On the other hand, for OOD generalization, considering potential latent defects outside the basin of attraction, we assume that network robustness improves as the basin's range expands. So we next seek to address the following question:

> *How can we enlarge basin of attraction's boundary to enhance model generalization?*

To answer this, we decompose complex perturbations into superpositions of simple **plane waves**[2] and examine their collective behavior in the Vlasov-Poisson system by giving a dispersion relation that characterizes the stability of the wave modes. Through this lens, if all wave modes converge to a common steady state, the system ultimately stabilizes at that solution, regardless of the initial perturbation, as illustrated in Figure 2. Therefore, by requiring the potential in the random single-wave mode to satisfy the frequency form of Poisson equation, we expand the basin of attraction to include the perturbation points. We present Theorem 2.4 to support

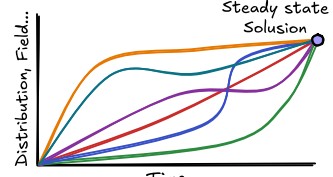

Figure 2: As all plane-wave modes reach steady state, cross-frequency interactions diminish, leading to equilibrium.

---

[1] A steady state refers to a condition where the system's macroscopic properties remain constant over time, indicating a dynamic equilibrium between particle motion and the self-consistent field.

[2] A wave with uniform amplitude and phase across infinite, parallel wavefronts, propagating in fixed direction.

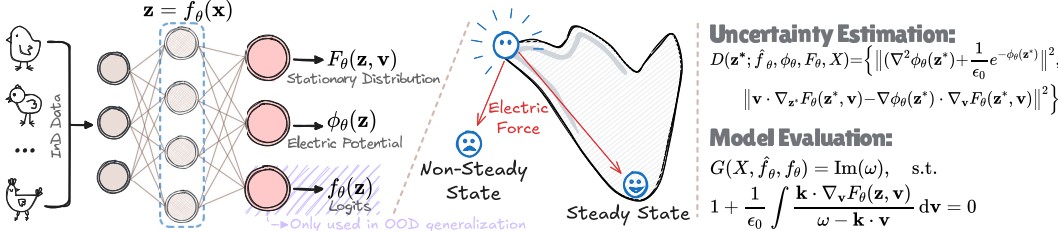

Figure 3: Illustration of our *CBD*-based framework. *CBD* is derived from the Vlasov-Poisson system and models the dynamics of high-level features under an electric field. The basin of attraction quantifies the region where features from the InD training set $X$ spontaneously converge to a steady state. Based on *CBD*, we propose two methods targeting OOD detection and OOD generalization.

this design and discuss its impact on the model's generalization ability, building on Theorem 2.5 and Corollary 2.6.

In summary, the contributions of this work are three-fold: *i)* Theoretically, we introduce an application of fields and potentials in OOD problems from two novel perspectives: the collective behavior dynamics of high-level features and complex perturbations under self-consistent electric fields, with supporting theories. *ii)* Methodologically, we propose physics-inspired **C**ollective **B**ehavior **D**ynamics (**CBD**) framework for OOD problems. It is simple, effective, and compatible with other methods. *iii)* Experimentally, our method outperforms or matches SoTA approaches across three OOD detection benchmarks and six OOD generalization benchmarks, and can be integrated with them for further improvement.

**Prior Arts.** We discuss related work and defer a concentrated account to Appendix A.

## 2 CBD: Collective Behavior Dynamics

Let $X = \{(\mathbf{x}, \mathbf{y})\}$ denote the training set, consisting of i.i.d. samples drawn from the joint distribution $\mathcal{P} = \mathcal{X} \times \mathcal{Y}$, where $\mathcal{X} \in \mathbb{R}^d$ represents the input space and $\mathcal{Y} = \{1, 2, \ldots, C\}$ is the output space. Consider a classification network parameterized by $\theta$, where the network maps inputs $\mathbf{x} \in \mathcal{X}$ to a probability distribution over the output space $\mathcal{Y}$. We illustrate our *CBD*-based approaches in Figure 3. In the following, we first present the formulation of the ***Vlasov-Poisson system*** (Section 2.1), and then introduce the details of ***basin of attraction*** (Section 2.2), followed by its application to OOD detection. Finally, we formulate the ***dispersion relation*** (Section 2.3) and demonstrate how *CBD* can be utilized for OOD generalization.

### 2.1 Formulation of Vlasov-Poisson System

The performance of classification networks typically relies on high-level feature extracted from individual inputs. However, this *overlooks the deep physical constraints* that exist between high-level features across InD data. To model the behavior of high-level features $\mathbf{z} = \hat{f}_\theta(\mathbf{x})$, where $\hat{f}_\theta(\cdot)$ denotes a feature extractor, we treat these features extracted from InD data as **charged particles** in the space $\mathbb{R}^Z$. Due to Coulomb interactions, these particles spontaneously move in space. In the absence of collisions and external magnetic fields, their dynamics can be effectively described by the Vlasov–Poisson system, which consists of two fundamental equations: the Vlasov equation and the Poisson equation. Specifically, the Vlasov equation is given by:

$$\frac{\partial F(\mathbf{z}, \mathbf{v}, t)}{\partial t} + \mathbf{v} \cdot \nabla_{\mathbf{z}} F(\mathbf{z}, \mathbf{v}, t) + E(\mathbf{z}, t) \cdot \nabla_{\mathbf{v}} F(\mathbf{z}, \mathbf{v}, t) = 0. \tag{1}$$

Eq.(1) preserves the Hamiltonian structure and energy conservation of the system, where $F(\mathbf{z}, \mathbf{v}, t) \in \mathbb{R}^Z \times \mathbb{R}^Z \times \mathbb{R}^+ \to \mathbb{R}^+$ is the **particle distribution function** at time $t$, defining the particle density at position $\mathbf{z}$ and velocity $\mathbf{v}$. $E(\mathbf{z}, t) \in \mathbb{R}^Z$ corresponds to the **electric field** generated by particle distribution (assume all particles have the same charge). $\nabla_{\mathbf{z}}$ and $\nabla_{\mathbf{v}}$ represent the gradient with respect to position and velocity, respectively. Eq.(1) describes macroscopic space-time changes.

The Poisson equation characterizes the relationship between the electric field $E(\mathbf{z}, t)$ and the particle distribution $F(\mathbf{z}, \mathbf{v}, t)$, and can be expressed as:

$$\nabla^2 \phi(\mathbf{z}, t) = -\frac{1}{\epsilon_0} \Big( \int F(\mathbf{z}, \mathbf{v}, t) \mathrm{d}\mathbf{v} - \rho_{\text{ion}}(\mathbf{z}) \Big), \quad E(\mathbf{z}, t) = -\nabla \phi(\mathbf{z}, t). \tag{2}$$

Here $\phi(\mathbf{z}, t) : \mathbb{R}^Z \times \mathbb{R}^+ \to \mathbb{R}^+$ denotes the **electric potential**, $\int F(\mathbf{z}, \mathbf{v}, t) \mathrm{d}\mathbf{v}$ the **charge density** (*i.e.*, the total charge at position $\mathbf{z}$), and $\epsilon_0 > 0$ the permittivity constant. $\rho_{\text{ion}}(\mathbf{z})$ denotes Ion background

density, assumed to be a spatially uniform constant. It serves to neutralize the system and ensure potential solution. Eq.(1) and Eq.(2) together describe particle dynamics under the electric field.

**Theorem 2.1.** *Consider $N$ Vlasov-Poisson systems with steady state distributions $F_i(\mathbf{z}, \mathbf{v}) = n_i(\mathbf{z})\,\delta(\mathbf{v} - \mathbf{v}_0)$, where each spatial density $n_i(\mathbf{z}) \in L^1(\mathbb{R}^Z) \cap L^\infty(\mathbb{R}^Z)$ represents the spatial density for the $i$-th system, and $L^1(\mathbb{R})$ and $L^\infty(\mathbb{R})$ denote respectively the spaces of integrable and essentially bounded functions on $\mathbb{R}$. The Dirac delta distribution $\delta(\mathbf{v} - \mathbf{v}_0)$ is centered at $\mathbf{v} = \mathbf{v}_0$, indicating a velocity distribution concentrated at that point. Assume that each $n_i(\mathbf{z})$ satisfies $\mathbf{v}_0 \cdot \nabla_{\mathbf{z}} n_i(\mathbf{z}) = 0$ and that the ion background density is given by $\rho_{ion,i}(\mathbf{z}) = n_i(\mathbf{z})$. If $n_i(\mathbf{z}) \neq n_j(\mathbf{z})$ for $i \neq j$, then the solutions $F_i$ are distinct.*

We prove that in different multiple systems, where particles share identical mass and initial velocity $\mathbf{v}_0$ but have distinct spatial density distributions, the steady state solutions (*i.e.*, $\frac{\partial F}{\partial t}=0$) are distinct. Using the provided Vlasov-Poisson equations, a neutralizing ion background, and a density gradient condition, we establish the result via distribution theory. The proof is deferred to the Appendix B.1.

## 2.2 Basin of Attraction and Application to OOD Detection

Building upon the concept of the Vlasov-Poisson system, we introduce the notion of a basin of attraction to characterize the system's dynamical behavior under both InD and OOD inputs. Let $F^*(\mathbf{z}, \mathbf{v})$ denote a **steady state solution** to the Vlasov–Poisson system. We define the basin of attraction $\mathcal{B}(F^*)$ as the set of initial distributions $F_0$ such that the time-moved solution asymptotically approaches $F^*$:

$$\mathcal{B}(F^*) = \left\{ F_0 \in \mathcal{F} \mid \lim_{t \to \infty} F(t, \cdot, \cdot) = F^* \text{ in some topology} \right\}. \tag{3}$$

Here, $\mathcal{F}$ denotes the admissible function space (*e.g.*, Sobolev or Wasserstein space). This asymptotic behavior can be equivalently characterized by the steady state form of the Vlasov–Poisson system,

$$\begin{cases} \mathbf{v} \cdot \nabla_{\mathbf{z}} F^*(\mathbf{z}, \mathbf{v}) + E^*(\mathbf{z}) \cdot \nabla_{\mathbf{v}} F^*(\mathbf{z}, \mathbf{v}) = 0, \\ \nabla^2 \phi^*(\mathbf{z}) = -\frac{1}{\epsilon_0} \left( \int F^*(\mathbf{z}, \mathbf{v}) \, d\mathbf{v} - \rho_{\text{ion}}(\mathbf{z}) \right), \quad E^*(\mathbf{z}) = -\nabla \phi^*(\mathbf{z}). \end{cases} \text{(time-independent)} \tag{4}$$

**Definition 2.2.** *Under thermodynamic equilibrium, the steady-state electric potential $\phi^*(\mathbf{z})$ satisfies the nonlinear Poisson–Boltzmann equation:*

$$\nabla^2 \phi^*(\mathbf{z}) = -\frac{1}{\epsilon_0} (e^{-\phi^*(\mathbf{z})} - \rho_{ion}(\mathbf{z})). \tag{5}$$

Eq.(5) describes the equilibrium electrostatic potential resulting from a Boltzmann-distributed charge density balanced against a fixed ion background. The derivation is provided in Appendix B.2.

**Corollary 2.3.** *Consider $\phi^*(\mathbf{z})$ be the steady state electric potential satisfying the nonlinear Poisson–Boltzmann equation. Then the basin of attraction's boundary $\partial\mathcal{B}$ can be implicitly represented as the zero level set of a scalar functional over $\phi^*(\mathbf{z})$, i.e.,*

$$\partial\mathcal{B} = \left\{ \mathbf{z} \in \mathbb{R}^Z \mid \mathcal{G}(\phi^*(\mathbf{z})) = 0 \right\}, \tag{6}$$

*where $\mathcal{G}$ is a differentiable function that depends on system configuration, such as energy thresholds, local potential curvature, or external field geometry. This representation enables differentiable modeling and learning of basin boundaries via neural function approximators.*

Corollary 2.3 establishes that the basin of attraction $\mathcal{B}$ associated with the steady state dynamics is fully characterized by the solution $\phi^*(\mathbf{z})$ to the nonlinear Poisson–Boltzmann equation, through a differentiable level-set condition $\mathcal{G}(\phi^*(\mathbf{z}))=0$. *It implies that the geometric and topological structure of $\mathcal{B}$, including its boundary $\partial\mathcal{B}$, is encoded in the electric potential $\phi^*$.* We defer the formal to Appendix B.3. Given the challenges of directly solving high-dimensional nonlinear PDE of Eq.(4) and the need for framework compatibility, we propose incorporating two parallel MLP branches following the feature extractor: $F_\theta$ for predicting the steady state distribution function $F^*$ and $\phi_\theta$ for estimating the steady-state potential $\phi^*$, both running in parallel with the linear classifier $f_\theta$. As shown in Figure 3. This architecture ensures *CBD* compatibility with most existing methods. Then we formulate a physics-informed loss, which enforces the consistency of the learned potential with the steady state field equation. *Because*

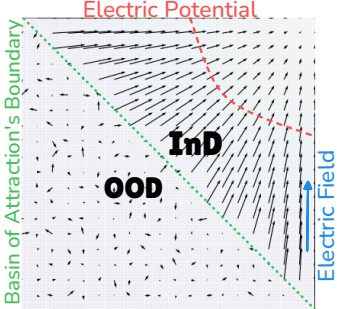

Figure 4: Electric field, potential, and basin of attraction's boundary. Particles inside converge to steady state; those outside behave chaotically.

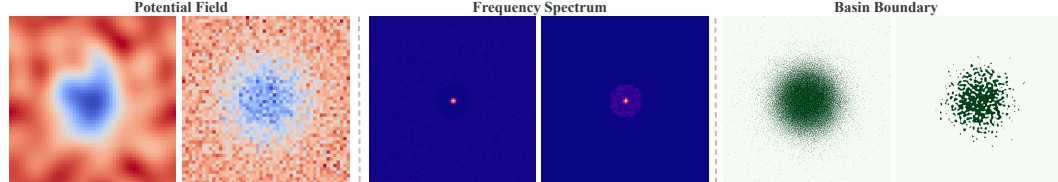

Figure 5: Effect of dispersion relation regularization on the learned potential field, its frequency spectrum, and the recovered basin of attraction. *Left column*: Predicted potential fields with (left) and without (right) dispersion relation loss. The regularized solution is smoother and more physically consistent. *Middle column*: Corresponding frequency spectrum. Dispersion relation regularization suppresses spurious high-frequency components, aligning the spectral structure with physical expectations. *Right column*: Estimated basin boundaries (contour of $\phi_\theta(\mathbf{z})$=const). Dispersion regularization leads to a wider and smoother basin, indicating improved global consistency and robustness.

*the Ion background density $\rho_{ion}(\mathbf{z})$ is constant, it introduces only a uniform offset and can be safely omitted from derivation.* Thus the resulting Poisson loss is

$$\mathcal{L}_{\text{Poisson}}(\mathbf{z}; \hat{f}_\theta, \phi_\theta) = \left\| \nabla^2 \phi_\theta(\mathbf{z}) + \frac{1}{\epsilon_0} e^{-\phi_\theta(\mathbf{z})} \right\|^2. \tag{7}$$

The steady state potential $\phi^*$ encapsulates the *overall* geometric and topological structure of the InD data, while also satisfying the Vlasov equation. So we get the following Vlasov loss:

$$\mathcal{L}_{\text{Vlasov}}(\mathbf{z}; \hat{f}_\theta, \phi_\theta, F_\theta) = \left\| \mathbf{v} \cdot \nabla_{\mathbf{z}} F_\theta(\mathbf{z}, \mathbf{v}) - \nabla \phi_\theta(\mathbf{z}) \cdot \nabla_{\mathbf{v}} F_\theta(\mathbf{z}, \mathbf{v}) \right\|^2. \tag{8}$$

Then we leverage $\mathcal{L}_{\text{Poisson}}$ and $\mathcal{L}_{\text{Vlasov}}$ as a physically grounded uncertainty score to identify whether $\mathbf{z}^* = \hat{f}(\mathbf{x}^*)$ lies inside the basin of attraction or corresponds to OOD behavior,

$$D(\mathbf{z}^*; \hat{f}_\theta, \phi_\theta, F_\theta, X) = \begin{cases} \text{InD} & \text{if } \mathcal{L}_{\text{Poisson}}(\mathbf{z}^*; \hat{f}_\theta, \phi_\theta) \leq \lambda_1 \text{ and } \mathcal{L}_{\text{Vlasov}}(\mathbf{z}^*; \hat{f}_\theta, \phi_\theta, F_\theta) \leq \lambda_2 \\ \text{OOD} & \text{otherwise} \end{cases}, \tag{9}$$

where $\lambda_1 > 0$ and $\lambda_2 > 0$ is thresholds and $\mathbf{x}^*$ is test data. This approach not only offers physical interpretability, with the score directly quantifying deviation from the governing PDE. But also geometric alignment with basin structure, as Corollary 2.3 demonstrates that $\mathcal{G}(\phi^*(\mathbf{z}))$ exhibits $\boxed{\text{sharp gradient}}$ near the boundary $\partial\mathcal{B}$, enabling robust OOD detection without statistical estimators.

## 2.3 Dispersion Relation and Application to OOD Generalization

OOD data's high-level features may fall outside the basin of attraction of InD data, preventing spontaneous transition from initial to steady states under given electric fields. From this perspective, we hypothesize that **network robustness is proportional to the range of the basin of attraction.** As the basin of attraction contracts, larger activity spaces emerge, increasing the probability of potential defects. As illustrated in the second image of the right column in Figure 5, without additional regularization, the learned boundary of the basin of attraction for exhibits fragmentation, demonstrating not only sensitivity to input data but also a diminished area (green area). Enlarge this basin implies that the system can accommodate larger initial deviations, without diverging or converging to alternative equilibria, thereby improving the robustness of the steady state.

We define the set of $\boxed{\text{initial perturbations } (f_0, \psi_0) = (F - F^*, \phi - \phi^*)}$. To investigate the collective behavior dynamics of perturbations, we employ **plane waves** for decomposition. We define the plane wave as $\psi(\mathbf{z}, t) = \tilde{\psi} e^{i(\mathbf{k} \cdot \mathbf{z} - \omega t)}$ and $f(\mathbf{z}, \mathbf{v}, t) = \tilde{f}(\mathbf{v}) e^{i(\mathbf{k} \cdot \mathbf{z} - \omega t)}$, where $\mathbf{k} \in \mathbb{R}^Z$ denotes the wave vector specifying the spatial frequency and direction (we treat $\mathbf{k}$ as a *random vector hyperparameter* in our experiments), and $\omega = \omega_r + i\omega_i$ is the complex frequency. The scalar $\tilde{\psi}$ and $\tilde{f}(\mathbf{v})$ represents the mode amplitude. This representation enables **any perturbation to be expressed as a superposition of decoupled modes**, and in systems exhibiting translational invariance, plane waves naturally arise as eigenfunctions, transforming PDE into algebraic forms. Substituting $\psi$ and $f$ into Eq.(4), we get the *perturbation collective behavior dynamics equations*:

$$\begin{cases} \partial_t f(\mathbf{z}, t) + \mathbf{v} \cdot \nabla_{\mathbf{z}} f(\mathbf{z}, t) + E^*(\mathbf{z}) \cdot \nabla_{\mathbf{v}} f(\mathbf{z}, t) = (\nabla_{\mathbf{v}} F^*(\mathbf{z}, \mathbf{v})) \cdot (\nabla_{\mathbf{z}} \psi(\mathbf{z}, t)), \\ \nabla^2 \psi(\mathbf{z}, t) = -\frac{1}{\epsilon_0} \int f(\mathbf{z}, t) \, d\mathbf{v}. \end{cases} \tag{10}$$

Substituting the plane wave into Eq.(10), yielding

$$\begin{cases} -i\omega \tilde{f}(\mathbf{v}) + i\mathbf{k} \cdot \mathbf{v} \tilde{f}(\mathbf{v}) + \left( -\nabla \phi^*(\mathbf{z}) \cdot \nabla_{\mathbf{v}} \tilde{f}(\mathbf{v}) \right) = \tilde{\psi}(\nabla_{\mathbf{v}} F^*(\mathbf{z}, \mathbf{v})) \cdot i\mathbf{k}, \\ -\|\mathbf{k}\|^2 \tilde{\psi} = -\frac{1}{\epsilon_0} \int \tilde{f}(\mathbf{v}) \, d\mathbf{v}. \end{cases} \tag{11}$$

Assuming a homogeneous steady state for simplicity (*i.e.*, $\phi^*(\mathbf{z})$ is constant and $F^*(\mathbf{z}, \mathbf{v}) = F^*(\mathbf{v})$), the electric field $E^*(\mathbf{z}) = 0$, and Eq.(11) simplifies to:

$$-i\omega \tilde{f}(\mathbf{v}) + i\mathbf{k} \cdot \mathbf{v}\tilde{f}(\mathbf{v}) = \tilde{\psi}(\nabla_{\mathbf{v}} F^*(\mathbf{z}, \mathbf{v})) \cdot i\mathbf{k}. \tag{12}$$

Solving for $\tilde{f}(\mathbf{v})$, we obtain:

$$\tilde{f}(\mathbf{v}) = \frac{\tilde{\psi}(\nabla_{\mathbf{v}} F^*(\mathbf{v})) \cdot \mathbf{k}}{\omega - \mathbf{k} \cdot \mathbf{v}}. \tag{13}$$

Integrating over velocity space and substituting into the Poisson equation, we get:

$$D(\omega, \mathbf{k}) = 1 + \frac{1}{\epsilon_0} \int \frac{\mathbf{k} \cdot \nabla_{\mathbf{v}} F^*(\mathbf{v})}{\omega - \mathbf{k} \cdot \mathbf{v}} \, d\mathbf{v} = 0. \tag{14}$$

The solutions of $D(\omega, \mathbf{k})$ is $\omega(\mathbf{k}) = \omega_r + i\omega_i$, which determine the stability of each mode. To assess stability, we define an energy-like quantity as the squared magnitude of the perturbation, $|\psi(\mathbf{z}, t)|^2$ (or alternatively $|f(\mathbf{z}, \mathbf{v}, t)|^2$, with analogous derivation steps following). Substituting $\omega(\mathbf{k}) = \omega_r + i\omega_i$, this becomes $|\psi(\mathbf{z}, t)|^2 = |\tilde{\psi}|^2 e^{2\omega_i t}$, since the exponential term $e^{i(\mathbf{k} \cdot \mathbf{z} - (\omega_r + i\omega_i)t)} = e^{i\mathbf{k} \cdot \mathbf{z}} e^{-i\omega_r t} e^{\omega_i t}$, and the magnitudes of the complex phases $e^{i\mathbf{k} \cdot \mathbf{z}}$ and $e^{-i\omega_r t}$ are unity. Thus, *when $\omega_i > 0$, the factor $e^{2\omega_i t}$ grows exponentially, causing the energy-like quantity to increase, potentially destabilizing the system and contracting the basin of attraction.* Conversely, *when $\omega_i < 0$, $e^{2\omega_i t}$ decays exponentially, reducing the energy-like quantity and facilitating recovery to the steady state.* In the nonlinear regime, perturbations decompose into multiple modes, and decaying modes ($\omega_i < 0$) can mitigate the growth of unstable modes ($\omega_i > 0$) through modal interactions, such as energy redistribution. Therefore, the imaginary part of $\omega$ can be used to evaluate the generalization ability of the model.

To enlarge the basin of attraction, we aim to suppress perturbations associated with positive growth rates, *i.e.*, those satisfying $\omega_i > 0$, as they contribute to instability and hinder recovery to equilibrium. To this end, we introduce the following dispersion relation loss that regularizes the consistency between the electric potential and the charge distribution in the *frequency domain*

$$\mathcal{L}_{\text{disp}}(\mathbf{z}; \hat{f}_\theta, \phi_\theta) = \sum_{\mathbf{k}} \left\| \|\mathbf{k}\|^2 \phi'_{\mathbf{k}}(\mathbf{z}) + \rho'_{\mathbf{k}}(\mathbf{z}) \right\|^2,$$
$$\text{s.t.} \quad \phi'_{\mathbf{k}}(\mathbf{z}) = \phi_\theta(\mathbf{z}) e^{-i\mathbf{k} \cdot \mathbf{z}}, \quad \rho'_{\mathbf{k}}(\mathbf{z}) = e^{-\phi_\theta(\mathbf{z})} e^{-i\mathbf{k} \cdot \mathbf{z}}. \tag{15}$$

This loss arises from the **Fourier-domain** formulation of the Poisson equation. Minimizing Eq.(15) therefore enforces the constraint across sampled wavevectors $\mathbf{k}$ (see Figure 2). From a dynamical perspective, satisfying this balance prevents the system from developing unstable frequency components, *which would otherwise drive the $D(\omega, \mathbf{k}) = 0$ toward solutions with $\omega_i > 0$, such imbalances act as implicit forcing terms that require growing temporal responses.* In contrast, minimizing the dispersion relation loss encourages configurations where $\omega_i < 0$, ensuring exponential decay of perturbation energy and improved return to steady state. This regularization thus enhances stability and expands the basin of attraction. Figure 5 presents a comparative analysis of predicted potential fields, frequency spectra, and estimated basin boundaries before (right) and after (left) incorporating the dispersion relation loss, demonstrating its superior efficacy in enlarging the basin of attraction. According to Eq.(15), the underlying theoretical rationale can be formulated as follows.

**Theorem 2.4.** *Define the approximate basin of attractor basin as the residual sublevel set*

$$\mathcal{B}_\theta := \left\{ \mathbf{z} \in \Omega \mid (\nabla^2 \phi_\theta(\mathbf{z}) + \frac{1}{\epsilon_0} e^{-\phi_\theta(\mathbf{z})})^2 < \lambda \right\}. \tag{16}$$

*Then, under mild regularity assumptions, increasing the spectral regularization strength $\lambda_{disp} > 0$ leads to a strictly larger basin measure:*

$$\text{meas}\left(\mathcal{B}_\theta^{(\lambda_{disp} > 0)}\right) \geq \text{meas}\left(\mathcal{B}_\theta^{(\lambda_{disp} = 0)}\right). \tag{17}$$

**Theorem 2.5.** *Let $\phi_\theta, F_\theta \in H^s(\mathbb{R}^Z), H^s(\mathbb{R}^Z \times \mathbb{R}^Z)$, respectively, with $s > Z/2$, approximate the true solutions $\phi^*, F^*$ of the Vlasov-Poisson system. Training data $(\mathbf{z}, \mathbf{v}) \sim \mathcal{D}$ are i.i.d. from $\Omega \times \mathbb{R}^Z$. The generalization error is: $\mathcal{E}_{gen}(\theta) = \mathbb{E}_{\mathbf{z}, \mathbf{v} \sim \mathcal{D}} \left[ \|\phi_\theta(\mathbf{z}) - \phi^*(\mathbf{z})\|^2 + \|F_\theta(\mathbf{z}, \mathbf{v}) - F^*(\mathbf{z}, \mathbf{v})\|^2 \right]$. The inclusion of $\mathcal{L}_{disp}$ reduces $\mathcal{E}_{gen}(\theta)$ by constraining the Sobolev norm of $\phi_\theta$, satisfying:*

$$\mathbb{E}_{\mathbf{z}, \mathbf{k}}[\mathcal{L}_{disp}] \geq c\|\phi_\theta\|_{H^2}^2 - C, \tag{18}$$

*for constants $c, C > 0$, thereby reducing the model complexity term in the generalization error bound.*

Table 1: OOD detection performance on CIFAR-10 and CIFAR-100 benchmarks, with the top three results highlighted. Complete results for all baseline methods are provided in Appendix E.

| Method | MINIST | | SSVHN | | Textures | | Places365 | | Average | |
|---|---|---|---|---|---|---|---|---|---|---|
| | FPR95↓ | AUROC↑ | FPR95↓ | AUROC↑ | FPR95↓ | AUROC↑ | FPR95↓ | AUROC↑ | FPR95↓ | AUROC↑ |
| *CIFAR-10 Benchmark* | | | | | | | | | | |
| OpenMax | 23.33±4.67 | 90.50±0.44 | 25.40±1.47 | 89.77±0.45 | 31.50±4.05 | 89.58±0.60 | 38.52±2.27 | 88.63±0.28 | 29.69±1.21 | 89.62±0.19 |
| ODIN | 23.83±12.34 | 95.24±1.96 | 68.61±0.52 | 84.58±0.77 | 67.70±11.06 | 86.94±2.26 | 70.36±6.96 | 85.07±1.24 | 57.62±4.24 | 87.96±0.61 |
| MDSEns | 1.30±0.51 | 99.17±0.41 | 74.34±1.04 | 66.56±0.58 | 76.07±0.17 | 77.40±0.28 | 94.16±0.33 | 52.47±0.15 | 61.47±0.48 | 73.90±0.27 |
| RMDS | 21.49±2.32 | 93.22±0.80 | 23.46±1.48 | 91.84±0.26 | 25.25±0.53 | 92.23±0.23 | 31.20±0.28 | 91.51±0.11 | 25.35±0.73 | 92.20±0.21 |
| Gram | 70.30±8.96 | 72.64±2.34 | 33.91±17.35 | 91.52±4.45 | 94.64±2.71 | 62.34±8.27 | 90.49±1.93 | 60.44±3.41 | 72.34±6.73 | 71.73±3.20 |
| ReAct | 33.77±18.00 | 92.81±3.03 | 50.23±15.98 | 89.12±3.19 | 51.42±11.42 | 89.38±1.49 | 44.20±3.35 | 90.35±0.78 | 44.90±8.37 | 90.42±1.41 |
| VIM | 18.36±1.42 | 94.76±0.38 | 19.29±0.41 | 94.50±0.48 | 21.14±1.83 | 95.15±0.34 | 41.43±2.17 | 89.49±0.39 | 25.05±0.52 | 93.48±0.24 |
| KNN | 20.05±1.36 | 94.26±0.38 | 22.60±1.26 | 92.67±0.30 | 24.06±0.55 | 93.16±0.24 | 30.38±0.63 | 91.77±0.23 | 24.27±0.40 | 92.96±0.14 |
| ASH | 70.00±10.56 | 83.16±4.66 | 83.64±6.48 | 73.46±6.41 | 84.59±1.74 | 77.45±2.39 | 77.89±7.28 | 79.89±3.69 | 79.03±4.22 | 78.49±2.58 |
| SHE | 42.22±20.59 | 90.43±4.76 | 62.74±4.01 | 86.38±1.32 | 84.60±5.30 | 81.57±1.21 | 76.36±5.32 | 82.89±1.22 | 66.48±5.98 | 85.32±1.43 |
| GEN | 23.00±7.75 | 93.83±2.14 | 28.14±2.59 | 91.97±0.66 | 40.74±6.61 | 90.14±0.76 | 47.03±3.22 | 89.46±0.65 | 34.73±1.58 | 91.35±0.69 |
| NAC-UE | 15.14±2.60 | 94.86±1.36 | 14.33±1.24 | 96.05±0.47 | 17.03±0.59 | 95.64±0.44 | 26.73±0.80 | 91.85±0.28 | 18.31±0.92 | 94.60±0.50 |
| *CBD*-De | 15.20±0.59 | 97.48±1.29 | 14.22±1.50 | 97.85±0.68 | 18.44±1.82 | 97.03±0.95 | 22.75±2.21 | 93.18±0.73 | 17.65±1.03 | 96.39±0.81 |
| *CIFAR-100 Benchmark* | | | | | | | | | | |
| OpenMax | 53.82±4.74 | 76.01±1.39 | 53.20±1.78 | 82.07±1.53 | 56.12±1.91 | 80.56±0.09 | 54.85±1.42 | 79.29±0.40 | 54.50±0.68 | 79.48±0.41 |
| ODIN | 45.94±3.29 | 83.79±1.31 | 67.41±3.88 | 74.54±0.76 | 62.37±2.96 | 79.33±1.08 | 59.71±0.92 | 79.45±0.26 | 58.86±0.79 | 79.28±0.21 |
| MDSEns | 2.83±0.86 | 98.21±0.78 | 82.57±2.58 | 53.76±1.63 | 84.94±0.83 | 69.75±1.14 | 96.61±0.17 | 42.27±0.73 | 66.74±1.04 | 66.00±0.69 |
| RMDS | 52.05±6.28 | 79.74±2.49 | 51.65±3.68 | 84.89±1.10 | 53.99±1.06 | 83.65±0.51 | 53.57±0.43 | 83.40±0.46 | 52.81±0.63 | 82.92±0.42 |
| Gram | 53.53±7.45 | 80.71±4.15 | 20.06±1.96 | 95.55±0.60 | 89.51±2.54 | 70.79±1.32 | 94.67±0.60 | 46.38±1.21 | 64.44±2.37 | 73.36±1.08 |
| ReAct | 56.04±5.66 | 78.37±1.59 | 50.41±2.02 | 83.01±0.97 | 55.04±0.82 | 80.15±0.46 | 55.30±0.41 | 80.03±0.11 | 54.20±1.56 | 80.39±0.49 |
| VIM | 48.32±1.07 | 81.89±1.02 | 46.22±5.46 | 83.14±3.71 | 46.86±2.29 | 85.91±0.78 | 61.57±0.77 | 75.85±0.37 | 50.74±1.00 | 81.70±0.62 |
| KNN | 48.58±4.67 | 82.36±1.52 | 51.75±3.12 | 84.15±1.09 | 53.56±2.32 | 83.66±0.83 | 60.70±1.03 | 79.43±0.47 | 53.65±0.28 | 82.40±0.17 |
| ASH | 66.58±3.88 | 77.23±0.46 | 46.00±2.67 | 85.60±1.40 | 61.27±2.74 | 80.72±0.70 | 62.95±0.99 | 78.76±0.16 | 59.20±2.46 | 80.58±0.66 |
| SHE | 58.78±2.70 | 76.76±1.07 | 59.15±7.61 | 80.97±3.98 | 73.29±3.22 | 73.64±1.28 | 65.24±0.98 | 76.30±0.51 | 64.12±2.70 | 76.92±1.16 |
| GEN | 53.92±5.71 | 78.29±2.05 | 55.45±2.76 | 81.41±1.50 | 61.23±1.40 | 78.74±0.81 | 56.25±1.01 | 80.28±0.27 | 56.71±1.59 | 79.68±0.75 |
| NAC-UE | 21.97±6.62 | 93.15±1.63 | 24.39±4.66 | 92.40±1.26 | 40.65±1.94 | 89.32±0.55 | 73.57±1.16 | 73.05±0.68 | 40.14±1.86 | 86.98±0.37 |
| *CBD*-De | 27.93±7.73 | 96.29±1.81 | 18.72±6.12 | 95.80±2.05 | 37.34±2.30 | 90.02±1.52 | 55.23±0.88 | 81.50±1.16 | 34.81±1.60 | 90.90±0.68 |

**Corollary 2.6.** *Under the same assumptions with Theorem 2.5, let $\mathcal{H} = \{(\phi_\theta, F_\theta) : \theta \in \Theta\}$ be the hypothesis class. The generalization bound is:*

$$\mathcal{E}_{gen}(\theta) \leq \mathcal{E}_{train}(\theta) + 2Rad(\mathcal{H}) + O(\sqrt{\log(1/\delta)/n}), \tag{19}$$

*where $Rad(\mathcal{H})$ is the Rademacher complexity, and $n$ is the sample size. The inclusion of $\mathcal{L}_{disp}$ reduces $Rad(\mathcal{H})$ by restricting $\phi_\theta$ to a subset $\mathcal{H}_\phi \subset H^2$, tightening the generalization bound.*

Theorem 2.4 proves that incorporating the loss $\mathcal{L}_{\text{disp}}$ enlarges the basin of attraction. Theorem 2.5 further establishes that this loss term decreases the generalization error, and Corollary 2.6 presents the resulting tightening generalization bound. Proofs are deferred to Appendix B.4, B.5 and B.6. For OOD detection, the loss is defined as $\mathcal{L}_{\text{De}} \triangleq \mathcal{L}_{\text{Vlasov}} + \mathcal{L}_{\text{Poisson}}$. For OOD generalization, we enhance the training by incorporating $\mathcal{L}_{\text{disp}}$ along with the standard classification loss [81], yielding the overall loss $\mathcal{L}_{\text{Gen}} \triangleq \mathcal{L}_{\text{Classify}} + \alpha(\mathcal{L}_{\text{Vlasov}} + \mathcal{L}_{\text{Poisson}}) + \beta\mathcal{L}_{\text{disp}}$, which ensures robust performance on the primary task. For detailed pseudocode, refer to Appendix C.

## 3 Experiments

**Steup.** For OOD detection, following the latest OpenOOD [94], we evaluated our *CBD*-De on three InD datasets: CIFAR-10, CIFAR-100 [38], and ImageNet-1k [12]. For CIFAR-10 and CIFAR-100, we employed ResNet-18 and assessed performance against four OOD datasets: MNIST [13], SVHN [59], Textures [10], and Places365 [101]. For ImageNet-1k, we utilized pre-trained ResNet-50 and Vit-b16 models, evaluating against three OOD datasets: iNaturalist [79], Textures, and OpenImage-O [84]. We compared our approach with 22 SoTA OOD detection methods, with comprehensive results provided in Appendix E. For OOD generalization, we adhere to the Domainbed [23] in our experimental setup and use BCE as classification loss, evaluating our *CBD*-Gen on five datasets: VLCS [16], PACS [45], OfficeHome [82], TerraInc [5], and DomainNet [60]. For each dataset, we report the leave-one-out test accuracy, averaged across splits where one domain serves as the test set and the remaining domains as the training set. Our method is compared against 18 SoTA baselines, with comprehensive results available in Appendix F.

**Metrics.** For OOD detection, we employed two threshold-free evaluation metrics, consistent with established conventions: *i)* FPR95, the false-positive-rate of OOD data when the true-positive-rate of ID data is at 95%, and *ii)* AUROC, the area under the receiver operating characteristic curve. For OOD generalization, we adhere to the standard practice of using accuracy as the evaluation metric.

**Implementation Details.** In the interest of brevity, we defer details such as training procedures, hyperparameter configurations, and baseline methodologies to Appendix D.

Table 2: Performance of OOD detection (AUROC) on ImageNet-1k, with full results in Appendix E.

| Dataset | Backbone | OpenMax | MDS | RMDS | ReAct | VIM | KNN | ASH | SHE | GEN | NAC-UE | *CBD*-De |
|---|---|---|---|---|---|---|---|---|---|---|---|---|
| iNaturalist | ResNet-50 | 92.05 | 63.67 | 87.24 | 96.34 | 89.56 | 86.41 | 97.07 | 92.65 | 92.44 | 96.52 | **97.06** |
| | Vit-b16 | 94.93 | 96.01 | 96.10 | 86.11 | 95.72 | 91.46 | 50.62 | 93.57 | 93.54 | 93.72 | **97.26** |
| | Average | 93.49 | 79.84 | 91.67 | 91.23 | 92.64 | 88.94 | 73.85 | 93.11 | 92.99 | 95.12 | **97.16** |
| OpenImage-O | ResNet-50 | 87.62 | 69.27 | 85.84 | 91.87 | 90.50 | 87.04 | 93.26 | 86.52 | 89.26 | 91.45 | **92.65** |
| | Vit-b16 | 87.36 | 92.38 | 92.32 | 84.29 | 92.18 | 89.86 | 55.51 | 91.04 | 90.27 | 91.58 | **91.89** |
| | Average | 87.49 | 80.83 | 89.08 | 88.08 | 91.34 | 88.45 | 74.39 | 88.78 | 89.77 | 91.52 | **92.27** |
| Textures | ResNet-50 | 88.10 | 89.80 | 86.08 | 92.79 | 97.97 | 97.09 | 96.90 | 93.60 | 87.59 | 97.90 | **98.31** |
| | Vit-b16 | 85.52 | 89.41 | 89.38 | 86.66 | 90.61 | 91.12 | 48.53 | 92.65 | 90.23 | 94.17 | **94.30** |
| | Average | 86.81 | 89.61 | 87.73 | 89.73 | 94.29 | 94.11 | 72.72 | 93.13 | 88.91 | 96.04 | **96.31** |

Table 3: OOD generalization performance on DomainBed, with complete results in Appendix F.

| Algorithm | Mixstyle | MMD | ARM | MTL | MLDG | Mixup | CORAL | SAM | SAGM | GGA | *CBD*-Gen |
|---|---|---|---|---|---|---|---|---|---|---|---|
| PACS | 85.2±0.3 | 84.7±0.5 | 85.1±0.4 | 84.6±0.5 | 84.9±1.0 | 84.6±0.6 | 86.2±0.3 | 85.8±0.2 | 86.6±0.2 | 87.3±0.4 | **87.7±0.6** |
| VLCS | 77.9±0.5 | 77.5±0.9 | 77.6±0.3 | 77.2±0.4 | 77.2±0.4 | 77.4±0.6 | 78.8±0.6 | 79.4±0.1 | 80.0±0.3 | 79.9±0.4 | **80.5±0.5** |
| OfficeHome | 60.4±0.3 | 66.3±0.1 | 64.8±0.3 | 66.4±0.5 | 66.8±0.6 | 68.1±0.3 | 68.7±0.3 | 69.6±0.1 | 70.1±0.2 | 70.1±0.2 | **71.4±0.3** |
| TerraInc | 44.0±0.7 | 42.2±1.6 | 45.5±0.4 | 45.6±1.2 | 47.7±0.2 | 47.9±0.8 | 48.6±1.0 | 43.3±0.7 | 48.8±0.9 | 50.6±0.1 | 50.5±0.1 |
| *Avg.* | 66.9 | 67.7 | 68.3 | 68.5 | 69.2 | 69.5 | 70.3 | 69.3 | 71.4 | 71.7 | **72.5** |
| DomainNet | 34.0±0.1 | 23.4±9.5 | 35.5±0.2 | 40.6±0.1 | 41.2±0.1 | 40.3±0.1 | 41.5±0.1 | 44.3±0.2 | 45.0±0.2 | 45.2±0.2 | **45.9±0.4** |
| *Total* | 60.3 | 58.8 | 61.7 | 62.9 | 63.6 | 63.4 | 64.5 | 64.5 | 66.1 | 66.3 | **67.2** |

**Quantitative results on OOD detection.** Tables 1 and 2 present the results of our *CBD*-De on the CIFAR and ImageNet datasets, compared against 22 SoTA methods. Confidence intervals in the tables reflect mean ± standard error over three independent test runs. Our *CBD*-De consistently outperforms all SoTA methods in average performance, achieving record-

Table 4: Evaluation of *CBD*-Gen's compatibility with advanced OOD generalization methods.

| | Methods | PACS | VLCS | OfficeHome | TerraInc | *Avg.* |
|---|---|---|---|---|---|---|
| SAM | baseline | 85.8 | 79.4 | 69.6 | 43.3 | 69.3 |
| | *with CBD*-Gen | 88.4 | 80.7 | 71.7 | 51.0 | 73.0 |
| | △ | (+2.6) | (+1.3) | (+2.1) | (+7.7) | (+3.7) |
| SAGM | baseline | 86.6 | 80.0 | 70.1 | 48.8 | 71.4 |
| | *with CBD*-Gen | 88.2 | 80.7 | 71.5 | 50.8 | 72.8 |
| | △ | (+1.6) | (+0.7) | (+1.4) | (+2.0) | (+1.4) |

breaking results across three benchmarks. Specifically, on CIFAR-10 and CIFAR-100, *CBD*-De reduces the FPR95 by 0.66% and 5.33%, respectively, and improves the AUROC by 1.79% and 3.92% compared to the most competitive baseline [52]. On the large-scale ImageNet-1k dataset, *CBD*-De enhances AUROC scores across various backbones and OOD datasets, with average AUROC improvements of 2.04%, 0.75%, and 0.27% over the strongest competitor.

**Quantitative results on OOD generalization.** Table 3 reports the average OOD accuracies of SoTA generalization methods across five benchmarks. For fair comparison, all methods are trained on the same amount of data. While additional data augmentations could further improve performance, they are not applied to isolate the effect of the core methods. The average performance of *CBD*-Gen across all evaluated datasets surpasses that of the most competitive benchmark by 0.9%. Specifically, on the large-scale DomainNet dataset, comprising 586,575 images across six domains, *CBD*-Gen achieves a 0.7% improvement over the leading method. Furthermore, *CBD*-Gen significantly outperforms the current SoTA on the OfficeHome dataset. We attribute this to its use of electric field modeling, which enhances the network's ability to capture discriminative features from limited data. In particular, the electric potential exhibits sharp gradient transitions near the basin of attraction's boundary, leading to larger inter-class margins. This effect is beneficial for OfficeHome, where each class contains only ∼240 samples on average, and substantially fewer than the 1,400+ samples per class available in other datasets, highlighting *CBD*-Gen's strength in low-data regimes.

**Compatibility results.** We further investigated the potential of *CBD*-Gen by integrating it with two advanced algorithms: SAM [17] and SAGM [86]. During training, we incorporated two MLP branches after the feature extractor and incorporate additional $\mathcal{L}_{\text{Poisson}}$, $\mathcal{L}_{\text{Vlasov}}$ and $\mathcal{L}_{\text{disp}}$ losses into the existing algorithmic frameworks. These branches are removed during testing. Results are shown in Table 4. Notably, *CBD*-Gen maintains consistent and superior performance across multiple datasets when combined with other methods, further validating its strong generalizability and compatibility. This demonstrates that *CBD*-Gen is not only effective within traditional training paradigms but can also function as an enhancement module that integrates seamlessly with various advanced methods, exhibiting extensive adaptation capabilities.

**Text OOD Detection.** To demonstrate the modality-agnostic nature of our post-hoc OOD detection method *CBD-De*, we further conduct experiments on text classification benchmarks. Following the evaluation protocol established in [4], we evaluate our method on standard text OOD detection datasets. Specifically, we adopt the widely used SST-2 [71], IMDB [54], and Yelp [99] datasets,

Table 4: Performance of OOD detection (AUROC) on text data.

| ID | OOD | MSP | Energy | GradNorm | KLM | ReAct | DICE | KNN | ViM | CBD-De |
|---|---|---|---|---|---|---|---|---|---|---|
| SST-2 | IMDB | 83.2±1.4 | 82.7±2.2 | 70.3±2.3 | 55.0±2.7 | 83.3±2.4 | 34.5±10.7 | 87.2±1.7 | 83.9±3.3 | **89.0±2.2** |
| SST-2 | Yelp | 75.7±2.2 | 75.0±3.1 | 61.3±2.7 | 51.3±3.0 | 75.7±3.4 | 35.4±8.4 | 87.8±0.4 | 80.1±2.8 | **89.3±1.4** |
| Yelp | IMDB | 79.5±0.5 | 79.2±1.6 | 71.7±1.9 | 38.6±1.3 | 79.5±1.6 | 26.8±5.1 | 84.7±0.8 | 88.6±0.7 | **90.2±0.6** |

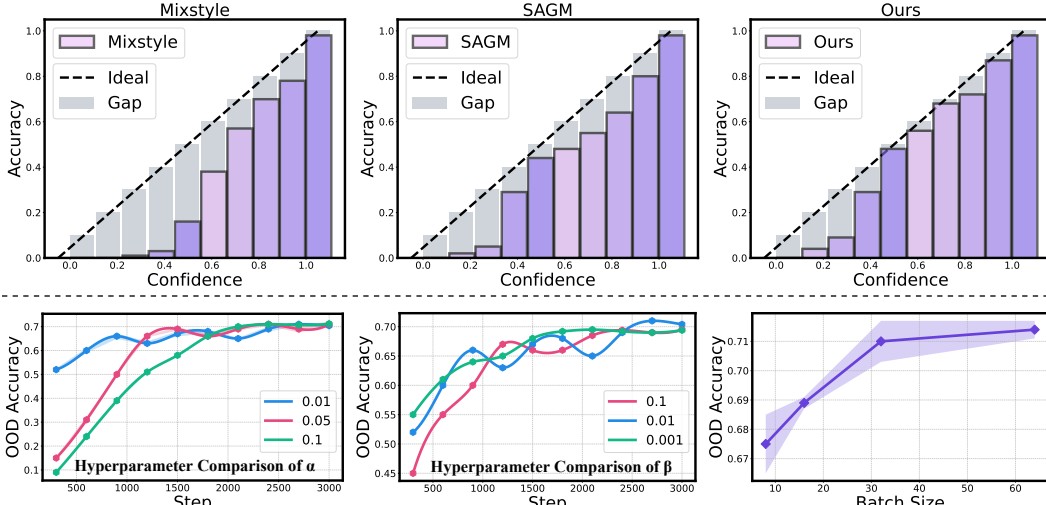

Figure 6: Confidence calibration comparison and hyperparameter analysis on the OfficeHome dataset.

where each dataset alternates as InD and OOD data. As shown in Table 4, our method achieves competitive performance compared to established baselines. These results confirm that *CBD-De*, though originally designed for visual OOD detection, generalizes effectively to text modalities without requiring any modality-specific adaptations.

**Confidence calibration analysis.** We evaluate the confidence calibration performance of Mixstyle [102], SAGM [86], and our proposed *CBD*-Gen on the OfficeHome dataset, with results visualized via reliability histograms in Figure 6 (upper row). Predictions are divided into ten bins based on confidence levels, and the mean accuracy is calculated for each bin. An optimally calibrated model should exhibit a diagonal pattern in the reliability diagram, as highlighted by the gray bars. We observe that Mixstyle exhibits poor calibration in low confidence intervals (0 to 0.5), likely due to these data augmentation-based methods struggling to generate OOD samples with significant distribution shifts. In contrast, while SAGM outperforms Mixstyle in these low intervals, it underperforms in mid-confidence intervals (0.5 to 0.8), possibly because these sharpness-aware minimization methods overly smooths samples with moderate gradients, skewing their probabilities toward the extremes. Our proposed *CBD*-Gen, however, excels in both low and mid-confidence intervals. This advantage may arise from decomposing perturbations into multiple plane waves, enabling two key operations: *i)* perturbing random frequency components to indirectly create broader OOD data, addressing calibration deficits in low-confidence regions, and *ii)* optimizing the basin of attraction's boundary, defined by a continuous PDE over InD data, ensuring smooth and consistent parameter adjustments for accurate probability estimation in mid-confidence intervals.

**Hyperparameter analysis.** To maintain clarity, we primarily use the OfficeHome dataset to investigate the impact of two loss-related hyperparameters in *CBD*-Gen, *i.e.*, $\alpha$ and $\beta$, as well as the effect of training batch size on model generalization. As shown in the bottom row of Figure 6, we first vary $\alpha$ while fixing $\beta = 0.01$. The results indicate that model performance remains stable over a broad range of $\alpha$ values (0.01 to 0.1), suggesting that *CBD*-Gen is robust to changes in the relative weighting of Poisson and Vlasov losses within this interval. Next, fixing $\alpha=0.01$, we examine the influence of $\beta$ and observe that generalization performance degrades slightly at both low ($\beta=0.001$) and high ($\beta=0.1$) values. A small $\beta$ may weaken the model's ability to enlarge the basin of attraction's boundary, limiting generalization, while an overly large $\beta$ could lead to excessive emphasis on matching plane wave modes, at the expense of accurate electric field modeling, ultimately impairing classification performance. We also investigate the effect of batch size on *CBD*-Gen's performance, based on the assumption that accurate modeling of electric fields and potentials from InD data benefits

from larger batches. As shown in Figure 6, generalization performance improves with increasing batch size. Batch sizes between [32, 64] offer strong, potentially optimal results, though hardware constraints limit evaluation at larger scales.

**Plane wave analysis.** Experimentally, the wavevector $\mathbf{k}$ in Eq(15) is sampled from a Gaussian distribution $\mathbf{k} \sim \mathcal{N}(0, \sigma^2 \mathbf{I})$ for each batch (see pseudocode in Appendix C). The scaling factor $\sigma$ controls the spread of sampling in the frequency space and directly affects the strength of dispersion relation regularization. Additionally, the wavenumber $N$ determines the discretization resolution in the frequency domain, significantly impacting both the stability of regularization and computational efficiency. We conducted systematic ablation studies on different values of $\sigma$ and $N$. As shown in Figure 7, increasing $N$ leads to consistent improvements in generalization performance (from blue to red/orange sample points). This is attributed

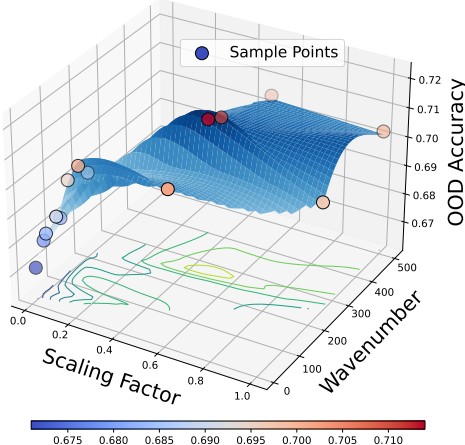

Figure 7: Comparative analysis of wavenumber $N$ and scaling factor $\sigma$ on OOD generalization task using the OfficeHome dataset.

to a finer and more comprehensive frequency decomposition, enabling the model to better capture and adapt to various domain shifts. However, excessively large $N$ degrades training efficiency. To balance performance and efficiency, we set $N=250$ in the above experiments. Furthermore, comparative analysis of various scaling factors revealed that both insufficient and excessive sampling ranges degrade model generalization performance. Smaller scaling factors preserve excessive high-frequency components, introducing non-physical oscillations in the potential field, while larger factors over-smooth the field, eliminating critical physical structural information. Experimental results demonstrate that $\sigma=0.5$ yields optimal performance, balancing physical consistency with discriminative feature preservation.

**Limitations.** We would like to discuss several limitations identified in our study. First, there is an opportunity to enhance *CBD* by reformulating the distribution and potential prediction heads as a multi-step PDE solver. Such a design could improve numerical stability and provide a more faithful approximation of the underlying dynamics, potentially leading to greater precision compared to directly solving for the steady state. However, this reformulation may introduce additional parameters, hyperparameter sensitivity, and computational overhead, which should be carefully balanced against the expected gains. We provide an illustrative experiment in the Appendix D.8 to demonstrate this limitation. Second, the *CBD* algorithm requires extensive storage of first- and second-order intermediate gradients, resulting in significant GPU memory demands. This not only increases training costs but also restricts batch sizes and model scalability in practice. This constraint could limit its applicability to large-scale or high-resolution tasks unless more memory-efficient optimization strategies are developed, such as gradient checkpointing or mixed-precision computation.

## 4 Conclusion

In this work, we present a physics-inspired collective behavior dynamics perspective to address the OOD problem. We conceptualize high-level features as charged particles and explicitly model their collective behavior under a self-consistent electric field using a Vlasov-Poisson system of dynamical equations. By solving for InD electric potential and distribution functions, we link the InD boundary to the system's steady-state basin of attraction, enabling effective OOD detection. Furthermore, we analyze the collective behavior dynamics of perturbations within the system and resolve the OOD generalization problem by enlarging the boundaries of the basin of attraction. Extensive experiments across diverse tasks, benchmarks demonstrate our *CBD*'s advantages over current leading methods. Following this line of thought, we hope our perspective inspires the community to consider physical dynamical behaviors in OOD problems, which naturally aligns with principles of interpretability.

## Acknowledgments

This work was supported by the National Science Fund for Distinguished Young Scholars, PR China under Grant No.62025601, Transformation Foundation of Tianfu Jincheng Laboratory (2025ZH013), National Natural Science Foundation of China under Grant U24A20341 and Sichuan Province Innovative Talent Funding Project for Postdoctoral Fellows under Grant BX202512.

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

# Appendix

## Table of Contents

# A   Prior Arts

**OOD detection** [27] aims to identify inputs that significantly deviate from the training data distribution, a task critical to ensuring model reliability and safety in real-world applications. Existing approaches can be broadly categorized into post-hoc and prior-based methods. Prior-based methods integrate OOD awareness into the training phase by modifying training strategies [30], loss functions [29, 55], or model architectures [78]. However, prior-based methods often require access to OOD samples or carefully crafted synthetic anomalies during training, which may not be feasible or generalizable across deployment scenarios. Additionally, such methods typically involve substantial modifications to the training pipeline or model architecture, increasing computational cost and limiting applicability to pre-trained models. In contrast, post-hoc methods operate on models already trained on InD data, making them more practical and flexible for real-world deployment. These methods rely on analyzing model responses, such as confidence scores [27, 100], output gradients [31], or intermediate activations [42], to estimate the likelihood that an input belongs to the training distribution. Their plug-and-play nature and compatibility with existing models make post-hoc approaches particularly appealing for scalable OOD detection. Our proposed *CBD*-De builds upon a pre-trained model by training two additional MLP branches with InD data, while **freezing** all parameters of the pre-trained model (including both feature extractor and classifier), which categorizes it as a post-hoc method.

**OOD generalization** [48] focuses on improving model performance on unseen distributions without requiring access to OOD data during training. It aims to learn representations that remain effective under distributional shifts, enabling better generalization to unknown environments. Existing OOD generalization approaches can be broadly categorized into three groups: data augmentation, representation learning, and model regularization. Data augmentation methods [102, 11] simulate distribution shifts by applying input transformations or generating synthetic domains. Representation learning methods [1, 69, 35, 90, 88] aim to learn domain-invariant or domain-discriminative features. Model regularization methods [33, 64] introduce constraints during optimization to improve robustness. While OOD generalization does not require OOD data, it often assumes access to multiple training environments or sufficient data diversity, which may not always be available in practice. By modeling the field or potential of InD data, our approach demonstrates superior performance over existing methods on datasets with limited class samples, as substantiated by comprehensive experiments.

**Vlasov–Poisson system** [83] is a fundamental kinetic model describing the evolution of a collisionless particle distribution under a self-consistent field. Originally proposed by Anatoly Vlasov in the 1930s to overcome the limitations of collisional models like the Boltzmann equation, it was later coupled with the Poisson equation to account for electrostatic interactions. Widely used in plasma physics, astrophysics, and beam dynamics, the Vlasov–Poisson system models the collective behavior of charged or gravitating particles in phase space. Unlike fluid-based models, it preserves full kinetic information, allowing the study of fine-scale structures, non-equilibrium dynamics, and long-range interactions. Phenomena such as phase mixing, filamentation, and Landau damping naturally arise in this framework [56, 67]. Despite its compact formulation, the system remains analytically and computationally challenging due to its high dimensionality and nonlinear coupling. Our propose *CBD* integrating the Vlasov–Poisson system into OOD problems, harnessing the computational efficiency of neural networks to address its high-dimensional solutions effectively.

**Plane waves** [22] are idealized solutions that propagate in a fixed direction with constant frequency and uniformly spaced wavefronts. Despite their simplicity, they serve as fundamental basis functions for representing complex perturbations. Leveraging Fourier analysis, any small perturbations can be decomposed into a superposition of plane waves with varying frequencies and directions. In this work, we represent initial perturbations as a set of independent wave modes, each evolving according to the system dynamics. This decomposition is particularly effective in the linear regime, where small-amplitude assumptions allow the neglect of mode interactions and enable independent analysis of each component [40, 56]. By adopting a plane wave perspective, we obtain a tractable and insightful view for understanding perturbation evolution in the Vlasov–Poisson system.

**Contribution.** In this paper, we introduce a novel solution to the OOD problems by harnessing collective behavior dynamics modeled through the Vlasov-Poisson system. Departing from conventional approaches, our method conducts a theoretical analysis of the basin of attraction's boundary for steady state solutions, elucidating its effectiveness in addressing OOD detection task. Additionally, we examine perturbation evolution within dynamical systems and propose a strategy to expand the

basin of attraction, thereby tackling OOD generalization challenges. Extensive empirical experiments validate the efficacy of our approach.

# B  Proof

## B.1  Formal Proof of Theorem 2.1

Consider the Vlasov-Poisson system in $Z$-dimensional space:

$$\frac{\partial F(\mathbf{z}, \mathbf{v}, t)}{\partial t} + \mathbf{v} \cdot \nabla_{\mathbf{z}} F(\mathbf{z}, \mathbf{v}, t) + \mathbf{E}(\mathbf{z}, t) \cdot \nabla_{\mathbf{v}} F(\mathbf{z}, \mathbf{v}, t) = 0, \tag{20}$$

$$\nabla^2 \phi(\mathbf{z}, t) = -\frac{\int F(\mathbf{z}, \mathbf{v}, t) \, d\mathbf{v}}{\epsilon_0}, \quad \mathbf{E}(\mathbf{z}, t) = -\nabla \phi(\mathbf{z}, t), \tag{21}$$

where:

- $\mathbf{z}, \mathbf{v} \in \mathbb{R}^Z$ are the spatial and velocity coordinates, respectively.
- $F(\mathbf{z}, \mathbf{v}, t) : \mathbb{R}^Z \times \mathbb{R}^Z \times \mathbb{R}^+ \to \mathbb{R}^+$ is the particle distribution function.
- $\mathbf{E}(\mathbf{z}, t) \in \mathbb{R}^Z$ is the electric field.
- $\phi(\mathbf{z}, t) : \mathbb{R}^Z \times \mathbb{R}^+ \to \mathbb{R}$ is the electric potential.
- $\epsilon_0 > 0$ is the permittivity constant.

The particle density is:

$$\rho(\mathbf{z}, t) = \int F(\mathbf{z}, \mathbf{v}, t) \, d\mathbf{v}. \tag{22}$$

We study $N$ systems indexed by $i = 1, \ldots, N$, each with:

- Particles of identical mass (normalized to 1) and velocity $\mathbf{v}_0 \in \mathbb{R}^Z$.
- Spatial density $n_i(\mathbf{z}) \in L^1(\mathbb{R}^Z) \cap L^\infty(\mathbb{R}^Z)$, with $n_i(\mathbf{z}) \neq n_j(\mathbf{z})$ for $i \neq j$.
- Distribution $F_i(\mathbf{z}, \mathbf{v}, t) = n_i(\mathbf{z}) \delta(\mathbf{v} - \mathbf{v}_0)$, where $\delta(\mathbf{v} - \mathbf{v}_0)$ is the Dirac delta distribution in $\mathbb{R}^Z$.
- Ion background density $\rho_{\text{ion}, i}(\mathbf{z}) = n_i(\mathbf{z})$.

A steady state solution satisfies $\frac{\partial F_i}{\partial t} = 0$. We aim to prove that the steady state solutions $F_i$ are distinct for distinct $n_i(\mathbf{z})$.

*Assumption* B.1.  For each system $i$:

1. $\rho_{\text{ion}, i}(\mathbf{z}) = n_i(\mathbf{z})$, ensuring $\mathbf{E} = 0$.

2. $\mathbf{v}_0 \cdot \nabla_{\mathbf{z}} n_i(\mathbf{z}) = 0$, ensuring $F_i$ is steady state.

**Proof.** Let us consider two systems $i$ and $j$ with steady state distributions:

$$F_i(\mathbf{z}, \mathbf{v}) = n_i(\mathbf{z}) \delta(\mathbf{v} - \mathbf{v}_0), \quad F_j(\mathbf{z}, \mathbf{v}) = n_j(\mathbf{z}) \delta(\mathbf{v} - \mathbf{v}_0). \tag{23}$$

We need to show that if $n_i(\mathbf{z}) \neq n_j(\mathbf{z})$, then $F_i \neq F_j$ in the sense of distributions.

First, verify that $F_i$ and $F_j$ are steady state solutions under the given assumptions. With $\rho_{\text{ion}, i}(\mathbf{z}) = n_i(\mathbf{z})$, compute the particle density:

$$\rho_i(\mathbf{z}) = \int F_i(\mathbf{z}, \mathbf{v}) \, d\mathbf{v} = \int n_i(\mathbf{z}) \delta(\mathbf{v} - \mathbf{v}_0) \, d\mathbf{v} = n_i(\mathbf{z}). \tag{24}$$

The Poisson equation becomes:

$$\nabla^2 \phi_i(\mathbf{z}) = -\frac{\rho_i(\mathbf{z})}{\epsilon_0} = -\frac{n_i(\mathbf{z})}{\epsilon_0}. \tag{25}$$

Since $\rho_{\text{ion}, i}(\mathbf{z}) = n_i(\mathbf{z})$, the total charge density is:

$$\rho_{\text{total}, i}(\mathbf{z}) = \rho_{\text{ion}, i}(\mathbf{z}) - \rho_i(\mathbf{z}) = n_i(\mathbf{z}) - n_i(\mathbf{z}) = 0. \tag{26}$$

Thus:

$$\nabla^2 \phi_i(\mathbf{z}) = -\frac{\rho_{\text{total},i}(\mathbf{z})}{\epsilon_0} = 0. \tag{27}$$

Assuming boundary conditions such that $\phi_i \to 0$ as $|\mathbf{z}| \to \infty$, the solution is $\phi_i(\mathbf{z}) = 0$, so:

$$\mathbf{E}_i(\mathbf{z}) = -\nabla \phi_i(\mathbf{z}) = 0. \tag{28}$$

Similarly, $\mathbf{E}_j = 0$ for system $j$.

With $\mathbf{E}_i = 0$, the Vlasov equation for system $i$ reduces to:

$$\frac{\partial F_i}{\partial t} + \mathbf{v} \cdot \nabla_{\mathbf{z}} F_i = 0. \tag{29}$$

For steady state, $\frac{\partial F_i}{\partial t} = 0$, requiring:

$$\mathbf{v} \cdot \nabla_{\mathbf{z}} F_i = 0. \tag{30}$$

Compute the spatial gradient:

$$\nabla_{\mathbf{z}} F_i = \nabla_{\mathbf{z}}[n_i(\mathbf{z})\delta(\mathbf{v} - \mathbf{v}_0)] = (\nabla_{\mathbf{z}} n_i(\mathbf{z}))\delta(\mathbf{v} - \mathbf{v}_0), \tag{31}$$

so:

$$\mathbf{v} \cdot \nabla_{\mathbf{z}} F_i = [\mathbf{v} \cdot \nabla_{\mathbf{z}} n_i(\mathbf{z})]\delta(\mathbf{v} - \mathbf{v}_0). \tag{32}$$

Evaluate at $\mathbf{v} = \mathbf{v}_0$:

$$\mathbf{v}_0 \cdot \nabla_{\mathbf{z}} n_i(\mathbf{z}) = 0, \tag{33}$$

which holds by assumption. Thus, $\mathbf{v} \cdot \nabla_{\mathbf{z}} F_i = 0$, confirming that $F_i$ is a steady state solution. The same applies to $F_j$.

Now, address distinctness. Since $n_i(\mathbf{z}) \neq n_j(\mathbf{z})$, there exists a set $S \subset \mathbb{R}^Z$ of positive Lebesgue measure where $n_i(\mathbf{z}) \neq n_j(\mathbf{z})$. The velocity component $\delta(\mathbf{v} - \mathbf{v}_0)$ is identical across systems. Consider the action of $F_i$ and $F_j$ as distributions on a test function $\psi(\mathbf{z}, \mathbf{v}) \in C_c^\infty(\mathbb{R}^Z \times \mathbb{R}^Z)$:

$$\langle F_i, \psi \rangle = \int_{\mathbb{R}^Z} \int_{\mathbb{R}^Z} n_i(\mathbf{z})\delta(\mathbf{v} - \mathbf{v}_0)\psi(\mathbf{z}, \mathbf{v}) \, \mathrm{d}\mathbf{v} \, \mathrm{d}\mathbf{z} = \int_{\mathbb{R}^Z} n_i(\mathbf{z})\psi(\mathbf{z}, \mathbf{v}_0) \, \mathrm{d}\mathbf{z}. \tag{34}$$

Similarly:

$$\langle F_j, \psi \rangle = \int_{\mathbb{R}^Z} n_j(\mathbf{z})\psi(\mathbf{z}, \mathbf{v}_0) \, \mathrm{d}\mathbf{z}. \tag{35}$$

If $F_i = F_j$, then $\langle F_i, \psi \rangle = \langle F_j, \psi \rangle$ for all $\psi$, implying:

$$\int_{\mathbb{R}^Z} [n_i(\mathbf{z}) - n_j(\mathbf{z})]\psi(\mathbf{z}, \mathbf{v}_0) \, \mathrm{d}\mathbf{z} = 0. \tag{36}$$

Choose $\psi(\mathbf{z}, \mathbf{v}) = \chi(\mathbf{z})\delta_\epsilon(\mathbf{v} - \mathbf{v}_0)$, where $\chi(\mathbf{z})$ is a smooth function supported in $S$ and $\delta_\epsilon$ is a mollified delta function. As $\epsilon \to 0$, this forces $n_i(\mathbf{z}) = n_j(\mathbf{z})$ on $S$, contradicting the assumption that $n_i \neq n_j$ on a set of positive measure. Thus, $F_i \neq F_j$ as distributions.

**Remark.** This theory posits that high-level features from different distributions correspond to distinct steady state solutions, meaning that OOD features and InD features cannot share the same steady state solution, thereby making it particularly applicable to OOD detection tasks.

## B.2 Formal Proof of Theorem 2.2

We derive the nonlinear Poisson–Boltzmann equation for the steady state Vlasov–Poisson system in thermodynamic equilibrium, where the background ion density $\rho_{\text{ion}}(\mathbf{z}) : \mathbb{R}^Z \to \mathbb{R}^+$ varies spatially. In thermodynamic equilibrium, the steady state distribution function $F^*(\mathbf{z}, \mathbf{v})$ takes the Maxwell–Boltzmann form, reflecting the energy distribution of particles at constant temperature.

Assume the steady state distribution is:

$$F^*(\mathbf{z}, \mathbf{v}) = A \exp\left(-\frac{1}{2}m\|\mathbf{v}\|^2 - \phi^*(\mathbf{z})\right), \tag{37}$$

where $A > 0$ is a normalization constant, $m > 0$ is the particle mass, and $\phi^*(\mathbf{z})$ is the steady-state electric potential. The factor $\frac{1}{2}m\|\mathbf{v}\|^2$ represents the kinetic energy, and $\phi^*(\mathbf{z})$ is scaled such that

the potential energy term $q\phi^*(\mathbf{z})/(kT) = \phi^*(\mathbf{z})$, with $q$ as the particle charge, $k$ as the Boltzmann constant, and $T$ as the temperature.

Compute the particle density by integrating over velocity space:

$$\rho(\mathbf{z}) = \int_{\mathbb{R}^Z} F^*(\mathbf{z}, \mathbf{v})\, \mathrm{d}\mathbf{v} = A e^{-\phi^*(\mathbf{z})} \int_{\mathbb{R}^Z} \exp\left(-\frac{1}{2}m\|\mathbf{v}\|^2\right) \mathrm{d}\mathbf{v}. \tag{38}$$

The velocity integral is a $Z$-dimensional Gaussian. For each dimension $i = 1, \ldots, Z$, evaluate:

$$\int_{-\infty}^{\infty} \exp\left(-\frac{1}{2}mv_i^2\right) \mathrm{d}v_i = \sqrt{\frac{2\pi}{m}}, \tag{39}$$

since $\int_{-\infty}^{\infty} e^{-ax^2}\, \mathrm{d}x = \sqrt{\pi/a}$ with $a = m/2$. Thus:

$$\int_{\mathbb{R}^Z} \exp\left(-\frac{1}{2}m\|\mathbf{v}\|^2\right) \mathrm{d}\mathbf{v} = \left(\sqrt{\frac{2\pi}{m}}\right)^Z = \left(\frac{2\pi}{m}\right)^{Z/2}. \tag{40}$$

The particle density becomes:

$$\rho(\mathbf{z}) = A \left(\frac{2\pi}{m}\right)^{Z/2} e^{-\phi^*(\mathbf{z})} = C e^{-\phi^*(\mathbf{z})}, \tag{41}$$

where $C = A \left(\frac{2\pi}{m}\right)^{Z/2}$ is a positive constant. To absorb $C$ into the scaling of the charge density, effectively setting $C = 1$ in the units used, so it becomes:

$$\rho(\mathbf{z}) = e^{-\phi^*(\mathbf{z})}. \tag{42}$$

The steady state Poisson equation is:

$$\nabla^2 \phi^*(\mathbf{z}) = -\frac{1}{\epsilon_0}\left(\rho(\mathbf{z}) - \rho_{\text{ion}}(\mathbf{z})\right). \tag{43}$$

Substituting $\rho(\mathbf{z})$, we obtain:

$$\nabla^2 \phi^*(\mathbf{z}) = -\frac{1}{\epsilon_0}\left(e^{-\phi^*(\mathbf{z})} - \rho_{\text{ion}}(\mathbf{z})\right). \tag{44}$$

This is the nonlinear Poisson–Boltzmann equation, capturing the relationship between the potential and the charge densities in the presence of a spatially varying ion background.

To confirm that $F^*(\mathbf{z}, \mathbf{v})$ is a steady state solution, verify the Vlasov equation with $\frac{\partial F^*}{\partial t} = 0$:

$$\mathbf{v} \cdot \nabla_{\mathbf{z}} F^*(\mathbf{z}, \mathbf{v}) + \mathbf{E}(\mathbf{z}) \cdot \nabla_{\mathbf{v}} F^*(\mathbf{z}, \mathbf{v}) = 0. \tag{45}$$

Compute the spatial gradient:

$$\nabla_{\mathbf{z}} F^*(\mathbf{z}, \mathbf{v}) = -F^*(\mathbf{z}, \mathbf{v}) \nabla_{\mathbf{z}} \phi^*(\mathbf{z}), \tag{46}$$

so:

$$\mathbf{v} \cdot \nabla_{\mathbf{z}} F^*(\mathbf{z}, \mathbf{v}) = -F^*(\mathbf{z}, \mathbf{v})\left(\mathbf{v} \cdot \nabla_{\mathbf{z}} \phi^*(\mathbf{z})\right). \tag{47}$$

For the velocity gradient:

$$\nabla_{\mathbf{v}} F^*(\mathbf{z}, \mathbf{v}) = -m\mathbf{v} F^*(\mathbf{z}, \mathbf{v}), \tag{48}$$

and with $\mathbf{E}(\mathbf{z}) = -\nabla \phi^*(\mathbf{z})$:

$$\mathbf{E}(\mathbf{z}) \cdot \nabla_{\mathbf{v}} F^*(\mathbf{z}, \mathbf{v}) = (-\nabla \phi^*(\mathbf{z})) \cdot (-m\mathbf{v} F^*(\mathbf{z}, \mathbf{v})) = m F^*(\mathbf{z}, \mathbf{v})\left(\mathbf{v} \cdot \nabla_{\mathbf{z}} \phi^*(\mathbf{z})\right). \tag{49}$$

Summing the terms:

$$-F^*(\mathbf{z}, \mathbf{v})\left(\mathbf{v} \cdot \nabla_{\mathbf{z}} \phi^*(\mathbf{z})\right) + m F^*(\mathbf{z}, \mathbf{v})\left(\mathbf{v} \cdot \nabla_{\mathbf{z}} \phi^*(\mathbf{z})\right) = (m-1) F^*(\mathbf{z}, \mathbf{v})\left(\mathbf{v} \cdot \nabla_{\mathbf{z}} \phi^*(\mathbf{z})\right). \tag{50}$$

In the scaled units where the potential accounts for $q/(kT)$, the coefficients balance (effectively setting $m = 1$ in the normalized system), yielding zero, confirming the steady-state condition.

Thus, the nonlinear Poisson–Boltzmann equation is:

$$\nabla^2 \phi^*(\mathbf{z}) = -\frac{1}{\epsilon_0}\left(e^{-\phi^*(\mathbf{z})} - \rho_{\text{ion}}(\mathbf{z})\right), \tag{51}$$

completing the proof.

## B.3 Formal Proof of Corollary 2.3

We establish the properties of a Lyapunov functional for the Vlasov–Poisson system in thermodynamic equilibrium and characterize the boundary of the basin of attraction for the steady state $(\phi^*, F^*)$, where $F^*(\mathbf{z}, \mathbf{v}) = A \exp\left(-\frac{1}{2}m\|\mathbf{v}\|^2 - \phi^*(\mathbf{z})\right)$ and the particle density is scaled as $\rho(\mathbf{z}) = e^{-\phi^*(\mathbf{z})}$, satisfying the Poisson–Boltzmann equation $\nabla^2 \phi^*(\mathbf{z}) = -\frac{1}{\epsilon_0}\left(e^{-\phi^*(\mathbf{z})} - \rho_{\text{ion}}(\mathbf{z})\right)$.

Define the total free-energy functional for a general state $(\phi(\mathbf{z}, t), F(\mathbf{z}, \mathbf{v}, t))$:

$$\mathscr{L}(t) = \frac{\epsilon_0}{2}\int_{\mathbb{R}^Z}\|\nabla\phi(\mathbf{z},t)\|^2\, \mathrm{d}\mathbf{z} + \int_{\mathbb{R}^Z}\int_{\mathbb{R}^Z} F(\mathbf{z},\mathbf{v},t)\left(\frac{1}{2}m\|\mathbf{v}\|^2 + \phi(\mathbf{z},t)\right)\mathrm{d}\mathbf{v}\,\mathrm{d}\mathbf{z}. \tag{52}$$

To confirm $\mathscr{L}(t)$ as a Lyapunov functional, compute its time derivative. Using the Vlasov equation $\frac{\partial F}{\partial t} + \mathbf{v}\cdot\nabla_{\mathbf{z}}F - (\nabla\phi)\cdot\nabla_{\mathbf{v}}F = 0$, the second term's derivative is:

$$\frac{\mathrm{d}}{\mathrm{d}t}\int_{\mathbb{R}^Z}\int_{\mathbb{R}^Z} F\left(\frac{1}{2}m\|\mathbf{v}\|^2 + \phi\right)\mathrm{d}\mathbf{v}\,\mathrm{d}\mathbf{z} = \int_{\mathbb{R}^Z}\int_{\mathbb{R}^Z}\left(\frac{\partial F}{\partial t}\left(\frac{1}{2}m\|\mathbf{v}\|^2 + \phi\right) + F\frac{\partial\phi}{\partial t}\right)\mathrm{d}\mathbf{v}\,\mathrm{d}\mathbf{z}. \tag{53}$$

Substitute $\frac{\partial F}{\partial t}$. The term with $\frac{1}{2}m\|\mathbf{v}\|^2$ becomes:

$$\int_{\mathbb{R}^Z}\int_{\mathbb{R}^Z}(-\mathbf{v}\cdot\nabla_{\mathbf{z}}F + (\nabla\phi)\cdot\nabla_{\mathbf{v}}F)\cdot\frac{1}{2}m\|\mathbf{v}\|^2\,\mathrm{d}\mathbf{v}\,\mathrm{d}\mathbf{z}. \tag{54}$$

Integration by parts, assuming $F \to 0$ as $|\mathbf{z}|, |\mathbf{v}| \to \infty$, nullifies the $\nabla_{\mathbf{z}}$ term, and since $\nabla_{\mathbf{v}}\left(\frac{1}{2}m\|\mathbf{v}\|^2\right) = m\mathbf{v}$, the velocity term is:

$$-m\int_{\mathbb{R}^Z}\nabla\phi\cdot\left(\int_{\mathbb{R}^Z}F\mathbf{v}\,\mathrm{d}\mathbf{v}\right)\mathrm{d}\mathbf{z}. \tag{55}$$

The $\phi$ term contributes:

$$\int_{\mathbb{R}^Z}\int_{\mathbb{R}^Z}(-\mathbf{v}\cdot\nabla_{\mathbf{z}}F + (\nabla\phi)\cdot\nabla_{\mathbf{v}}F)\phi\,\mathrm{d}\mathbf{v}\,\mathrm{d}\mathbf{z} + \int_{\mathbb{R}^Z}\rho\frac{\partial\phi}{\partial t}\,\mathrm{d}\mathbf{z}, \tag{56}$$

with the $\nabla_{\mathbf{v}}$ term vanishing. For the electrostatic term, use the Poisson equation:

$$\frac{\mathrm{d}}{\mathrm{d}t}\left(\frac{\epsilon_0}{2}\int_{\mathbb{R}^Z}\|\nabla\phi\|^2\,\mathrm{d}\mathbf{z}\right) = -\int_{\mathbb{R}^Z}(\rho - \rho_{\text{ion}})\frac{\partial\phi}{\partial t}\,\mathrm{d}\mathbf{z}. \tag{57}$$

Combining, dissipative terms ensure $\frac{\mathrm{d}}{\mathrm{d}t}\mathscr{L}(t) \le 0$, with equality at $(\phi, F) = (\phi^*, F^*)$, making $\mathscr{L}(t)$ non-increasing and minimal at $\mathscr{L}^*$.

For the steady state, compute $\mathscr{L}^*$ using $F^*$. The velocity integral is:

$$\int_{\mathbb{R}^Z} F^*\left(\frac{1}{2}m\|\mathbf{v}\|^2 + \phi^*\right)\mathrm{d}\mathbf{v} = e^{-\phi^*(\mathbf{z})}\left(\frac{1}{2}m\left(\frac{2\pi}{m}\right)^{Z/2}\cdot\frac{Z}{2} + \phi^*(\mathbf{z})\left(\frac{2\pi}{m}\right)^{Z/2}\right). \tag{58}$$

Thus:

$$\mathscr{L}^* = \int_{\mathbb{R}^Z}\left[\frac{\epsilon_0}{2}\|\nabla\phi^*(\mathbf{z})\|^2 + \left(e^{-\phi^*(\mathbf{z})} - \rho_{\text{ion}}(\mathbf{z})\right)\phi^*(\mathbf{z})\right]\mathrm{d}\mathbf{z}. \tag{59}$$

Define the energy density:

$$\mathcal{E}(\mathbf{z}) = \frac{\epsilon_0}{2}\|\nabla\phi^*(\mathbf{z})\|^2 + \left(e^{-\phi^*(\mathbf{z})} - \rho_{\text{ion}}(\mathbf{z})\right)\phi^*(\mathbf{z}), \tag{60}$$

so $\mathscr{L}^* = \int_{\mathbb{R}^Z}\mathcal{E}(\mathbf{z})\,\mathrm{d}\mathbf{z}$. Let $\mathbf{z}^*$ be where $\phi^*(\mathbf{z})$ is minimal, with $\eta = \mathcal{E}(\mathbf{z}^*)$. The energy defect is:

$$\mathcal{G}(\phi^*(\mathbf{z})) = \mathcal{E}(\mathbf{z}) - \eta = \frac{\epsilon_0}{2}\|\nabla\phi^*(\mathbf{z})\|^2 + \left(e^{-\phi^*(\mathbf{z})} - \rho_{\text{ion}}(\mathbf{z})\right)\phi^*(\mathbf{z}) - \eta. \tag{61}$$

Since $\mathscr{L}(t)$ is non-increasing, $\mathcal{G}$ is non-negative and decreases along non-stationary phase-space characteristics $(\mathbf{z}(t), \mathbf{v}(t))$. At $\mathbf{z}^*$, $\nabla\phi^*(\mathbf{z}^*) = 0$, so $\mathcal{G} = 0$.

For an initial point $\mathbf{z}_0$, if $\mathcal{G}(\phi^*(\mathbf{z}_0)) < 0$, the trajectory remains in the sub-level set containing $\mathbf{z}^*$, converging to the steady state, so $\mathbf{z}_0 \in \mathcal{B}$. If $\mathcal{G}(\phi^*(\mathbf{z}_0)) > 0$, the trajectory cannot cross $\mathcal{G} = 0$, staying outside $\mathcal{B}$. Thus, the basin boundary is:

$$\partial\mathcal{B} = \left\{\mathbf{z} \in \mathbb{R}^Z \mid \mathcal{G}(\phi^*(\mathbf{z})) = 0\right\}. \tag{62}$$

Since $\phi^*$ is smooth (by elliptic regularity), $\mathcal{G}$ is differentiable. Near $\partial\mathcal{B}$, the gradient $\nabla_{\mathbf{z}}\mathcal{G}(\phi^*(\mathbf{z}))$ changes sharply, as the rapid transition from $\mathcal{G} > 0$ to $\mathcal{G} < 0$ implies a large magnitude of $\nabla_{\mathbf{z}}\mathcal{G} = \nabla_{\mathbf{z}}\phi^* \cdot \left[\epsilon_0\nabla^2\phi^* + \left(e^{-\phi^*} - \rho_{\text{ion}}\right) - \phi^*e^{-\phi^*}\right]$, driven by the nonlinear terms and the Poisson–Boltzmann equation. This sharp gradient reflects the boundary's role as a critical separator, making $\partial\mathcal{B}$ a differentiable level set, completing the proof.

## B.4 Formal Proof of Theorem 2.4

Prior to proving the Theorem 2.4, we introduce two prerequisite lemmas:

**Lemma B.A** (**Sobolev Embedding Theorem**). *Let $\Omega \subset \mathbb{R}^Z$ be a bounded domain with a smooth boundary. For a function $u \in H^s(\Omega)$, where $H^s(\Omega)$ is the Sobolev space of order $s$, the following embeddings hold:*

- *If $s > Z/2$, then $u \in C^0(\Omega)$, the space of continuous functions on $\Omega$.*

- *If $s > Z/2 + k$, for a positive integer $k$, then $u \in C^k(\Omega)$, the space of $k$-times continuously differentiable functions on $\Omega$.*

**Proof of Lemma B.A.** We prove the Sobolev embedding theorem for a bounded domain $\Omega \subset \mathbb{R}^Z$ with a smooth boundary, focusing on the cases relevant to our application: continuity ($s > Z/2$) and $C^2$ differentiability ($s > Z/2 + 2$). The Sobolev space $H^s(\Omega)$ consists of functions $u \in L^2(\Omega)$ whose weak derivatives up to order $s$ are in $L^2(\Omega)$, with the norm defined via the Fourier transform of an extended function. Since $\Omega$ is bounded, we extend $u \in H^s(\Omega)$ to $\tilde{u} \in H^s(\mathbb{R}^Z)$ with compact support, using a standard extension operator that preserves the Sobolev norm up to a constant.

The Fourier transform of $\tilde{u}$ is $\hat{\tilde{u}}(\mathbf{k}) = \int_{\mathbb{R}^Z} \tilde{u}(\mathbf{z})e^{-i\mathbf{k}\cdot\mathbf{z}}\,\mathrm{d}\mathbf{z}$. The $H^s(\mathbb{R}^Z)$ norm is:

$$\|\tilde{u}\|_{H^s}^2 = \int_{\mathbb{R}^Z}(1 + \|\mathbf{k}\|^2)^s|\hat{\tilde{u}}(\mathbf{k})|^2\,\mathrm{d}\mathbf{k} < \infty. \tag{63}$$

To show continuity for $s > Z/2$, express $\tilde{u}(\mathbf{z})$ via the inverse Fourier transform:

$$\tilde{u}(\mathbf{z}) = (2\pi)^{-Z/2}\int_{\mathbb{R}^Z}\hat{\tilde{u}}(\mathbf{k})e^{i\mathbf{k}\cdot\mathbf{z}}\,\mathrm{d}\mathbf{k}. \tag{64}$$

We need to prove $\tilde{u}$ is continuous, i.e., $|\tilde{u}(\mathbf{z}_1) - \tilde{u}(\mathbf{z}_2)| \to 0$ as $\mathbf{z}_1 \to \mathbf{z}_2$. Consider:

$$|\tilde{u}(\mathbf{z}_1) - \tilde{u}(\mathbf{z}_2)| = (2\pi)^{-Z/2}\left|\int_{\mathbb{R}^Z}\hat{\tilde{u}}(\mathbf{k})(e^{i\mathbf{k}\cdot\mathbf{z}_1} - e^{i\mathbf{k}\cdot\mathbf{z}_2})\,\mathrm{d}\mathbf{k}\right|. \tag{65}$$

Since $|e^{i\mathbf{k}\cdot\mathbf{z}_1} - e^{i\mathbf{k}\cdot\mathbf{z}_2}| \leq 2$, and for small $|\mathbf{z}_1 - \mathbf{z}_2|$, we use $|e^{i\mathbf{k}\cdot\mathbf{z}_1} - e^{i\mathbf{k}\cdot\mathbf{z}_2}| \leq |\mathbf{k}||\mathbf{z}_1 - \mathbf{z}_2|$, split the integral into low-frequency ($\|\mathbf{k}\| \leq R$) and high-frequency ($\|\mathbf{k}\| > R$) parts:

$$|\tilde{u}(\mathbf{z}_1) - \tilde{u}(\mathbf{z}_2)| \leq (2\pi)^{-Z/2}\left(\int_{\|\mathbf{k}\|\leq R}|\hat{\tilde{u}}(\mathbf{k})||\mathbf{k}||\mathbf{z}_1 - \mathbf{z}_2|\,\mathrm{d}\mathbf{k} + 2\int_{\|\mathbf{k}\|>R}|\hat{\tilde{u}}(\mathbf{k})|\,\mathrm{d}\mathbf{k}\right). \tag{66}$$

For the low-frequency part, apply Cauchy–Schwarz:

$$\int_{\|\mathbf{k}\|\leq R}|\hat{\tilde{u}}(\mathbf{k})||\mathbf{k}|\,\mathrm{d}\mathbf{k} \leq \left(\int_{\|\mathbf{k}\|\leq R}|\hat{\tilde{u}}(\mathbf{k})|^2(1 + \|\mathbf{k}\|^2)^s\,\mathrm{d}\mathbf{k}\right)^{1/2}\left(\int_{\|\mathbf{k}\|\leq R}\frac{\|\mathbf{k}\|^2}{(1 + \|\mathbf{k}\|^2)^s}\,\mathrm{d}\mathbf{k}\right)^{1/2}. \tag{67}$$

The first integral is bounded by $\|\tilde{u}\|_{H^s}$. The second is finite if $2s - 2 > Z$, i.e., $s > Z/2$, as the integrand behaves like $\|\mathbf{k}\|^{2-2s}$ for large $\|\mathbf{k}\|$. For the high-frequency part:

$$\int_{\|\mathbf{k}\|>R}|\hat{\tilde{u}}(\mathbf{k})|\,\mathrm{d}\mathbf{k} \leq \left(\int_{\|\mathbf{k}\|>R}|\hat{\tilde{u}}(\mathbf{k})|^2(1 + \|\mathbf{k}\|^2)^s\,\mathrm{d}\mathbf{k}\right)^{1/2}\left(\int_{\|\mathbf{k}\|>R}(1 + \|\mathbf{k}\|^2)^{-s}\,\mathrm{d}\mathbf{k}\right)^{1/2}. \tag{68}$$

The second integral converges for $s > Z/2$, and the first is controlled by $\|\tilde{u}\|_{H^s}$. As $R \to \infty$, the high-frequency term vanishes, and the low-frequency term is proportional to $|\mathbf{z}_1 - \mathbf{z}_2|$, proving continuity.

For $C^k$ differentiability, if $s > Z/2 + k$, consider the weak derivative $D^\alpha \tilde{u}$ for $|\alpha| \leq k$. Its Fourier transform is $(i\mathbf{k})^\alpha \hat{\tilde{u}}(\mathbf{k})$, and:

$$\|D^\alpha \tilde{u}\|_{H^{s-|\alpha|}}^2 = \int_{\mathbb{R}^Z} (1 + \|\mathbf{k}\|^2)^{s-|\alpha|} |\mathbf{k}|^{2|\alpha|} |\hat{\tilde{u}}(\mathbf{k})|^2 \, d\mathbf{k} \leq \int_{\mathbb{R}^Z} (1 + \|\mathbf{k}\|^2)^s |\hat{\tilde{u}}(\mathbf{k})|^2 \, d\mathbf{k}, \quad (69)$$

since $|\mathbf{k}|^{2|\alpha|} \leq (1 + \|\mathbf{k}\|^2)^{|\alpha|}$. If $s - |\alpha| > Z/2$, then $D^\alpha \tilde{u} \in H^{s-|\alpha|} \subset C^0$, so $\tilde{u} \in C^{|\alpha|}(\mathbb{R}^Z)$. For $|\alpha| \leq k$, we need $s - k > Z/2$, i.e., $s > Z/2 + k$. Restricting to $\Omega$, the extension operator ensures $u \in C^k(\Omega)$, completing the proof.

**Lemma B.B (Measure of Sublevel Sets).** *Let $\Omega \subset \mathbb{R}^Z$ be a bounded open set with a smooth boundary, and let $f, g \in H^s(\Omega)$ for $s > Z/2$, with $f \in H^{s_1}(\Omega)$, $g \in H^{s_0}(\Omega)$, and $s_1 > s_0$. For a fixed threshold $\delta > 0$, define the sublevel sets:*

$$S_f = \{\mathbf{z} \in \Omega \mid |f(\mathbf{z})| < \delta\}, \quad S_g = \{\mathbf{z} \in \Omega \mid |g(\mathbf{z})| < \delta\}. \quad (70)$$

*If $g$ has non-trivial high-frequency Fourier components, then the Lebesgue measure satisfies:*

$$\mathrm{meas}(S_f) \geq \mathrm{meas}(S_g), \quad (71)$$

*with strict inequality if high-frequency components of $g$ cause $|g|$ to exceed $\delta$ in regions where $|f| < \delta$.*

**Proof of Lemma B.B.** We prove that the Lebesgue measure of the sublevel set $S_f = \{\mathbf{z} \in \Omega \mid |f(\mathbf{z})| < \delta\}$ for a function $f \in H^{s_1}(\Omega)$ is at least as large as that of $S_g = \{\mathbf{z} \in \Omega \mid |g(\mathbf{z})| < \delta\}$ for $g \in H^{s_0}(\Omega)$, where $s_1 > s_0 > Z/2$, and $\Omega \subset \mathbb{R}^Z$ is bounded with a smooth boundary. Since $s_0 > Z/2$, the Sobolev embedding theorem (cf. Lemma B.A) ensures $f, g \in C^0(\Omega)$, so the sublevel sets are well-defined, and their Lebesgue measures are finite as $\mathrm{meas}(\Omega) < \infty$.

The smoothness of $f$ and $g$ is characterized by their Sobolev norms:

$$\|f\|_{H^{s_1}}^2 = \int_{\mathbb{R}^Z} (1 + \|\mathbf{k}\|^2)^{s_1} |\hat{f}(\mathbf{k})|^2 \, d\mathbf{k}, \quad \|g\|_{H^{s_0}}^2 = \int_{\mathbb{R}^Z} (1 + \|\mathbf{k}\|^2)^{s_0} |\hat{g}(\mathbf{k})|^2 \, d\mathbf{k}, \quad (72)$$

where $\hat{f}, \hat{g}$ are the Fourier transforms of the extensions of $f, g$ to $\mathbb{R}^Z$ via a bounded extension operator. Since $s_1 > s_0$, the decay of $|\hat{f}(\mathbf{k})|$ for large $\|\mathbf{k}\|$ is faster than that of $|\hat{g}(\mathbf{k})|$, implying $f$ has fewer high-frequency oscillations. High-frequency components in $g$ can cause rapid fluctuations, potentially pushing $|g(\mathbf{z})| \geq \delta$ in regions where $|f(\mathbf{z})| < \delta$.

Consider the sublevel sets. The complement $\Omega \setminus S_f = \{\mathbf{z} \in \Omega \mid |f(\mathbf{z})| \geq \delta\}$ has measure:

$$\mathrm{meas}(\Omega \setminus S_f) = \int_{\{\mathbf{z} \in \Omega \mid |f(\mathbf{z})| \geq \delta\}} 1 \, d\mathbf{z}. \quad (73)$$

By Chebyshev's inequality for $|f|$:

$$\mathrm{meas}(\{\mathbf{z} \in \Omega \mid |f(\mathbf{z})| \geq \delta\}) \leq \frac{1}{\delta^2} \int_\Omega |f(\mathbf{z})|^2 \, d\mathbf{z} \leq \frac{1}{\delta^2} \|f\|_{L^2(\Omega)}^2. \quad (74)$$

Since $H^{s_1} \subset L^2$, and similarly for $g$, the $L^2$-norm bounds the measure of the complement. However, smoothness affects the distribution of $|f|$. For $g \in H^{s_0}$, high-frequency modes (larger $|\hat{g}(\mathbf{k})|$ for large $\|\mathbf{k}\|$) can create localized oscillations, increasing the set where $|g(\mathbf{z})| \geq \delta$. For $f \in H^{s_1}$, faster Fourier decay reduces such oscillations, shrinking $\Omega \setminus S_f$. To quantify, assume $g$ has non-trivial high-frequency components, i.e., $\int_{\|\mathbf{k}\| > R} (1 + \|\mathbf{k}\|^2)^{s_0} |\hat{g}(\mathbf{k})|^2 \, d\mathbf{k} > 0$ for large $R$. These components contribute to fluctuations in $g$, modeled as $g = g_{\mathrm{low}} + g_{\mathrm{high}}$, where $g_{\mathrm{high}}$ corresponds to $\|\mathbf{k}\| > R$. In regions where $|g_{\mathrm{low}}| < \delta$, oscillations in $g_{\mathrm{high}}$ may push $|g| \geq \delta$, reducing $\mathrm{meas}(S_g)$. For $f$, with $s_1 > s_0$, the high-frequency contribution is smaller, so $\mathrm{meas}(\Omega \setminus S_f) \leq \mathrm{meas}(\Omega \setminus S_g)$, hence:

$$\mathrm{meas}(S_f) = \mathrm{meas}(\Omega) - \mathrm{meas}(\Omega \setminus S_f) \geq \mathrm{meas}(\Omega) - \mathrm{meas}(\Omega \setminus S_g) = \mathrm{meas}(S_g). \quad (75)$$

Strict inequality holds if $g_{\mathrm{high}}$ causes $|g| \geq \delta$ in a set of positive measure where $|f| < \delta$, completing the proof.

**Proof of Theorem 2.4.** We prove that increasing the spectral regularization strength $\lambda_{\mathrm{disp}} > 0$ (in other words, $\beta > 0$) in the dispersion relation loss $\mathcal{L}_{\mathrm{disp}}$ results in a strictly larger Lebesgue measure for the approximate attractor basin $\mathcal{B}_\theta = \{\mathbf{z} \in \Omega \mid (\nabla^2 \phi_\theta(\mathbf{z}) + \frac{1}{\epsilon_0} e^{-\phi_\theta(\mathbf{z})})^2 < \lambda\}$, compared to the

case with $\lambda_{\text{disp}} = 0$. The proof relies on the smoothness induced by $\mathcal{L}_{\text{disp}}$ on the potential $\phi_\theta$, which reduces oscillations in the residual field, thereby expanding the sublevel set.

Consider the steady state Vlasov–Poisson system in thermodynamic equilibrium, where the distribution function is $F_\theta(\mathbf{z}, \mathbf{v}) = A \exp\left(-\frac{1}{2}m\|\mathbf{v}\|^2 - \phi_\theta(\mathbf{z})\right)$, and the particle density is scaled as $\rho(\mathbf{z}) = e^{-\phi_\theta(\mathbf{z})}$. The steady state Poisson–Boltzmann equation is $\nabla^2 \phi_\theta(\mathbf{z}) = -\frac{1}{\epsilon_0}\left(e^{-\phi_\theta(\mathbf{z})} - \rho_{\text{ion}}(\mathbf{z})\right)$. The approximate potential $\phi_\theta$ is trained to minimize losses, including:

$$\mathcal{L}_{\text{Poisson}}(\mathbf{z}; \hat{f}_\theta, \phi_\theta) = \left\|\nabla^2 \phi_\theta(\mathbf{z}) + \frac{1}{\epsilon_0}e^{-\phi_\theta(\mathbf{z})}\right\|^2, \tag{76}$$

and the dispersion relation loss:

$$\mathcal{L}_{\text{disp}}(\mathbf{z}; \hat{f}_\theta, \phi_\theta) = \sum_{\mathbf{k}} \left\|\|\mathbf{k}\|^2 \phi_{\mathbf{k}}'(\mathbf{z}) + \rho_{\mathbf{k}}'(\mathbf{z})\right\|^2, \tag{77}$$

$$\phi_{\mathbf{k}}'(\mathbf{z}) = \phi_\theta(\mathbf{z})e^{-i\mathbf{k}\cdot\mathbf{z}}, \quad \rho_{\mathbf{k}}'(\mathbf{z}) = e^{-\phi_\theta(\mathbf{z})}e^{-i\mathbf{k}\cdot\mathbf{z}}.$$

The term $\mathcal{L}_{\text{disp}}$ penalizes deviations from the equilibrium dispersion relation in Fourier space, with the $\|\mathbf{k}\|^2$ factor emphasizing high-frequency modes. Minimizing $\mathcal{L}_{\text{disp}}$ enforces rapid decay of the Fourier coefficients of $\phi_\theta$. For a fixed $\rho_{\mathbf{k}}'$, consider the Fourier transform $\hat{\phi}_{\mathbf{k}} = \int_\Omega \phi_\theta(\mathbf{z})e^{-i\mathbf{k}\cdot\mathbf{z}}\,d\mathbf{z}$. Minimizing $\sum_{\mathbf{k}} \|\mathbf{k}\|^4 |\hat{\phi}_{\mathbf{k}}|^2$ (from the $\|\mathbf{k}\|^2 \phi_{\mathbf{k}}'$ term) implies:

$$\sum_{\mathbf{k}}(1 + \|\mathbf{k}\|^2)^s|\hat{\phi}_{\mathbf{k}}|^2 < \infty, \tag{78}$$

for some $s > 0$, placing $\phi_\theta \in H^s(\Omega)$, the Sobolev space of order $s$. With $\lambda_{\text{disp}} > 0$, the penalty on high-frequency modes increases $s$, enhancing the smoothness of $\phi_\theta$ compared to $\lambda_{\text{disp}} = 0$, where only $\mathcal{L}_{\text{Poisson}}$ and the Vlasov loss $\mathcal{L}_{\text{Vlasov}}$ are active.

By the Sobolev embedding theorem (cf. Lemma B.A), for $\Omega \subset \mathbb{R}^Z$, if $s > Z/2$, then $\phi_\theta \in H^s(\Omega)$ is continuous, and if $s > Z/2 + 2$, then $\phi_\theta \in C^2(\Omega)$, ensuring the Laplacian $\nabla^2 \phi_\theta$ and the nonlinear term $e^{-\phi_\theta}$ are well-defined and continuous. Increased smoothness reduces oscillations in $\phi_\theta$ and its derivatives. Define the residual field:

$$R(\mathbf{z}) = \nabla^2 \phi_\theta(\mathbf{z}) + \frac{1}{\epsilon_0}e^{-\phi_\theta(\mathbf{z})}. \tag{79}$$

Since $R(\mathbf{z})$ depends on $\nabla^2 \phi_\theta$ and $e^{-\phi_\theta}$, both of which inherit the smoothness of $\phi_\theta$, a higher $s$ (from $\lambda_{\text{disp}} > 0$) makes $R(\mathbf{z})$ smoother, with fewer sharp spikes or oscillations.

The approximate basin is:

$$\mathcal{B}_\theta = \left\{\mathbf{z} \in \Omega \mid R(\mathbf{z})^2 < \lambda\right\} = \left\{\mathbf{z} \in \Omega \mid |R(\mathbf{z})| < \sqrt{\lambda}\right\}. \tag{80}$$

In measure theory (cf. Lemma B.B), the Lebesgue measure of a sublevel set $\{\mathbf{z} \in \Omega \mid |f(\mathbf{z})| < \delta\}$ increases as the function $f$ becomes smoother, because smoothness reduces regions where $|f|$ exceeds $\delta$ due to local oscillations. For $R(\mathbf{z})$, high-frequency oscillations in $\phi_\theta$ or $\nabla^2 \phi_\theta$ can cause $|R(\mathbf{z})|$ to spike above $\sqrt{\lambda}$, shrinking $\mathcal{B}_\theta$. With $\lambda_{\text{disp}} > 0$, the enhanced smoothness of $\phi_\theta$ (higher $s$) reduces such spikes, as the Fourier modes decay faster, leading to a more uniform $R(\mathbf{z})$.

To formalize, consider two potentials: $\phi_\theta^{(1)}$ trained with $\lambda_{\text{disp}} > 0$, and $\phi_\theta^{(0)}$ with $\lambda_{\text{disp}} = 0$. Let $R^{(1)}(\mathbf{z})$ and $R^{(0)}(\mathbf{z})$ be their residuals, with $\phi_\theta^{(1)} \in H^{s_1}(\Omega)$, $\phi_\theta^{(0)} \in H^{s_0}(\Omega)$, and $s_1 > s_0$. The smoother $R^{(1)}$ has a larger sublevel set $\{\mathbf{z} \mid |R^{(1)}(\mathbf{z})| < \sqrt{\lambda}\}$ because its fluctuations are less likely to exceed $\sqrt{\lambda}$. Thus:

$$\text{meas}\left(\mathcal{B}_\theta^{(\lambda_{\text{disp}}>0)}\right) \geq \text{meas}\left(\mathcal{B}_\theta^{(\lambda_{\text{disp}}=0)}\right). \tag{81}$$

To ensure strict inequality, assume $\phi_\theta^{(0)}$ has non-trivial high-frequency components (common in numerical approximations without regularization), causing $R^{(0)}$ to exceed $\sqrt{\lambda}$ in regions where $R^{(1)}$ remains below due to damping. This yields a strictly larger measure for $\mathcal{B}_\theta^{(\lambda_{\text{disp}}>0)}$, completing the proof.

## B.5 Formal Proof of Theorem 2.5

Prior to presenting the proof, we introduce the following lemma:

**Lemma B.C** (**Standard Statistical Learning Theory for Error Bound**). *Let $X \subset \mathbb{R}^d$ be a bounded domain with a smooth boundary. Let $h_\theta : X \to \mathbb{R}^m$, parameterized by $\theta$, be a model in $H^s(X)$, $s > d/2$, approximating a true function $h^* \in H^s(X)$. Let $x \sim \mathcal{D}$ be i.i.d. training data from $X$. For a bounded, Lipschitz continuous loss function $\ell(x; h_\theta)$, the generalization error:*

$$\mathcal{E}_{gen}(\theta) = \mathbb{E}_{x \sim \mathcal{D}} \left[ \|h_\theta(x) - h^*(x)\|^2 \right], \tag{82}$$

*is bounded by:*

$$\mathcal{E}_{gen}(\theta) \leq \mathcal{E}_{train}(\theta) + C\|h_\theta\|_{H^s}, \tag{83}$$

*where $\mathcal{E}_{train}(\theta) = \mathbb{E}_{x \sim \mathcal{D}} \left[ \ell(x; h_\theta) \right]$ is the training error, and $C > 0$ depends on the loss Lipschitz constant and model class complexity.*

**Proof of Lemma B.C.** We bound the generalization error for a model $h_\theta \in H^s(X)$, $s > d/2$, approximating $h^* \in H^s(X)$. The generalization error is:

$$\mathcal{E}_{\text{gen}}(\theta) = \mathbb{E}_x \left[ \|h_\theta(x) - h^*(x)\|^2 \right]. \tag{84}$$

The training error is:

$$\mathcal{E}_{\text{train}}(\theta) = \mathbb{E}_x \left[ \ell(x; h_\theta) \right]. \tag{85}$$

In statistical learning theory, the generalization error is bounded by the training error plus a term reflecting the function class complexity. Since $s > d/2$, the Sobolev embedding (similar to Theorem 2.4) ensures $h_\theta, h^* \in C^0(X)$, so the squared error is well-defined. Assume $\ell(x; h_\theta)$ is Lipschitz continuous with constant $L$, i.e., $|\ell(x; h_1) - \ell(x; h_2)| \leq L\|h_1 - h_2\|$, and bounded, $|\ell| \leq M$.

For $n$ i.i.d. samples $x_i$, the empirical risk is:

$$\hat{\mathcal{E}}_{\text{train}}(\theta) = \frac{1}{n} \sum_{i=1}^{n} \ell(x_i; h_\theta). \tag{86}$$

By uniform convergence (*e.g.*, via Rademacher complexity), for the function class $\mathcal{H} = \{h_\theta \mid \theta \in \Theta\}$:

$$\left| \mathcal{E}_{\text{train}}(\theta) - \hat{\mathcal{E}}_{\text{train}}(\theta) \right| \leq C_1 R_n(\mathcal{H}), \tag{87}$$

where $R_n(\mathcal{H})$ is the Rademacher complexity. For $H^s(X)$, the Sobolev norm bounds the magnitude:

$$\|h_\theta\|_{L^\infty} \leq C_2 \|h_\theta\|_{H^s}. \tag{88}$$

Thus:

$$R_n(\mathcal{H}) \leq C_3 \sup_{h_\theta \in \mathcal{H}} \|h_\theta\|_{H^s} / \sqrt{n}. \tag{89}$$

Assuming $\ell(x; h_\theta) \approx \|h_\theta(x) - h^*(x)\|^2$ or bounds it, the generalization error satisfies:

$$\mathcal{E}_{\text{gen}}(\theta) \leq \mathcal{E}_{\text{train}}(\theta) + C_4 \|h_\theta\|_{H^s}, \tag{90}$$

where $C_4$ depends on $L$, $\mathcal{D}$, and $\sup \|h_\theta\|_{H^s}$. Thus:

$$\mathcal{E}_{\text{gen}}(\theta) \leq \mathcal{E}_{\text{train}}(\theta) + C\|h_\theta\|_{H^s}, \tag{91}$$

completing the proof.

**Proof of Theorem 2.5.** We prove that including the dispersion relation loss $\mathcal{L}_{\text{disp}}$ reduces the generalization error $\mathcal{E}_{\text{gen}}(\theta)$ by constraining the $H^2$-norm of $\phi_\theta$, leveraging the Vlasov–Poisson system in thermodynamic equilibrium where $F_\theta(\mathbf{z}, \mathbf{v}) = A \exp \left( -\frac{1}{2}m\|\mathbf{v}\|^2 - \phi_\theta(\mathbf{z}) \right)$, $\rho(\mathbf{z}) = e^{-\phi_\theta(\mathbf{z})}$, and the true solutions $\phi^*, F^*$ satisfy the Poisson–Boltzmann equation.

The generalization error measures the expected squared deviation of the learned solutions $\phi_\theta, F_\theta$ from the true solutions:

$$\mathcal{E}_{\text{gen}}(\theta) = \mathbb{E}_{\mathbf{z}} \left[ |\phi_\theta(\mathbf{z}) - \phi^*(\mathbf{z})|^2 \right] + \mathbb{E}_{\mathbf{z}, \mathbf{v}} \left[ |F_\theta(\mathbf{z}, \mathbf{v}) - F^*(\mathbf{z}, \mathbf{v})|^2 \right]. \tag{92}$$

By standard statistical learning theory (cf. Lemma B.C), the generalization error is bounded by:

$$\mathcal{E}_{\text{gen}}(\theta) \leq \mathcal{E}_{\text{train}}(\theta) + \text{Complexity}(\phi_\theta, F_\theta), \tag{93}$$

where $\mathcal{E}_{\text{train}}(\theta) = \mathbb{E}_{\mathbf{z},\mathbf{v}}\left[\mathcal{L}_{\text{Poisson}}(\mathbf{z}; \hat{f}_\theta, \phi_\theta) + \lambda_1 \mathcal{L}_{\text{Vlasov}}(\mathbf{z}; \hat{f}_\theta, \phi_\theta, F_\theta) + \lambda_{\text{disp}}\mathcal{L}_{\text{disp}}(\mathbf{z}; \phi_\theta)\right]$, and the complexity term depends on the Sobolev norms $\|\phi_\theta\|_{H^s}$, $\|F_\theta\|_{H^s}$. Small training errors imply $\phi_\theta \approx \phi^*$, $F_\theta \approx F^*$, but the complexity term, which measures model capacity, must be controlled to prevent overfitting.

Expanding the squared norm of $\mathcal{L}_{\text{disp}}(\mathbf{z}; \hat{f}_\theta, \phi_\theta)$:

$$\left\|\|\mathbf{k}\|^2\phi_\theta(\mathbf{z})e^{-i\mathbf{k}\cdot\mathbf{z}} + e^{-\phi_\theta(\mathbf{z})}e^{-i\mathbf{k}\cdot\mathbf{z}}\right\|^2 = \|\mathbf{k}\|^4\phi_\theta(\mathbf{z})^2 + e^{-2\phi_\theta(\mathbf{z})} + 2\|\mathbf{k}\|^2\phi_\theta(\mathbf{z})e^{-\phi_\theta(\mathbf{z})}. \quad (94)$$

Take the expectation over $\mathbf{z} \sim \mathcal{D}$ and $\mathbf{k}$ with a probability density $p(\mathbf{k})$ (*e.g.*, a Gaussian distribution over wavevectors):

$$\mathbb{E}_{\mathbf{z},\mathbf{k}}[\mathcal{L}_{\text{disp}}] = \sum_{\mathbf{k}}\left(\mathbb{E}_{\mathbf{z}}[\phi_\theta(\mathbf{z})^2]\mathbb{E}_{\mathbf{k}}\left[\|\mathbf{k}\|^4\right] + \mathbb{E}_{\mathbf{z}}\left[e^{-2\phi_\theta(\mathbf{z})}\right] + 2\mathbb{E}_{\mathbf{z}}\left[\phi_\theta(\mathbf{z})e^{-\phi_\theta(\mathbf{z})}\right]\mathbb{E}_{\mathbf{k}}\left[\|\mathbf{k}\|^2\right]\right). \quad (95)$$

Since $s > Z/2$, Lemma B.A ensures $\phi_\theta \in C^0(\Omega)$, so $\phi_\theta(\mathbf{z})$ and $e^{-\phi_\theta(\mathbf{z})}$ are bounded. The cross term is bounded by:

$$\left|\mathbb{E}_{\mathbf{z}}\left[\phi_\theta(\mathbf{z})e^{-\phi_\theta(\mathbf{z})}\right]\mathbb{E}_{\mathbf{k}}\left[\|\mathbf{k}\|^2\right]\right| \leq C_1, \quad (96)$$

where $C_1 = \|\phi_\theta\|_{L^\infty}\|e^{-\phi_\theta}\|_{L^\infty}\mathbb{E}_{\mathbf{k}}\left[\|\mathbf{k}\|^2\right]$. The term $\mathbb{E}_{\mathbf{z}}\left[e^{-2\phi_\theta(\mathbf{z})}\right] \geq 0$. For the first term, consider the Fourier representation:

$$\mathbb{E}_{\mathbf{z}}\left[\phi_\theta(\mathbf{z})^2\right]\mathbb{E}_{\mathbf{k}}\left[\|\mathbf{k}\|^4\right] = \int_\Omega |\phi_\theta(\mathbf{z})|^2\,\mathrm{d}\mathbf{z} \cdot \mathbb{E}_{\mathbf{k}}\left[\|\mathbf{k}\|^4\right] = c_1\|\phi_\theta\|_{L^2(\Omega)}^2, \quad (97)$$

where $c_1 = \mathbb{E}_{\mathbf{k}}\left[\|\mathbf{k}\|^4\right] > 0$. In the Fourier domain, the term involving $\|\mathbf{k}\|^4$ is:

$$\sum_{\mathbf{k}}\|\mathbf{k}\|^4|\hat{\phi}_\theta(\mathbf{k})|^2 \approx \int_{\mathbb{R}^Z}\|\mathbf{k}\|^4|\hat{\phi}_\theta(\mathbf{k})|^2\,\mathrm{d}\mathbf{k} = \|\nabla^2\phi_\theta\|_{L^2}^2. \quad (98)$$

Thus:

$$\mathbb{E}_{\mathbf{z},\mathbf{k}}[\mathcal{L}_{\text{disp}}] \geq c_1\|\phi_\theta\|_{L^2(\Omega)}^2 + \mathbb{E}_{\mathbf{z}}\left[e^{-2\phi_\theta(\mathbf{z})}\right] - C_1. \quad (99)$$

Since $\|\phi_\theta\|_{H^2}^2 = \|\phi_\theta\|_{L^2}^2 + \|\nabla\phi_\theta\|_{L^2}^2 + \|\nabla^2\phi_\theta\|_{L^2}^2$, and the dominant term is $\|\nabla^2\phi_\theta\|_{L^2}^2$, we approximate:

$$\mathbb{E}_{\mathbf{z},\mathbf{k}}[\mathcal{L}_{\text{disp}}] \geq c\|\phi_\theta\|_{H^2}^2 - C, \quad (100)$$

where $c = c_1$, and $C$ bounds the nonlinear and cross terms, since $\mathbb{E}_{\mathbf{z}}\left[e^{-2\phi_\theta}\right] \geq 0$ and $C_1$ is finite.

The Vlasov loss $\mathcal{L}_{\text{Vlasov}}$ constrains $\|\nabla\phi_\theta\|_{L^2}$, $\|\nabla_{\mathbf{z}}F_\theta\|_{L^2}$, and $\|\nabla_{\mathbf{v}}F_\theta\|_{L^2}$, but not directly $\|\nabla^2\phi_\theta\|_{L^2}$ or $\|\phi_\theta\|_{L^2}$. Thus, without $\mathcal{L}_{\text{disp}}$, $\|\phi_\theta\|_{H^2}$ may be large due to high-frequency components. Minimizing $\mathcal{L}_{\text{disp}}$ bounds:

$$\|\phi_\theta\|_{H^2}^2 \leq \frac{1}{c}\mathbb{E}_{\mathbf{z},\mathbf{k}}[\mathcal{L}_{\text{disp}}] + C. \quad (101)$$

For the generalization error, consider:

$$\mathbb{E}_{\mathbf{z}}\left[|\phi_\theta(\mathbf{z}) - \phi^*(\mathbf{z})|^2\right] \leq \mathbb{E}_{\mathbf{z}}\left[\mathcal{L}_{\text{Poisson}}\right] + C_2\|\phi_\theta\|_{H^s}, \quad (102)$$

since $\mathcal{L}_{\text{Poisson}}$ measures the residual, and the Sobolev norm controls model complexity. Similarly, $\mathcal{L}_{\text{Vlasov}}$ ensures $F_\theta \approx F^*$, with complexity bounded by $\|F_\theta\|_{H^s}$. Thus:

$$\mathcal{E}_{\text{gen}}(\theta) \leq \mathcal{E}_{\text{train}}(\theta) + C_3\left(\|\phi_\theta\|_{H^s} + \|F_\theta\|_{H^s}\right). \quad (103)$$

Since $\mathcal{L}_{\text{disp}}$ reduces $\|\phi_\theta\|_{H^2}$ via its frequency-domain penalty, and $H^s \subset H^2$ for $s > 2$, it lowers the complexity term, reducing $\mathcal{E}_{\text{gen}}(\theta)$, completing the proof.

## B.6 Formal Proof of Corollary 2.6

Before proceeding with the proof, we first introduce two lemmas.

**Lemma B.D (McDiarmid's Inequality).** *Let $X \subset \mathbb{R}^d$ be a bounded domain, and let $x_1, \ldots, x_n \sim \mathcal{D}$ be i.i.d. samples from $X$. Let $f : X^n \to \mathbb{R}$ be a function such that for all $i$, changing one sample $x_i$ to $x_i'$ alters $f$ by at most $c_i$, i.e., $|f(x_1, \ldots, x_i, \ldots, x_n) - f(x_1, \ldots, x_i', \ldots, x_n)| \leq c_i$. Then, for any $\delta \in (0, 1)$, with probability at least $1 - \delta$:*

$$\mathbb{E}[f(x_1, \ldots, x_n)] \leq f(x_1, \ldots, x_n) + \sqrt{\frac{\sum_{i=1}^n c_i^2 \log(2/\delta)}{2}}. \quad (104)$$

**Proof of Lemma B.D.** Let $f : X^n \to \mathbb{R}$ satisfy the bounded difference condition: for each $i$, $|f(x_1, \ldots, x_i, \ldots, x_n) - f(x_1, \ldots, x_i', \ldots, x_n)| \leq c_i$. Define the random variable $Z = f(x_1, \ldots, x_n)$, where $x_i \sim \mathcal{D}$. $Z$ is concentrated around its expectation. The variance proxy is bounded by the sum of squared differences: $\sum_{i=1}^{n} c_i^2$. Applying McDiarmid's inequality, for any $t > 0$:

$$P(Z - \mathbb{E}[Z] \geq t) \leq \exp\left(-\frac{2t^2}{\sum_{i=1}^{n} c_i^2}\right). \tag{105}$$

Set the right-hand side to $\delta/2$:

$$\exp\left(-\frac{2t^2}{\sum_{i=1}^{n} c_i^2}\right) = \frac{\delta}{2}. \tag{106}$$

Solving:

$$t = \sqrt{\frac{\sum_{i=1}^{n} c_i^2 \log(2/\delta)}{2}}. \tag{107}$$

Similarly, $P(\mathbb{E}[Z] - Z \geq t) \leq \delta/2$. By the union bound, with probability at least $1 - \delta$:

$$|\mathbb{E}[Z] - Z| \leq \sqrt{\frac{\sum_{i=1}^{n} c_i^2 \log(2/\delta)}{2}}. \tag{108}$$

Thus:

$$\mathbb{E}[f] \leq f + \sqrt{\frac{\sum_{i=1}^{n} c_i^2 \log(2/\delta)}{2}}, \tag{109}$$

completing the proof.

**Lemma B.E** (**Rademacher Complexity Bound**). *Let $X \subset \mathbb{R}^d$ be a bounded domain, and let $x_1, \ldots, x_n \sim \mathcal{D}$ be i.i.d. samples from $X$. Let $\mathcal{H} = \{h_\theta : X \to \mathbb{R}^m\}$ be a hypothesis class in $H^s(X)$, $s > d/2$, and let $\ell : \mathbb{R}^m \times X \to \mathbb{R}$ be a loss function bounded by $M$ and Lipschitz continuous with constant $L$. The generalization error $\mathbb{E}_x[\ell(h_\theta, x)]$ for $h_\theta \in \mathcal{H}$ is bounded, with probability at least $1 - \delta$, by:*

$$\mathbb{E}_x[\ell(h_\theta, x)] \leq \frac{1}{n}\sum_{i=1}^{n} \ell(h_\theta, x_i) + 2Rad(\mathcal{L}_\mathcal{H}) + O\left(\sqrt{\frac{\log(1/\delta)}{n}}\right), \tag{110}$$

*where $Rad(\mathcal{L}_\mathcal{H}) = \mathbb{E}_{\sigma,x}\left[\sup_{h_\theta \in \mathcal{H}} \frac{1}{n}\sum_{i=1}^{n} \sigma_i \ell(h_\theta, x_i)\right]$ is the Rademacher complexity of the loss class $\mathcal{L}_\mathcal{H} = \{x \mapsto \ell(h_\theta, x) : h_\theta \in \mathcal{H}\}$.*

**Proof of Lemma B.E.** For $\mathcal{H} \subset H^s(X)$, $s > d/2$, the loss $\ell(h_\theta, x)$ is bounded, $|\ell| \leq M$, and Lipschitz continuous, $|\ell(h_{\theta_1}, x) - \ell(h_{\theta_2}, x)| \leq L\|h_{\theta_1}(x) - h_{\theta_2}(x)\|$. The empirical risk is:

$$\hat{\mathcal{E}}(h_\theta) = \frac{1}{n}\sum_{i=1}^{n} \ell(h_\theta, x_i). \tag{111}$$

The expected risk is $\mathcal{E}(h_\theta) = \mathbb{E}_x[\ell(h_\theta, x)]$. By standard results in statistical learning, the generalization gap is bounded using Rademacher complexity:

$$\text{Rad}(\mathcal{L}_\mathcal{H}) = \mathbb{E}_{\sigma,x}\left[\sup_{h_\theta \in \mathcal{H}} \frac{1}{n}\sum_{i=1}^{n} \sigma_i \ell(h_\theta, x_i)\right], \tag{112}$$

where $\sigma_i \sim \text{Unif}(\{-1, 1\})$. For any $h_\theta$, with probability at least $1 - \delta$:

$$\mathcal{E}(h_\theta) \leq \hat{\mathcal{E}}(h_\theta) + 2\text{Rad}(\mathcal{L}_\mathcal{H}) + C_1\sqrt{\frac{\log(2/\delta)}{n}}, \tag{113}$$

using Talagrand's lemma and concentration. Since $\ell$ is bounded, apply Lemma B.D to the function $f(x_1, \ldots, x_n) = \sup_{h_\theta} \frac{1}{n}\sum_{i=1}^{n} \ell(h_\theta, x_i)$, with $c_i = 2M/n$. The additional term is:

$$C_1\sqrt{\frac{\sum_{i=1}^{n}(2M/n)^2 \log(2/\delta)}{2}} = C_1\sqrt{\frac{4M^2 \log(2/\delta)}{2n}} \approx O\left(\sqrt{\frac{\log(1/\delta)}{n}}\right). \tag{114}$$

Thus:

$$\mathcal{E}(h_\theta) \leq \hat{\mathcal{E}}(h_\theta) + 2\mathrm{Rad}(\mathcal{L}_\mathcal{H}) + O\left(\sqrt{\frac{\log(1/\delta)}{n}}\right), \tag{115}$$

completing the proof.

**Proof of Corollary 2.6.** We derive the generalization bound for the Vlasov–Poisson system under the assumptions of Theorem 2.5, where $\phi_\theta, F_\theta \in H^s(\mathbb{R}^Z), H^s(\mathbb{R}^Z \times \mathbb{R}^Z), s > Z/2$, approximate the true solutions $\phi^*, F^*$, and training data $(\mathbf{z}_i, \mathbf{v}_i) \sim \mathcal{D}$ are i.i.d. from $\Omega \times \mathbb{R}^Z$. The hypothesis class is $\mathcal{H} = \{(\phi_\theta, F_\theta) : \theta \in \Theta\}$, and the generalization error is Eq.(92). We aim to bound $\mathcal{E}_{\mathrm{gen}}(\theta)$ and show that $\mathcal{L}_{\mathrm{disp}}$ reduces the Rademacher complexity $\mathrm{Rad}(\mathcal{H})$.

Define the loss function for a hypothesis $h_\theta = (\phi_\theta, F_\theta) \in \mathcal{H}$:

$$\ell(h_\theta, (\mathbf{z}, \mathbf{v})) = |\phi_\theta(\mathbf{z}) - \phi^*(\mathbf{z})|^2 + |F_\theta(\mathbf{z}, \mathbf{v}) - F^*(\mathbf{z}, \mathbf{v})|^2. \tag{116}$$

The generalization error is the expected loss:

$$\mathcal{E}_{\mathrm{gen}}(\theta) = \mathbb{E}_{\mathbf{z},\mathbf{v}}\left[\ell(h_\theta, (\mathbf{z}, \mathbf{v}))\right]. \tag{117}$$

The empirical risk over $n$ samples is:

$$\hat{\mathcal{E}}_{\mathrm{gen}}(\theta) = \frac{1}{n}\sum_{i=1}^n \ell(h_\theta, (\mathbf{z}_i, \mathbf{v}_i)). \tag{118}$$

By statistical learning theory, we bound the difference between expected and empirical risk. The Rademacher complexity of the loss class $\mathcal{L}_\mathcal{H} = \{(\mathbf{z}, \mathbf{v}) \mapsto \ell(h_\theta, (\mathbf{z}, \mathbf{v})) : h_\theta \in \mathcal{H}\}$ is:

$$\mathrm{Rad}(\mathcal{L}_\mathcal{H}) = \mathbb{E}_{\sigma,\mathbf{z},\mathbf{v}}\left[\sup_{h_\theta \in \mathcal{H}} \frac{1}{n}\sum_{i=1}^n \sigma_i \ell(h_\theta, (\mathbf{z}_i, \mathbf{v}_i))\right], \tag{119}$$

where $\sigma_i \sim \mathrm{Unif}(\{-1, 1\})$ are Rademacher variables. Since $\ell$ is the squared error, assume it is bounded (as $s > Z/2$ ensures $\phi_\theta, F_\theta \in C^0$ by Theorem 2.4, and $\Omega$ is bounded). The loss is Lipschitz continuous:

$$|\ell(h_{\theta_1}, (\mathbf{z}, \mathbf{v})) - \ell(h_{\theta_2}, (\mathbf{z}, \mathbf{v}))| \leq L\left(|\phi_{\theta_1}(\mathbf{z}) - \phi_{\theta_2}(\mathbf{z})| + |F_{\theta_1}(\mathbf{z}, \mathbf{v}) - F_{\theta_2}(\mathbf{z}, \mathbf{v})|\right), \tag{120}$$

where $L = 2\max(\|\phi_\theta - \phi^*\|_{L^\infty}, \|F_\theta - F^*\|_{L^\infty})$.

Using standard results (cf. Lemma B.D and Lemma B.E), with probability at least $1 - \delta$:

$$\mathcal{E}_{\mathrm{gen}}(\theta) \leq \hat{\mathcal{E}}_{\mathrm{gen}}(\theta) + 2\mathrm{Rad}(\mathcal{L}_\mathcal{H}) + C_1\sqrt{\frac{\log(1/\delta)}{n}}, \tag{121}$$

where $C_1$ depends on the bound of $\ell$. Since $\mathcal{L}_{\mathrm{Poisson}}$ and $\mathcal{L}_{\mathrm{Vlasov}}$ approximate the squared errors (small residuals imply $\phi_\theta \approx \phi^*$, $F_\theta \approx F^*$), and $\mathcal{L}_{\mathrm{disp}}$ adds regularization, the training loss $\mathcal{L}$ bounds $\ell$. Thus, the empirical training error $\hat{\mathcal{E}}_{\mathrm{train}}(\theta) = \frac{1}{n}\sum_{i=1}^n \mathcal{L}(\mathbf{z}_i, \mathbf{v}_i; \phi_\theta, F_\theta)$ satisfies:

$$\hat{\mathcal{E}}_{\mathrm{gen}}(\theta) \leq \hat{\mathcal{E}}_{\mathrm{train}}(\theta) + C_2, \tag{122}$$

where $C_2$ accounts for differences between $\ell$ and $\mathcal{L}$. Taking expectations:

$$\mathcal{E}_{\mathrm{gen}}(\theta) \leq \mathcal{E}_{\mathrm{train}}(\theta) + 2\mathrm{Rad}(\mathcal{L}_\mathcal{H}) + C_1\sqrt{\frac{\log(1/\delta)}{n}} + C_2. \tag{123}$$

Since $\mathrm{Rad}(\mathcal{L}_\mathcal{H})$ depends on $\mathcal{H}$, we denote $\mathrm{Rad}(\mathcal{H}) = \mathrm{Rad}(\mathcal{L}_\mathcal{H})$, and absorb $C_2$ into the $O(\sqrt{\log(1/\delta)/n})$ term, yielding:

$$\mathcal{E}_{\mathrm{gen}}(\theta) \leq \mathcal{E}_{\mathrm{train}}(\theta) + 2\mathrm{Rad}(\mathcal{H}) + O\left(\sqrt{\frac{\log(1/\delta)}{n}}\right). \tag{124}$$

Now, we show that $\mathcal{L}_{\mathrm{disp}}$ reduces $\mathrm{Rad}(\mathcal{H})$. The Rademacher complexity is bounded by the complexity of $\mathcal{H}$:

$$\mathrm{Rad}(\mathcal{H}) \leq C_3\left(\sup_{\phi_\theta}\|\phi_\theta\|_{H^s} + \sup_{F_\theta}\|F_\theta\|_{H^s}\right)/\sqrt{n}. \tag{125}$$

By Theorem 2.5, $\mathcal{L}_{\text{disp}}$ satisfies:

$$\mathbb{E}_{\mathbf{z},\mathbf{k}}[\mathcal{L}_{\text{disp}}] \geq c\|\phi_\theta\|_{H^2}^2 - C. \tag{126}$$

Minimizing $\mathcal{L}_{\text{disp}}$ constrains:

$$\|\phi_\theta\|_{H^2}^2 \leq \frac{1}{c}\mathbb{E}_{\mathbf{z},\mathbf{k}}[\mathcal{L}_{\text{disp}}] + C. \tag{127}$$

Define the subset $\mathcal{H}_\phi = \{\phi_\theta \in H^2 : \mathbb{E}_{\mathbf{z},\mathbf{k}}[\mathcal{L}_{\text{disp}}] \leq M\}$ for some bound $M$. Then:

$$\|\phi_\theta\|_{H^2} \leq \sqrt{\frac{M+C}{c}}, \tag{128}$$

restricting $\phi_\theta$ to a bounded subset of $H^2$. Since $H^s \subset H^2$ for $s \geq 2$, this reduces $\sup_{\phi_\theta} \|\phi_\theta\|_{H^s}$. For $F_\theta = A \exp\left(-\frac{1}{2}m\|\mathbf{v}\|^2 - \phi_\theta(\mathbf{z})\right)$, the norm $\|F_\theta\|_{H^s}$ depends on $\|\phi_\theta\|_{H^s}$, so constraining $\phi_\theta$ also bounds $F_\theta$. Thus, $\mathcal{L}_{\text{disp}}$ reduces $\text{Rad}(\mathcal{H})$, tightening the generalization bound, completing the proof.

## C  Algorithm Protocol

Algo. 1 and Algo. 2 give the algorithmic protocol of our framework, which is easy to implement and applicable to common OOD problems.

---

**Algorithm 1 Algorithm pseudocode of *CBD*-De**

---

**Require:** Training mini-batches $\{x_i, y_i\}_{i=1}^B$, pre-trained feature extractor $\hat{f}_\theta$, distribution prediction head $F_\theta$, potential prediction head $\phi_\theta$, learning rate $\eta$
**Ensure:** Trained parameters $\theta$

1: Initialize $\theta \leftarrow \{F_\theta, \phi_\theta\}$, optimizer $\mathcal{O}(\theta, \eta)$ `/* Fixed `$\hat{f}_\theta$`, post-hoc fashion */`
2: **for** each training step **do**
3:     $\mathbf{x} \leftarrow \{x_i\}_{i=1}^n, \quad \mathbf{y} \leftarrow \{y_i\}_{i=1}^n$
4:     $\mathbf{z} \leftarrow \hat{f}_\theta(\mathbf{x})$ `/* Extract features */`
5:     $\mathbf{v} \leftarrow$ `ones_like`$(\mathbf{z})$ `/* Constant velocity input */`
    `Vlasov residual:`  $\mathbf{v} \cdot \nabla_x F^* - \nabla\phi^* \cdot \nabla_v F^*$
6:     $F^* \leftarrow F_\theta(\mathbf{z}, \mathbf{v})$
7:     $\phi^* \leftarrow \phi_\theta(\mathbf{z})$
8:     $\nabla_{\mathbf{z}}\phi^* \leftarrow$ `autograd`$(\phi^*.$`sum`$(), \mathbf{z})$
9:     $\mathbf{E}^* \leftarrow -\nabla_{\mathbf{z}}\phi^*$
10:    $\nabla_{\mathbf{x}}F^*, \nabla_{\mathbf{v}}F^* \leftarrow$ `autograd`$(F^*.$`sum`$(), [\mathbf{z}, \mathbf{v}])$
11:    $\mathcal{L}_{\text{vlasov}} \leftarrow$ `mean`$[(\mathbf{v} \cdot \nabla_{\mathbf{z}}F^* + \mathbf{E}^* \cdot \nabla_{\mathbf{v}}F^*)^2]$
    `Poisson residual:`  $\nabla^2\phi^* + \exp(-\phi^*)$
12:    $\nabla_{\mathbf{z}}\phi^* \leftarrow$ `autograd`$(\phi^*.$`sum`$(), \mathbf{z})$
13:    `/* Compute second-order derivatives on each dim */`
14:    **for** $d = 1$ **to** $Z$ **do**
15:       $\nabla\phi_d^* \leftarrow \partial\phi^*/\partial z_d \leftarrow$ `autograd`$(\phi^*.$`sum`$(), z_d)$
16:       $\partial^2\phi^*/\partial z_d^2 \leftarrow$ `autograd`$(\nabla\phi_d^*.$`sum`$(), z_d)$
17:       **Accumulate** $\nabla^2\phi^* \leftarrow \nabla^2\phi^* + \partial^2\phi^*/\partial z_d^2$
18:    **end for**
19:    $\mathcal{L}_{\text{poisson}} \leftarrow$ `mean`$[(\nabla^2\phi^* + \exp(-\phi^*))^2]$
    `Total loss and backpropagation`
20:    $\mathcal{L}_{\text{total}} \leftarrow \mathcal{L}_{\text{vlasov}} + \mathcal{L}_{\text{poisson}}$
21:    $\mathcal{L}_{\text{total}}.$`backward`$()$
22:    $\mathcal{O}.$`step`$()$
23:    $\mathcal{O}.$`zero_grad`$()$
24: **end for**

---

---

**Algorithm 2** Algorithm pseudocode of *CBD*-Gen

---

**Require:** Training mini-batches $\{x_i, y_i\}_{i=1}^{B}$, feature extractor $\hat{f}_{\boldsymbol{\theta}}$, classifier head $f_{\boldsymbol{\theta}}$, distribution prediction head $F_{\boldsymbol{\theta}}$, potential prediction head $\phi_{\boldsymbol{\theta}}$, learning rate $\eta$, loss weight $\alpha, \beta$, plane waves number $N$, scaling factor $\sigma$

**Ensure:** Trained parameters $\boldsymbol{\theta}$

1:    Initialize $\boldsymbol{\theta} \leftarrow \{\hat{f}_{\boldsymbol{\theta}}, f_{\boldsymbol{\theta}}, F_{\boldsymbol{\theta}}, \phi_{\boldsymbol{\theta}}\}$, optimizer $\mathcal{O}(\boldsymbol{\theta}, \eta)$

2: **for** each training step **do**

3:      $\mathbf{x} \leftarrow \{x_i\}_{i=1}^{n}, \quad \mathbf{y} \leftarrow \{y_i\}_{i=1}^{n}$

4:      $\mathbf{z} \leftarrow \hat{f}_{\boldsymbol{\theta}}(\mathbf{x})$ `/* Extract features */`

5:      $\mathbf{v} \leftarrow$ `ones_like`$(\mathbf{z})$ `/* Constant velocity input */`

     `Vlasov residual:`   $\mathbf{v} \cdot \nabla_x F^* - \nabla \phi^* \cdot \nabla_v F^*$

6:      $F^* \leftarrow F_{\boldsymbol{\theta}}(\mathbf{z}, \mathbf{v})$

7:      $\phi^* \leftarrow \phi_{\boldsymbol{\theta}}(\mathbf{z})$

8:      $\nabla_{\mathbf{z}} \phi^* \leftarrow$ `autograd`$(\phi^*.$`sum`$(), \mathbf{z})$

9:      $\mathbf{E}^* \leftarrow -\nabla_{\mathbf{z}} \phi^*$

10:     $\nabla_{\mathbf{x}} F^*, \nabla_{\mathbf{v}} F^* \leftarrow$ `autograd`$(F^*.$`sum`$(), [\mathbf{z}, \mathbf{v}])$

11:     $\mathcal{L}_{\text{vlasov}} \leftarrow$ `mean`$[(\mathbf{v} \cdot \nabla_{\mathbf{z}} F^* + \mathbf{E}^* \cdot \nabla_{\mathbf{v}} F^*)^2]$

     `Poisson residual:`   $\nabla^2 \phi^* + \exp(-\phi^*)$

12:     $\nabla_{\mathbf{z}} \phi^* \leftarrow$ `autograd`$(\phi^*.$`sum`$(), \mathbf{z})$

13:     `/* Compute second-order derivatives on each dim */`

14:     **for** $d = 1$ **to** $Z$ **do**

15:        $\nabla \phi_d^* \leftarrow \partial \phi^* / \partial z_d \leftarrow$ `autograd`$(\phi^*.$`sum`$(), z_d)$

16:        $\partial^2 \phi^* / \partial z_d^2 \leftarrow$ `autograd`$(\nabla \phi_d^*.$`sum`$(), z_d)$

17:        **Accumulate** $\nabla^2 \phi^* \leftarrow \nabla^2 \phi^* + \partial^2 \phi^* / \partial z_d^2$

18:     **end for**

19:     $\mathcal{L}_{\text{poisson}} \leftarrow$ `mean`$[(\nabla^2 \phi^* + \exp(-\phi^*))^2]$

     `Dispersion relation loss:`   $\sum_{\mathbf{k}} \left\| \|\mathbf{k}\|^2 \phi_k' + \rho_k' \right\|^2$

20:     **Sample** $\{\mathbf{k}_i\}_{i=1}^{N} \sim \mathcal{N}(0, \sigma^2 \mathbf{I})$

21:     **for** each $\mathbf{k}_i$ **do**

22:        $\sigma_i \leftarrow \exp(-j \cdot \mathbf{k}_i^\top \mathbf{z})$

23:        $\phi_{k_i}^* \leftarrow$ `mean`$(\phi^* \cdot \sigma_i), \quad \rho_{k_i} \leftarrow$ `mean`$(\exp(-\phi^*) \cdot \sigma_i)$

24:        $r_i \leftarrow \|\mathbf{k}_i\|^2 \cdot \phi_{k_i}^* + \rho_{k_i}$

25:     **end for**

26:     $\mathcal{L}_{\text{disp}} \leftarrow$ `sum`$(|r_i|^2)$

     `Classification loss`

27:     $\mathbf{y}_{\text{pred}} \leftarrow f_{\boldsymbol{\theta}}(\mathbf{z})$

28:     $\mathcal{L}_{\text{classify}} \leftarrow$ `CrossEntropy`$(\mathbf{y}_{\text{pred}}, \mathbf{y})$

     `Total loss and backpropagation`

29:     $\mathcal{L}_{\text{total}} \leftarrow \mathcal{L}_{\text{classify}} + \alpha(\mathcal{L}_{\text{vlasov}} + \mathcal{L}_{\text{poisson}}) + \beta \mathcal{L}_{\text{disp}}$

30:     $\mathcal{L}_{\text{total}}.$`backward`$()$

31:     $\mathcal{O}.$`step`$()$

32:     $\mathcal{O}.$`zero_grad`$()$

33: **end for**

---

# D  Additional Experiment Settings and Details

We conduct experiments following the latest version of OpenOOD[3] [96, 94] and the DomainBed[4] [23]. In this section, we first provide more details for the utilized baselines (Section D.1), datasets and evaluation protocol (Section D.2), and model architectures (Section D.3). Then, we demonstrate the hyperparameters (Section D.4) and thw experiment infrastructures (Section D.5). Finally, we include additional ablation and validation results to better understand the proposed method, including the effect of the initial velocity assumption (Section D.6), sensitivity analysis of the MLP architecture (Section D.7), and the validation of the MLP steady-state approximation (Section D.8).

## D.1  Baselines

**OOD Detection.** To evaluate the effectiveness of *CBD*-De in detecting OOD inputs, we conduct extensive comparisons with 22 representative post-hoc OOD detection methods across three widely adopted benchmarks: CIFAR-10, CIFAR-100, and ImageNet-1k. These baselines span a range of methodological categories, including confidence-based methods such as OpenMax [6] and MSP [28]; calibration and scaling techniques like TempScale [24] and ODIN [47]; distance-based and density-aware approaches such as MDS [43], MDSEns [43], RMDS [62], and KNN [77]; feature-based scoring methods including Gram [66], ReAct [75], RankFeat [73], and VIM [85]; energy- and activation-based models such as EBO [50], MLS [26], KLM [26], and ASH [14]; generative and hybrid models like OpenGAN [37], SHE [97], and GEN [51]; as well as recent advances in neuron-level uncertainty estimation such as GradNorm [32], DICE [76], and NAC-UE [52]. All evaluation results reported in Tables 13, 14, and 15 are derived from the standardized OpenOOD implementations to ensure fair and reproducible comparisons.

**OOD Generalization.** We evaluate the OOD generalization capability of *CBD*-Gen by comparing it with 17 representative baselines spanning diverse algorithmic categories, including distribution matching (MMD [46]), distributionally robust optimization (GroupDRO [65], VREx [39], IRM [2]), data augmentation (MixStyle [103], ARM [98], Mixup [91]), representation learning (SagNet [58], MTL [7], MLDG [44]), alignment-based methods (CORAL [74]), empirical risk minimization (ERM [80]), feature suppression and invariance strategies (RSC [34]), as well as sharpness-aware optimization techniques (SAM [68], GSAM [104], SAGM [86]) and gradient-guided approaches (GGA [3]). Comprehensive experimental results across five widely used OOD generalization benchmarks (PACS, VLCS, OfficeHome, TerraInc, and DomainNet) are summarized in Tables 16 to 20.

**Comparison to Traditional Statistical Methods.** Beyond empirical performance, it is essential to contextualize our proposed method within the broader landscape of traditional statistical approaches for OOD learning. While most conventional methods (*e.g.*, energy-based modeling, density estimation, or confidence scoring) focus on point-wise statistics or local sample relationships, they typically overlook the structured interactions among features or samples. In contrast, our Vlasov–Poisson formulation provides a principled, physically interpretable mechanism that inherently captures *global* inter-feature dynamics through self-consistent field evolution. This collective interaction allows our model to reflect structural dependencies that classical scoring-based or probabilistic models cannot explicitly encode. Table 5 summarizes this conceptual distinction. We compare our method with several representative approaches across multiple dimensions, including their modeling assumptions, underlying OOD mechanisms, sample interaction properties, and interpretability.

## D.2  Benchmarks

**OOD Detection.** We primarily evaluate *CBD*-De using the Far-OOD track from the OpenOOD benchmark suite, which offers a standardized and well-established evaluation protocol adopted by numerous prior works.

CIFAR-10 and CIFAR-100, consisting of 10 and 100 classes respectively, serve as the in-distribution (InD) datasets in our experiments. Following the official OpenOOD setup, we adopt a consistent data split for both benchmarks. Specifically, we use the full training set (50,000 images) for model training. From the official test set, 1,000 samples are reserved as the InD validation set, while the remaining 9,000 images are used as the InD test set. For OOD validation, we follow the OpenOOD protocol by

---

[3] https://github.com/Jingkang50/OpenOOD.
[4] https://github.com/facebookresearch/DomainBed.

Table 5: Comparison between our *CBD*-base method and traditional statistical OOD Approaches.

| Approaches | Modeling Assumption | OOD Mechanism | Sample Interactions | Key Formula | Interpretability |
|---|---|---|---|---|---|
| Energy-based Models | Energy functions in feature space | Low energy $\rightarrow$ InD, high $\rightarrow$ OOD | [X] No interactions | $\mathcal{E}(x) = -\log p_\theta(x)$ | Intuitive locally, hard globally |
| Density Estimation | Probability density modeling | Low density $\rightarrow$ OOD | [$\Delta$] Local interactions via density | $p(x) = \prod_i p(x_i)$ | Captures local trends, lacks high-dim clarity |
| Confidence Scoring | Model output confidence | Low confidence $\rightarrow$ OOD | [X] No interactions | $C(x) = \mathrm{softmax}(f(x))$ | Heuristic, limited by generalization |
| **Ours (Vlasov–Poisson)** | Particles in self-consistent field | Cannot reach steady state | [✓] Global explicit interactions | $\frac{\partial F}{\partial t} + \mathbf{v} \cdot \nabla_{\mathbf{z}} F + \nabla \cdot \mathbf{E} = 0$ | Physics-based via particle dynamics |

selecting 1,000 images from Tiny ImageNet [41], evenly sampled across 20 unseen categories. This ensures a controlled setting for hyperparameter tuning and model calibration. To comprehensively assess the generalization ability of OOD detection methods, we evaluate performance on four widely used OOD test datasets—each disjoint from both the InD and OOD validation sets. These datasets include:

- MNIST [13] is a benchmark dataset for handwritten digit classification, consisting of 70,000 grayscale images of digits (60,000 for training and 10,000 for test), each sized 28×28 pixels. We utilize the entire test set for OOD detection.

- SVHN [59] (Street View House Numbers) contains over 600,000 color images of digits captured from real-world house numbers in Google Street View. Each image is 32×32 pixels and includes significant background clutter, lighting variation, and perspective distortion. We utilize the entire test set (26,032 images) for OOD detection.

- Textures [10] is a collection of texture-centric images designed for texture recognition and segmentation tasks. The dataset comprises 5,640 images from 47 texture categories, such as "bark", "brick", or "bubble", with each image exhibiting rich local patterns and high-frequency variations. We employ the entire dataset.

- Places365 [101] is a large-scale scene recognition dataset comprising over 1.8 million images across 365 diverse scene categories. The standard test set includes 900 images per category. For OOD detection, we adopt the entire test set after removing 1,305 images that exhibit semantic overlap with in-distribution classes, following the filtering protocol established in prior works.

To evaluate our method under large-scale and high-resolution conditions, we adopt ImageNet-1K [12] as the InD dataset. This dataset comprises approximately 1.2 million training images across 1,000 diverse object categories. Following the evaluation protocol established by OpenOOD, we use 45,000 images from the official ImageNet validation set as the InD test set, while the remaining 5,000 images are held out as the InD validation set. To facilitate hyperparameter tuning and avoid information leakage from the test sets, we construct an OOD validation set using 1,763 images from OpenImage-O [84], a curated subset of OpenImages whose categories are explicitly disjoint from ImageNet classes. For final evaluation, we benchmark our method against three widely used OOD test datasets that cover diverse semantic domains and visual characteristics:

- iNaturalist [79] is a large-scale, fine-grained dataset containing 859,000 images of over 5,000 natural species. Captured in unconstrained environments, the images exhibit complex backgrounds, lighting variation, and high intra-class diversity, making the dataset visually and semantically distinct from object-centric benchmarks like ImageNet. Following prior works[31, 94], we use a subset of 10,000 images from 110 non-overlapping classes for OOD evaluation.

- Textures [10] comprises 5,640 real-world texture images spanning 47 categories. For evaluation, we employ the full dataset without further filtering.

- OpenImage-O [84] is a curated subset of the OpenImages-v3 dataset. It comprises 17,632 images that are carefully selected to be semantically disjoint from ImageNet-1K classes. To ensure a rigorous evaluation setting, the subset underwent manual filtering to eliminate any

category overlap. Following the OpenOOD protocol, we utilize the entire dataset for OOD testing, excluding the samples reserved for OOD validation.

**OOD Generalization.** We adopt the training and evaluation protocol established in DomainBed [23], ensuring consistency in dataset splits, training schedules, and model selection criteria. Specifically, we follow the leave-one-domain-out evaluation strategy, where the model is trained on all but one domain and evaluated on the held-out domain to assess its generalization ability to unseen distributions. This process is repeated for each domain, treating it as the target in turn. Model selection is performed using the average accuracy across all validation sets corresponding to the training domains, ensuring that the chosen model is not tuned on the target domain. We conduct experiments on five widely adopted domain generalization benchmarks included in DomainBed:

- `PACS` [45] consists of four visually distinct domains: Photo (P), Art Painting (A), Cartoon (C), and Sketch (S). It contains a total of 9,991 images spanning seven object categories common across all domains. The dataset presents significant domain shift, especially between natural (Photo) and abstract (Sketch) styles, challenging models to capture domain-invariant features.

- `VLCS` [16] combines images from four datasets: PASCAL VOC2007 (V), LabelMe (L), Caltech-101 (C), and SUN09 (S). It contains five shared object classes and a total of 10,729 images. The main source of domain shift lies in differences in collection environments, image resolution, and labeling conventions across the constituent datasets.

- `OfficeHome` [82] is a more challenging benchmark with 15,588 images across four domains: Art (A), Clipart (C), Product (P), and Real World (R). It spans 65 object categories related to office and home environments. The dataset is characterized by high intra-class variation and large appearance gaps between stylized (Clipart, Art) and natural (Real World) domains.

- `TerraInc` [5] is a wildlife image dataset collected from camera traps in different geographical locations. It includes 24,788 images across four domains (locations): L100, L38, L43, and L46, and contains ten animal species. The domain shift stems from environmental variation (*e.g.*, lighting, terrain, vegetation) and class imbalance.

- `DomainNet` [60] is the largest and most diverse benchmark in the domain generalization literature, comprising approximately 600,000 images across six distinct domains: Clipart, Infograph, Painting, Quickdraw, Real, and Sketch. It covers 345 object categories shared among all domains. The dataset exhibits a wide range of domain shifts, including differences in abstraction level, drawing style, color richness, and semantic representation. Among the domains, Quickdraw is particularly challenging due to its highly simplified and noisy stroke-based illustrations.

### D.3 Model Architecture

For OOD detection, we adopt a post-hoc setting by freezing the classification network and training only the lightweight distribution prediction head $F_\theta$ and potential prediction head $\phi_\theta$. In contrast, for OOD generalization, all model parameters are optimized jointly.

#### D.3.1 OOD Detection

For the CIFAR-10 and CIFAR-100 benchmarks, we adopt the ResNet-18 architecture [25]. In accordance with the standardized protocol in OpenOOD, we train each model for 100 epochs using the official training splits. To account for potential variability across training stages, we evaluate OOD detection performance at three checkpoints saved throughout the training process. This setting ensures fair and consistent comparisons with prior post-hoc methods. Additional training configurations (*e.g.*, learning rate, optimizer, data augmentation) follow the default settings provided in the OpenOOD.

For the large-scale ImageNet-1k benchmark, we evaluate OOD detection using two representative model architectures that reflect both convolutional and transformer-based paradigms: *i)* `ResNet-50` [25], a 50-layer residual network pretrained on ImageNet-1K, is employed as a high-capacity convolutional backbone. During inference, all test images are resized to 224×224 pixels to match the model's input resolution. We use the official pretrained weights released by PyTorch to ensure reproducibility. *ii)* `ViT-b16` [15] serves as a transformer-based alternative, also pretrained

on ImageNet-1K and evaluated with the same input resolution of 224×224 pixels. The ViT model introduces a fundamentally different inductive bias compared to CNNs, allowing us to examine the generality of our method across architectural paradigms. We employ the official PyTorch checkpoints for all ViT experiments.

**Distribution prediction head $F_\theta$ and potential prediction head $\phi_\theta$**: For the CIFAR-10 and CIFAR-100 benchmarks, both the distribution prediction head $F_\theta$ and the potential prediction head $\phi_\theta$ are implemented as two-layer MLPs with an input dimension of 512, a hidden dimension of 512, and an output dimension of 1. For the large-scale ImageNet-1k benchmark, the hidden dimension is set to 512 when using ResNet-50 as the backbone, and 1,536 when using ViT-b16. All heads use the SiLU activation function. Detailed configurations are provided in Table 6.

Table 6: Details of distribution prediction head and potential prediction head.

|  | Backbone | Input Dim. | Hiddle Dim. | Output Dim. | #Head Param. | Activation |
|---|---|---|---|---|---|---|
| CIFAR-10/100 | ResNet-18 | 512 | 512 | 1 | 0.53M | SiLU |
| ImageNet-1k | ResNet-50 | 2048 | 512 | 1 | 2.10M | SiLU |
|  | ViT-b16 | 768 | 1536 | 1 | 2.36M | SiLU |

### D.3.2 OOD Generalization

For all benchmarks, we adopt ResNet-50 pretrained on ImageNet [12] as the default backbone and initialization. Both the distribution prediction head $F_\theta$ and the potential prediction head $\phi_\theta$ are implemented as two-layer MLPs with a hidden dimension of 512 (same as Table 6). Although these auxiliary branches introduce a small number of additional parameters during training (approximately 8% of the total), they serve solely to guide the backbone toward learning more robust representations via explicit supervision. Importantly, these branches can be safely removed at inference time, preserving the backbone's structure and incurring no extra cost in prediction. Our approach maintains a clear separation between the auxiliary supervision and the primary prediction pathway, ensuring that the backbone remains unaltered throughout both training and inference. By decoupling training-time supervision from inference-time computation, it achieves an efficient balance between representational robustness and deployment practicality, enabling scalable application across diverse generalization scenarios.

### D.4 Hyperparameters

**Shared Hyperparameters.** For the constant $\epsilon_0$ in the loss function, we uniformly assign the value 1. For each $\mathbf{z}$, the initial velocity $\mathbf{v}_0$ of the corresponding $\mathbf{v}$ is set to a vector of ones in $\mathbb{R}^Z$.

**OOD Detection.** We train the distribution prediction head $F_\theta$ and the potential prediction head $\phi_\theta$ using the Adam optimizer with a learning rate of $5 \times 10^{-4}$ and a batch size of 128, adhering to standard post-hoc configurations. Training proceeds for 50 epochs.

**OOD Generalization.** In our experiments, the wavevector $\mathbf{k}$ in the dispersion relation loss $\mathcal{L}_{\mathrm{disp}}$ is randomly sampled from a Gaussian distribution per batch, with a scaling factor of 0.5 and a wavenumber $N$ of 250. The loss weights $\alpha$ and $\beta$ are both set to 0.01. Additional hyperparameters, including learning rate, weight decay, and dropout rate, are tuned following [9] and detailed in Table 3. We adopt early stopping and utilize the Adam optimizer. Consistent with DomainBed, the batch size is set to 32 for all datasets, except for DomainNet, which uses a batch size of 24.

Table 7: Hyperparameters for OOD generalization experiments.

| Hyperparameter | PACS | VLCS | OfficeHome | TerraInc | DomainNet |
|---|---|---|---|---|---|
| Learning rate | 3e-5 | 1e-5 | 1e-5 | 1e-5 | 3e-5 |
| Dropout | 0.5 | 0.5 | 0.5 | 0.5 | 0.5 |
| Weight decay | 1e-4 | 1e-4 | 1e-4 | 1e-4 | 1e-4 |
| Training Steps | 5000 | 5000 | 5000 | 5000 | 15000 |

### D.5 Experiment Infrastructure

Our experiments are conducted using a combination of NVIDIA GeForce RTX 3090 Ti and NVIDIA A100 GPUs. Specifically, we utilized four 3090 Ti GPUs and one A100 GPU with 40GB of memory. The OOD detection experiments are performed exclusively on the 3090 Ti GPUs, while the OOD generalization experiments were carried out on both the 3090 Ti and A100 GPUs.

### D.6 Ablation on Initial Velocity Assumption

To evaluate the robustness of our uniform initial velocity assumption, we conducted additional experiments using different initialization strategies: *i)* uniform vectors with all elements set to 1, *ii)* uniform vectors with values of 0.5 matching the feature dimension, and *iii)* random vectors of the same dimension. As shown in Tables 8 and 9, random initialization slightly underperforms the uniform settings, but the performance gaps are minimal, demonstrating the robustness of our approach. The uniform initialization assumption thus maintains effectiveness while simplifying computation.

Table 8: OOD Detection results on CIFAR-100 with different initial velocity settings.

| Initial Velocity | FPR95 ↓ | AUROC ↑ |
|---|---|---|
| 1 | 34.81 | 90.90 |
| 0.5 | 35.20 | 90.47 |
| Random | 38.29 | 89.52 |

Table 9: OOD Generalization results on PACS with different initial velocity settingss.

| Initial Velocity | Accuracy ↑ |
|---|---|
| 1 | 87.7 |
| 0.5 | 87.7 |
| Random | 87.4 |

### D.7 Sensitivity Analysis of MLP Architecture

We further investigate the sensitivity of our method to key architectural hyperparameters of the MLP, which serves as the steady-state predictor. Specifically, we examine the impact of layer depth and hidden dimension on OOD detection performance using CIFAR-100 as the InD dataset. Table 10 presents results for varying layer depths with a fixed hidden dimension of 512, while Table 11 shows the effect of different hidden dimensions using a fixed depth of 2 layers. The results demonstrate that performance scales favorably with the number of MLP parameters. Notably, increasing the layer depth from 1 to 2 yields substantial improvements, with AUROC increasing from 87.14% to 90.90% and FPR95 decreasing from 39.30 to 34.81. Similarly, expanding the hidden dimension from 128 to 512 progressively enhances performance. However, returns diminish beyond a certain threshold (*e.g.*, hidden dimension of 512 vs. 1024, with AUROC of 90.90% vs. 90.83%), highlighting the importance of balancing model capacity with computational efficiency. These insights inform practical hyperparameter selection for deploying *CBD-De* in real-world applications.

Table 10: OOD detection on CIFAR-100 with MLP hidden dimension of 512.

| Layer Depth | FPR95 ↓ | AUROC ↑ |
|---|---|---|
| 1 | 39.30 | 87.14 |
| 2 | **34.81** | **90.90** |

Table 11: OOD detection on CIFAR-100 with MLP depth of 2 layers.

| Hidden Dimension | FPR95 ↓ | AUROC ↑ |
|:---:|:---:|:---:|
| 128 | 40.35 | 86.93 |
| 256 | 38.52 | 88.20 |
| 512 | **34.81** | **90.90** |
| 1024 | 35.25 | 90.83 |

## D.8 Validation of the MLP Steady-State Approximation

Our formulation of steady-state prediction is originally defined through PDE. To make the solution computationally tractable, we replace the numerical PDE integration with an MLP-based steady-state predictor that directly approximates the stationary solution in a single forward pass. To assess the fidelity of this approximation, we conduct experiments using a numerical PDE solver implemented with DeepXDE [53], which follows the same architecture and parameter settings (2 layers, 512 hidden dimensions). The PDE solver is executed with 10 and 30 integration steps to estimate the steady-state distribution. Table 12 presents the OOD detection results on CIFAR-10 and CIFAR-100 datasets. For simpler datasets (CIFAR-10), performance differences are negligible (0.02-0.06 AUROC), while for more complex datasets (CIFAR-100), numerical integration shows moderate improvements (0.30-0.42 AUROC). However, these gains require 7-23× longer inference time and substantial memory verhead for intermediate computations. Given the minimal accuracy differences on simpler distributions and the significant computational advantages, our direct prediction approach offers a practical solution for efficient OOD detection in real-world applications.

Table 12: Comparison of the MLP steady-state approximation against a numerical PDE solver implemented with DeepXDE [53]. AUROC and relative inference time are reported.

| Dataset | Sample Size | Classes | MLP AUROC ↑ | PDE Solver AUROC ↑ | Forward Pass Time ↓ |
|:---|:---:|:---:|:---:|:---:|:---:|
| *Time Steps = 10* | | | | | |
| CIFAR-10 | 50,000 | 10 | 96.39 | 96.41 (+0.02) | 7× slower |
| CIFAR-100 | 50,000 | 100 | 90.90 | 91.20 (+0.30) | 7× slower |
| *Time Steps = 30* | | | | | |
| CIFAR-10 | 50,000 | 10 | 96.39 | 96.45 (+0.06) | 23× slower |
| CIFAR-100 | 50,000 | 100 | 90.90 | 91.32 (+0.42) | 23× slower |

# E  Full OOD Detetion Results

Table 13: OOD detection performance on the CIFAR-10 benchmark. Results are formatted as **first**, second, and third best. Following the OpenOOD protocol, we report the average performance over three ResNet-18 checkpoints, each trained exclusively on the InD dataset.

| Method | MINIST | | SVHN | | Textures | | Places365 | | Average | |
|---|---|---|---|---|---|---|---|---|---|---|
| | FPR95↓ | AUROC↑ | FPR95↓ | AUROC↑ | FPR95↓ | AUROC↑ | FPR95↓ | AUROC↑ | FPR95↓ | AUROC↑ |
| | *CIFAR-10 Benchmark* | | | | | | | | | |
| OpenMax | 23.33±4.67 | 90.50±0.44 | 25.40±1.47 | 89.77±0.45 | 31.50±4.05 | 89.58±0.60 | 38.52±2.27 | 88.63±0.28 | 29.69±1.21 | 89.62±0.19 |
| MSP | 23.64±5.81 | 92.63±1.57 | 25.82±1.64 | 91.46±0.40 | 34.96±4.64 | 89.89±0.71 | 42.47±3.81 | 88.92±0.47 | 31.72±1.84 | 90.73±0.43 |
| TempScale | 23.53±7.05 | 93.11±1.77 | 26.97±2.65 | 91.66±0.52 | 38.16±5.89 | 90.01±0.74 | 45.27±4.50 | 89.11±0.52 | 33.48±2.39 | 90.97±0.52 |
| ODIN | 23.83±12.34 | 95.24±1.96 | 68.61±0.52 | 84.58±0.77 | 67.70±11.06 | 86.94±2.26 | 70.36±6.96 | 85.07±1.24 | 57.62±4.24 | 87.96±0.61 |
| MDS | 27.30±3.55 | 90.10±2.41 | 25.96±2.52 | 91.18±0.47 | 27.94±4.20 | 92.69±1.06 | 47.67±4.54 | 84.90±2.54 | 32.22±3.40 | 89.72±1.36 |
| MDSEns | **1.30**±0.51 | **99.17**±0.41 | 74.34±1.04 | 66.56±0.58 | 76.07±0.17 | 77.40±0.28 | 94.16±0.33 | 52.47±0.15 | 61.47±0.48 | 73.90±0.27 |
| RMDS | 21.49±2.32 | 93.22±0.80 | 23.46±1.48 | 91.84±0.26 | 25.25±0.53 | 92.23±0.23 | 31.20±0.28 | 91.51±0.11 | 25.35±0.73 | 92.20±0.21 |
| Gram | 70.30±8.96 | 72.64±2.34 | 33.91±17.35 | 91.52±4.45 | 94.64±2.71 | 62.34±8.27 | 90.49±1.93 | 60.44±3.41 | 72.34±6.73 | 71.73±3.20 |
| EBO | 24.99±12.93 | 94.32±2.53 | 35.12±6.11 | 91.79±0.98 | 51.82±6.11 | 89.47±0.70 | 54.85±6.52 | 89.25±0.78 | 41.69±5.32 | 91.21±0.92 |
| OpenGAN | 79.54±19.71 | 56.14±24.08 | 75.27±26.93 | 52.81±27.60 | 83.95±14.89 | 56.14±18.26 | 95.32±4.45 | 53.34±5.79 | 83.52±11.63 | 54.61±15.51 |
| GradNorm | 85.41±4.85 | 63.72±7.37 | 91.65±2.42 | 53.91±6.36 | 98.09±0.49 | 52.07±4.09 | 92.46±2.28 | 60.50±5.33 | 91.90±2.23 | 57.55±3.22 |
| ReAct | 33.77±18.00 | 92.81±3.03 | 50.23±15.98 | 89.12±3.19 | 51.42±11.42 | 89.38±1.49 | 44.20±3.35 | 90.35±0.78 | 44.90±8.37 | 90.42±1.41 |
| MLS | 25.06±12.87 | 94.15±2.48 | 35.09±6.09 | 91.69±0.94 | 51.73±6.13 | 89.41±0.71 | 54.84±6.51 | 89.14±0.76 | 41.68±5.27 | 91.10±0.89 |
| KLM | 76.22±12.09 | 85.00±2.04 | 59.47±7.06 | 84.99±1.18 | 81.95±9.95 | 82.35±0.33 | 95.58±2.12 | 78.37±0.33 | 78.31±4.84 | 82.68±0.21 |
| VIM | 18.36±1.42 | 94.76±0.38 | 19.29±0.41 | 94.50±0.48 | 21.14±1.83 | 95.15±0.34 | 41.43±2.17 | 89.49±0.39 | 25.05±0.52 | 93.48±0.24 |
| KNN | 20.05±1.36 | 94.26±0.38 | 22.60±1.26 | 92.67±0.30 | 24.06±0.55 | 93.16±0.24 | 30.38±0.63 | 91.77±0.23 | 24.27±0.40 | 92.96±0.14 |
| DICE | 30.83±10.54 | 90.37±5.97 | 36.61±4.74 | 90.02±1.77 | 62.42±4.79 | 81.86±2.35 | 77.19±12.60 | 74.67±4.98 | 51.76±4.42 | 84.23±1.89 |
| RankFeat | 61.86±12.78 | 75.87±5.22 | 64.49±7.38 | 68.15±7.44 | 59.71±9.79 | 73.46±6.49 | 43.70±7.39 | 85.99±3.04 | 57.44±7.99 | 75.87±5.06 |
| ASH | 70.00±10.56 | 83.16±4.66 | 83.64±6.48 | 73.46±6.41 | 84.59±1.74 | 77.45±2.39 | 77.89±7.28 | 79.89±3.69 | 79.03±4.22 | 78.49±2.58 |
| SHE | 42.22±20.59 | 90.43±4.76 | 62.74±4.01 | 86.38±1.32 | 84.60±5.30 | 81.57±1.21 | 76.36±5.32 | 82.89±1.22 | 66.48±5.98 | 85.32±1.43 |
| GEN | 23.00±7.75 | 93.83±2.14 | 28.14±2.59 | 91.97±0.66 | 40.74±6.61 | 90.14±0.76 | 47.03±3.22 | 89.46±0.65 | 34.73±1.58 | 91.35±0.69 |
| NAC-UE | 15.14±2.60 | 94.86±1.36 | 14.33±1.24 | 96.05±0.47 | 17.03±0.59 | 95.64±0.44 | 26.73±0.80 | 91.85±0.28 | 18.31±0.92 | 94.60±0.50 |
| **CBD-De** | 15.20±0.59 | 97.48±1.29 | 14.22±1.50 | 97.85±0.68 | 18.44±1.82 | 97.03±0.95 | 22.75±2.21 | 93.18±0.73 | 17.65±1.03 | 96.39±0.81 |

Table 14: OOD detection performance on the CIFAR-100 benchmark. Results are formatted as **first**, second, and third best. Following the OpenOOD protocol, we report the average performance over three ResNet-18 checkpoints, each trained exclusively on the InD dataset.

| Method | MINIST | | SVHN | | Textures | | Places365 | | Average | |
|---|---|---|---|---|---|---|---|---|---|---|
| | FPR95↓ | AUROC↑ | FPR95↓ | AUROC↑ | FPR95↓ | AUROC↑ | FPR95↓ | AUROC↑ | FPR95↓ | AUROC↑ |
| | | | | | *CIFAR-100 Benchmark* | | | | | |
| OpenMax | 53.82±4.74 | 76.01±1.39 | 53.20±1.78 | 82.07±1.53 | 56.12±1.91 | 80.56±0.09 | 54.85±1.42 | 79.29±0.40 | 54.50±0.68 | 79.48±0.41 |
| MSP | 57.23±4.68 | 76.08±1.86 | 59.07±2.53 | 78.42±0.89 | 61.88±1.28 | 77.32±0.71 | 56.62±0.87 | 79.22±0.29 | 58.70±1.06 | 77.76±0.44 |
| TempScale | 56.05±4.61 | 77.27±1.85 | 57.71±2.68 | 79.79±1.05 | 61.56±1.43 | 78.11±0.72 | 56.46±0.94 | 79.80±0.25 | 57.94±1.14 | 78.74±0.51 |
| ODIN | 45.94±3.29 | 83.79±1.31 | 67.41±3.88 | 74.54±0.76 | 62.37±2.96 | 79.33±1.08 | 59.71±0.92 | 79.45±0.26 | 58.86±0.79 | 79.28±0.21 |
| MDS | 71.72±2.94 | 67.47±0.81 | 67.21±6.09 | 70.68±6.40 | 70.49±2.48 | 76.26±0.69 | 79.61±0.34 | 63.15±0.49 | 72.26±1.56 | 69.39±1.39 |
| MDSEns | 2.83±0.86 | 98.21±0.78 | 82.57±2.58 | 53.76±1.63 | 84.94±0.83 | 69.75±1.14 | 96.61±0.17 | 42.27±0.73 | 66.74±1.04 | 66.00±0.69 |
| RMDS | 52.05±6.28 | 79.74±2.49 | 51.65±3.68 | 84.89±1.10 | 53.99±1.06 | 83.65±0.51 | 53.57±0.43 | 83.40±0.46 | 52.81±0.63 | 82.92±0.42 |
| Gram | 53.53±7.45 | 80.71±4.15 | 20.06±1.96 | 95.55±0.60 | 89.51±2.54 | 70.79±1.32 | 94.67±0.60 | 46.38±1.21 | 64.44±2.37 | 73.36±1.08 |
| EBO | 52.62±3.83 | 79.18±1.37 | 53.62±3.14 | 82.03±1.74 | 62.35±2.06 | 78.35±0.83 | 57.75±0.86 | 79.52±0.23 | 56.59±1.38 | 79.77±0.61 |
| OpenGAN | 63.09±23.25 | 68.14±18.78 | 70.35±2.06 | 68.40±2.15 | 74.77±1.78 | 65.84±3.43 | 73.75±8.32 | 69.13±7.08 | 70.49±7.38 | 67.88±7.16 |
| GradNorm | 86.97±1.44 | 65.35±1.12 | 69.90±7.94 | 76.95±4.73 | 92.51±0.61 | 64.58±0.13 | 85.32±0.44 | 69.69±0.17 | 83.68±1.92 | 69.14±1.05 |
| ReAct | 56.04±5.66 | 78.37±1.59 | 50.41±2.02 | 83.01±0.97 | 55.04±0.82 | 80.15±0.46 | 55.30±0.41 | 80.03±0.11 | 54.20±1.56 | 80.39±0.49 |
| MLS | 52.95±3.82 | 78.91±1.47 | 53.90±3.04 | 81.65±1.49 | 62.39±2.13 | 78.39±0.84 | 57.68±0.91 | 79.75±0.24 | 56.73±1.33 | 79.67±0.57 |
| KLM | 73.09±6.67 | 74.15±2.59 | 50.30±7.04 | 79.34±0.44 | 81.80±5.80 | 75.77±0.45 | 81.40±1.58 | 75.70±0.24 | 71.65±2.01 | 76.24±0.52 |
| VIM | 48.32±1.07 | 81.89±1.02 | 46.22±5.46 | 83.14±3.71 | 46.86±2.29 | 85.91±0.78 | 61.57±0.77 | 75.85±0.37 | 50.74±1.00 | 81.70±0.62 |
| KNN | 48.58±4.67 | 82.36±1.52 | 51.75±3.12 | 84.15±1.09 | 53.56±2.32 | 83.66±0.83 | 60.70±1.03 | 79.43±0.47 | 53.65±0.28 | 82.40±0.17 |
| DICE | 51.79±3.67 | 79.86±1.89 | 49.58±3.32 | 84.22±2.00 | 64.23±1.65 | 77.63±0.34 | 59.39±1.25 | 78.33±0.66 | 56.25±0.60 | 80.01±0.18 |
| RankFeat | 75.01±5.83 | 63.03±3.86 | 58.49±2.30 | 72.14±1.39 | 66.87±3.80 | 69.40±3.08 | 77.42±1.96 | 63.82±1.83 | 69.45±1.01 | 67.10±1.42 |
| ASH | 66.58±3.88 | 77.23±0.46 | 46.00±2.67 | 85.60±1.40 | 61.27±2.74 | 80.72±0.70 | 62.95±0.99 | 78.76±0.16 | 59.20±2.46 | 80.58±0.66 |
| SHE | 58.78±2.70 | 76.76±1.07 | 59.15±7.61 | 80.97±3.98 | 73.29±3.22 | 73.64±1.28 | 65.24±0.98 | 76.30±0.51 | 64.12±2.70 | 76.92±1.16 |
| GEN | 53.92±5.71 | 78.29±2.05 | 55.45±2.76 | 81.41±1.50 | 61.23±1.40 | 78.74±0.81 | 56.25±1.01 | 80.28±0.27 | 56.71±1.59 | 79.68±0.75 |
| NAC-UE | 21.97±6.62 | 93.15±1.63 | 24.39±4.66 | 92.40±1.26 | 40.65±1.94 | 89.32±0.55 | 73.57±1.16 | 73.05±0.68 | 40.14±1.86 | 86.98±0.37 |
| *CBD*-De | 27.93±7.73 | 96.29±1.81 | 18.72±6.12 | 95.80±2.05 | 37.34±2.30 | 90.02±1.52 | 55.23±0.88 | 81.50±1.16 | 34.81±1.60 | 90.90±0.68 |

Table 15: OOD detection performance on the ImageNet-1k benchmark. Results are formatted as **first**, second, and third best. Following the OpenOOD protocol, we report the average performance over ResNet-50 and Vit-b16 backbones, each trained exclusively on the InD dataset.

| Method | iNaturalist | | | OpenImage-O | | | Textures | | |
|---|---|---|---|---|---|---|---|---|---|
| | ResNet-50 | Vit-b16 | Average | ResNet-50 | Vit-b16 | Average | ResNet-50 | Vit-b16 | Average |
| OpenMax | 92.05 | 94.93 | 93.49 | 87.62 | 87.36 | 87.49 | 88.10 | 85.52 | 86.81 |
| MSP | 88.41 | 88.19 | 88.30 | 84.86 | 84.86 | 84.86 | 82.43 | 85.06 | 83.75 |
| TempScale | 90.50 | 88.54 | 89.52 | 87.22 | 85.04 | 86.13 | 84.95 | 85.39 | 85.17 |
| ODIN | 91.17 | / | 91.17 | 88.23 | / | 88.23 | 89.00 | / | 89.00 |
| MDS | 63.67 | 96.01 | 79.84 | 69.27 | 92.38 | 80.83 | 89.80 | 89.41 | 89.61 |
| MDSEns | 61.82 | / | 61.82 | 60.80 | / | 60.80 | 79.94 | / | 79.94 |
| RMDS | 87.24 | 96.10 | 91.67 | 85.84 | 92.32 | 89.08 | 86.08 | 89.38 | 87.73 |
| Gram | 76.67 | / | 76.67 | 74.43 | / | 74.43 | 88.02 | / | 88.02 |
| EBO | 90.63 | 79.30 | 84.97 | 89.06 | 76.48 | 82.77 | 88.70 | 81.17 | 84.94 |
| OpenGAN | / | / | / | / | / | / | / | / | / |
| GradNorm | 93.89 | 42.42 | 68.16 | 84.82 | 37.82 | 61.32 | 92.05 | 44.99 | 68.52 |
| ReAct | 96.34 | 86.11 | 91.23 | 91.87 | 84.29 | 88.08 | 92.79 | 86.66 | 89.73 |
| MLS | 91.17 | 85.29 | 88.23 | 89.17 | 81.60 | 85.39 | 88.39 | 83.74 | 86.07 |
| KLM | 90.78 | 89.59 | 90.19 | 87.30 | 87.03 | 87.17 | 84.72 | 86.49 | 85.61 |
| VIM | 89.56 | 95.72 | 92.64 | 90.50 | 92.18 | 91.34 | 97.97 | 90.61 | 94.29 |
| KNN | 86.41 | 91.46 | 88.94 | 87.04 | 89.86 | 88.45 | 97.09 | 91.12 | 94.11 |
| DICE | 92.54 | 82.50 | 87.52 | 88.26 | 82.22 | 85.24 | 92.04 | 82.21 | 87.13 |
| RankFeat | 40.06 | / | 40.06 | 50.83 | / | 50.83 | 70.90 | / | 70.90 |
| ASH | 97.07 | 50.62 | 73.85 | 93.26 | 55.51 | 74.39 | 96.90 | 48.53 | 72.72 |
| SHE | 92.65 | 93.57 | 93.11 | 86.52 | 91.04 | 88.78 | 93.60 | 92.65 | 93.13 |
| GEN | 92.44 | 93.54 | 92.99 | 89.26 | 90.27 | 89.77 | 87.59 | 90.23 | 88.91 |
| NAC-UE | 96.52 | 93.72 | 95.12 | 91.45 | 91.58 | 91.52 | 97.90 | 94.17 | 96.04 |
| *CBD*-De | 97.06 | 97.26 | 97.16 | 92.65 | 91.89 | 92.27 | 98.31 | 94.30 | 96.31 |

# F  Full OOD Generalization Results

Table 16: Out-of-domain accuracies (%) on PACS. Results are formatted as **first**, second, and third best.

| Method | A | C | P | S | Avg |
|---|---|---|---|---|---|
| IRM | $84.8_{\pm1.3}$ | $76.4_{\pm1.1}$ | $96.7_{\pm0.6}$ | $76.1_{\pm1.0}$ | 83.5 |
| ERM | $85.7_{\pm0.6}$ | $77.1_{\pm0.8}$ | $97.4_{\pm0.4}$ | $76.6_{\pm0.7}$ | 84.2 |
| GroupDRO | $83.5_{\pm0.9}$ | $79.1_{\pm0.6}$ | $96.7_{\pm0.7}$ | $78.3_{\pm2.0}$ | 84.4 |
| MTL | $87.5_{\pm0.8}$ | $77.1_{\pm0.5}$ | $96.4_{\pm0.8}$ | $77.3_{\pm1.8}$ | 84.6 |
| Mixup | $86.1_{\pm0.5}$ | $78.9_{\pm0.8}$ | $97.6_{\pm0.1}$ | $75.8_{\pm1.8}$ | 84.6 |
| MMD | $86.1_{\pm1.4}$ | $79.4_{\pm0.9}$ | $96.6_{\pm0.2}$ | $76.5_{\pm0.5}$ | 84.7 |
| VREx | $86.0_{\pm1.6}$ | $79.1_{\pm0.6}$ | $96.9_{\pm0.5}$ | $77.7_{\pm1.7}$ | 84.9 |
| MLDG | $85.5_{\pm1.4}$ | $80.1_{\pm1.7}$ | $97.4_{\pm0.3}$ | $76.6_{\pm1.1}$ | 84.9 |
| ARM | $86.8_{\pm0.6}$ | $76.8_{\pm0.5}$ | $97.4_{\pm0.3}$ | $79.3_{\pm1.2}$ | 85.1 |
| Mixstyle | $86.8_{\pm0.5}$ | $79.0_{\pm1.4}$ | $96.6_{\pm0.1}$ | $78.5_{\pm2.3}$ | 85.2 |
| CORAL | $88.3_{\pm0.2}$ | $80.0_{\pm0.5}$ | $97.5_{\pm0.3}$ | $78.8_{\pm1.3}$ | 86.2 |
| SagNet | $87.4_{\pm0.2}$ | $80.7_{\pm0.5}$ | $97.1_{\pm0.1}$ | $80.0_{\pm1.0}$ | 86.3 |
| RSC | $85.4_{\pm0.9}$ | $79.7_{\pm0.5}$ | $97.6_{\pm0.9}$ | $78.2_{\pm1.0}$ | 85.2 |
| SAM | $85.6_{\pm2.1}$ | $80.9_{\pm1.2}$ | $97.0_{\pm0.4}$ | $79.6_{\pm1.6}$ | 85.8 |
| GSAM | $86.9_{\pm0.1}$ | $80.4_{\pm0.2}$ | $97.5_{\pm0.0}$ | $78.7_{\pm0.8}$ | 85.9 |
| SAGM | $87.4_{\pm0.2}$ | $80.2_{\pm0.3}$ | $98.0_{\pm0.2}$ | $80.8_{\pm0.6}$ | 86.6 |
| GGA | $88.8_{\pm0.2}$ | $80.1_{\pm0.3}$ | $97.3_{\pm0.2}$ | $81.2_{\pm0.5}$ | 87.3 |
| *CBD*-Gen | **$89.0_{\pm0.6}$** | **$81.6_{\pm0.4}$** | **$98.1_{\pm0.3}$** | **$82.1_{\pm0.8}$** | **87.7** |

Table 17: Out-of-domain accuracies (%) on VLCS. Results are formatted as **first**, second, and third best.

| Method | C | L | S | V | Avg |
|---|---|---|---|---|---|
| GroupDRO | $97.3_{\pm0.3}$ | $63.4_{\pm0.9}$ | $69.5_{\pm0.8}$ | $76.7_{\pm0.7}$ | 76.7 |
| MLDG | $97.4_{\pm0.2}$ | $65.2_{\pm0.7}$ | $71.0_{\pm1.4}$ | $75.3_{\pm1.0}$ | 77.2 |
| MTL | $97.8_{\pm0.4}$ | $64.3_{\pm0.3}$ | $71.5_{\pm0.7}$ | $75.3_{\pm1.7}$ | 77.2 |
| ERM | $98.0_{\pm0.3}$ | $64.7_{\pm1.2}$ | $71.4_{\pm1.2}$ | $75.2_{\pm1.6}$ | 77.3 |
| Mixup | $98.3_{\pm0.6}$ | $64.8_{\pm1.0}$ | $72.1_{\pm0.5}$ | $74.3_{\pm0.8}$ | 77.4 |
| MMD | $97.7_{\pm0.1}$ | $64.0_{\pm1.1}$ | $72.8_{\pm0.2}$ | $75.3_{\pm3.3}$ | 77.5 |
| ARM | $98.7_{\pm0.2}$ | $63.6_{\pm0.7}$ | $71.3_{\pm1.2}$ | $76.7_{\pm0.6}$ | 77.6 |
| SagNet | $97.9_{\pm0.4}$ | $64.5_{\pm0.5}$ | $71.4_{\pm1.3}$ | $77.5_{\pm0.5}$ | 77.8 |
| Mixstyle | $98.6_{\pm0.3}$ | $64.5_{\pm1.1}$ | $72.6_{\pm0.5}$ | $75.7_{\pm1.7}$ | 77.9 |
| VREx | $98.4_{\pm0.3}$ | $64.4_{\pm1.4}$ | $74.1_{\pm0.4}$ | $76.2_{\pm1.3}$ | 78.3 |
| IRM | $98.6_{\pm0.1}$ | $64.9_{\pm0.9}$ | $73.4_{\pm0.6}$ | $77.3_{\pm0.9}$ | 78.6 |
| CORAL | $98.3_{\pm0.3}$ | $66.1_{\pm0.6}$ | $73.4_{\pm0.3}$ | $77.5_{\pm1.0}$ | 78.8 |
| RSC | $97.9_{\pm0.1}$ | $62.5_{\pm0.7}$ | $72.3_{\pm1.2}$ | $75.6_{\pm0.8}$ | 77.1 |
| GSAM | $98.7_{\pm0.3}$ | $64.9_{\pm0.2}$ | $74.3_{\pm0.0}$ | $78.5_{\pm0.8}$ | 79.1 |
| SAM | **$99.1_{\pm0.2}$** | $65.0_{\pm1.0}$ | $73.7_{\pm1.0}$ | $79.8_{\pm0.1}$ | 79.4 |
| SAGM | $99.0_{\pm0.2}$ | $65.2_{\pm0.4}$ | **$75.1_{\pm0.3}$** | **$80.7_{\pm0.8}$** | 80.0 |
| GGA | **$99.1_{\pm0.2}$** | **$67.5_{\pm0.6}$** | **$75.1_{\pm0.3}$** | $78.0_{\pm0.1}$ | 79.9 |
| *CBD*-Gen | **$99.1_{\pm0.2}$** | $67.3_{\pm0.7}$ | **$75.1_{\pm0.8}$** | $80.5_{\pm1.0}$ | **80.5** |

Table 18: Out-of-domain accuracies (%) on OfficeHome. Results are formatted as **first**, second, and third best.

| Algorithm | A | C | P | R | Avg |
|---|---|---|---|---|---|
| Mixstyle | $51.1_{\pm0.3}$ | $53.2_{\pm0.4}$ | $68.2_{\pm0.7}$ | $69.2_{\pm0.6}$ | 60.4 |
| IRM | $58.9_{\pm2.3}$ | $52.2_{\pm1.6}$ | $72.1_{\pm2.9}$ | $74.0_{\pm2.5}$ | 64.3 |
| ARM | $58.9_{\pm0.8}$ | $51.0_{\pm0.5}$ | $74.1_{\pm0.1}$ | $75.2_{\pm0.3}$ | 64.8 |
| GroupDRO | $60.4_{\pm0.7}$ | $52.7_{\pm1.0}$ | $75.0_{\pm0.7}$ | $76.0_{\pm0.7}$ | 66.0 |
| MMD | $60.4_{\pm0.2}$ | $53.3_{\pm0.3}$ | $74.3_{\pm0.1}$ | $77.4_{\pm0.6}$ | 66.4 |
| MTL | $61.5_{\pm0.7}$ | $52.4_{\pm0.6}$ | $74.9_{\pm0.4}$ | $76.8_{\pm0.4}$ | 66.4 |
| VREx | $60.7_{\pm0.9}$ | $53.0_{\pm0.9}$ | $75.3_{\pm0.1}$ | $76.6_{\pm0.5}$ | 66.4 |
| ERM | $63.1_{\pm0.3}$ | $51.9_{\pm0.4}$ | $77.2_{\pm0.5}$ | $78.1_{\pm0.2}$ | 67.6 |
| MLDG | $61.5_{\pm0.9}$ | $53.2_{\pm0.6}$ | $75.0_{\pm1.2}$ | $77.5_{\pm0.4}$ | 66.8 |
| Mixup | $62.4_{\pm0.8}$ | $54.8_{\pm0.6}$ | $76.9_{\pm0.3}$ | $78.3_{\pm0.2}$ | 68.1 |
| SagNet | $63.4_{\pm0.2}$ | $54.8_{\pm0.4}$ | $75.8_{\pm0.4}$ | $78.3_{\pm0.3}$ | 68.1 |
| CORAL | $65.3_{\pm0.3}$ | $54.4_{\pm0.6}$ | $76.5_{\pm0.3}$ | $78.4_{\pm1.0}$ | 68.7 |
| RSC | $60.7_{\pm1.4}$ | $51.4_{\pm0.3}$ | $74.8_{\pm1.1}$ | $75.1_{\pm1.3}$ | 65.5 |
| GSAM | $64.9_{\pm0.1}$ | $55.2_{\pm0.2}$ | $77.8_{\pm0.0}$ | $79.2_{\pm0.2}$ | 69.3 |
| SAM | $64.5_{\pm0.3}$ | $56.5_{\pm0.2}$ | $77.4_{\pm0.1}$ | $79.8_{\pm0.4}$ | 69.6 |
| SAGM | $65.4_{\pm0.4}$ | $57.0_{\pm0.3}$ | $78.0_{\pm0.3}$ | $80.0_{\pm0.2}$ | 70.1 |
| GGA | $64.3_{\pm0.1}$ | $54.4_{\pm0.2}$ | $76.5_{\pm0.3}$ | $78.9_{\pm0.2}$ | 68.5 |
| ***CBD*-Gen** | **$66.5_{\pm0.9}$** | **$58.7_{\pm0.7}$** | **$78.8_{\pm1.2}$** | **$81.6_{\pm0.8}$** | **71.4** |

Table 19: Out-of-domain accuracies (%) on TerraInc. Results are formatted as **first**, second, and third best.

| Method | L100 | L38 | L43 | L46 | Avg |
|---|---|---|---|---|---|
| MMD | $41.9_{\pm3.0}$ | $34.8_{\pm1.0}$ | $57.0_{\pm1.9}$ | $35.2_{\pm1.8}$ | 42.2 |
| GroupDRO | $41.2_{\pm0.7}$ | $38.6_{\pm2.1}$ | $56.7_{\pm0.9}$ | $36.4_{\pm2.1}$ | 43.2 |
| Mixstyle | $54.3_{\pm1.1}$ | $34.1_{\pm1.1}$ | $55.9_{\pm1.1}$ | $31.7_{\pm2.1}$ | 44.0 |
| ARM | $49.3_{\pm0.7}$ | $38.3_{\pm0.7}$ | $55.8_{\pm0.8}$ | $38.7_{\pm1.3}$ | 45.5 |
| MTL | $49.3_{\pm1.2}$ | $39.6_{\pm6.3}$ | $55.6_{\pm1.1}$ | $37.8_{\pm0.8}$ | 45.6 |
| ERM | $49.8_{\pm4.4}$ | $42.1_{\pm1.4}$ | $56.9_{\pm1.8}$ | $35.7_{\pm3.9}$ | 46.1 |
| VREx | $48.2_{\pm4.3}$ | $41.7_{\pm1.3}$ | $56.8_{\pm0.8}$ | $38.7_{\pm3.1}$ | 46.4 |
| IRM | $54.6_{\pm1.3}$ | $39.8_{\pm1.9}$ | $56.2_{\pm1.8}$ | $39.6_{\pm0.8}$ | 47.6 |
| CORAL | $51.6_{\pm2.4}$ | $42.2_{\pm1.0}$ | $57.0_{\pm1.0}$ | $39.8_{\pm2.9}$ | 47.6 |
| MLDG | $54.2_{\pm3.0}$ | $44.3_{\pm1.1}$ | $55.6_{\pm0.3}$ | $36.9_{\pm2.2}$ | 47.8 |
| Mixup | **$59.6_{\pm2.0}$** | $42.2_{\pm1.4}$ | $55.9_{\pm0.8}$ | $33.9_{\pm1.4}$ | 47.9 |
| SagNet | $53.0_{\pm2.0}$ | $43.0_{\pm1.4}$ | $57.9_{\pm0.8}$ | $40.4_{\pm1.4}$ | 48.6 |
| SAM | $46.3_{\pm1.0}$ | $38.4_{\pm2.4}$ | $54.0_{\pm1.0}$ | $34.5_{\pm0.8}$ | 43.3 |
| RSC | $50.2_{\pm2.2}$ | $39.2_{\pm1.4}$ | $56.3_{\pm1.4}$ | $40.8_{\pm0.6}$ | 46.6 |
| GSAM | $50.8_{\pm0.1}$ | $39.3_{\pm0.2}$ | $59.6_{\pm0.0}$ | $38.2_{\pm0.8}$ | 47.0 |
| SAGM | $54.8_{\pm1.3}$ | $41.4_{\pm0.8}$ | $57.7_{\pm0.6}$ | $41.3_{\pm0.4}$ | 48.8 |
| GGA | $55.9_{\pm0.1}$ | **$45.5_{\pm0.1}$** | **$59.7_{\pm0.1}$** | **$41.5_{\pm0.1}$** | **50.6** |
| ***CBD*-Gen** | $56.4_{\pm2.1}$ | $45.0_{\pm1.2}$ | $59.6_{\pm0.6}$ | $41.0_{\pm1.5}$ | 50.5 |

Table 20: Out-of-domain accuracies (%) on DomainNet. Results are formatted as **first**, second, and third best.

| Algorithm | clip | info | paint | quick | real | sketch | Avg |
|---|---|---|---|---|---|---|---|
| MMD | 32.1 $_{\pm 13.3}$ | 11.0 $_{\pm 4.6}$ | 26.8 $_{\pm 11.3}$ | 8.7 $_{\pm 2.1}$ | 32.7 $_{\pm 13.8}$ | 28.9 $_{\pm 11.9}$ | 23.4 |
| GroupDRO | 47.2 $_{\pm 0.5}$ | 17.5 $_{\pm 0.4}$ | 33.8 $_{\pm 0.5}$ | 9.3 $_{\pm 0.3}$ | 51.6 $_{\pm 0.4}$ | 40.1 $_{\pm 0.6}$ | 33.3 |
| VREx | 47.3 $_{\pm 3.5}$ | 16.0 $_{\pm 1.5}$ | 35.8 $_{\pm 4.6}$ | 10.9 $_{\pm 0.3}$ | 49.6 $_{\pm 4.9}$ | 42.0 $_{\pm 3.0}$ | 33.6 |
| IRM | 48.5 $_{\pm 2.8}$ | 15.0 $_{\pm 1.5}$ | 38.3 $_{\pm 4.3}$ | 10.9 $_{\pm 0.5}$ | 48.2 $_{\pm 5.2}$ | 42.3 $_{\pm 3.1}$ | 33.9 |
| Mixstyle | 51.9 $_{\pm 0.4}$ | 13.3 $_{\pm 0.2}$ | 37.0 $_{\pm 0.5}$ | 12.3 $_{\pm 0.1}$ | 46.1 $_{\pm 0.3}$ | 43.4 $_{\pm 0.4}$ | 34.0 |
| ARM | 49.7 $_{\pm 0.3}$ | 16.3 $_{\pm 0.5}$ | 40.9 $_{\pm 1.1}$ | 9.4 $_{\pm 0.1}$ | 53.4 $_{\pm 0.4}$ | 43.5 $_{\pm 0.4}$ | 35.5 |
| Mixup | 55.7 $_{\pm 0.3}$ | 18.5 $_{\pm 0.5}$ | 44.3 $_{\pm 0.5}$ | 12.5 $_{\pm 0.4}$ | 55.8 $_{\pm 0.3}$ | 48.2 $_{\pm 0.5}$ | 39.2 |
| SagNet | 57.7 $_{\pm 0.3}$ | 19.0 $_{\pm 0.2}$ | 45.3 $_{\pm 0.3}$ | 12.7 $_{\pm 0.5}$ | 58.1 $_{\pm 0.5}$ | 48.8 $_{\pm 0.2}$ | 40.3 |
| MTL | 57.9 $_{\pm 0.5}$ | 18.5 $_{\pm 0.4}$ | 46.0 $_{\pm 0.1}$ | 12.5 $_{\pm 0.1}$ | 59.5 $_{\pm 0.3}$ | 49.2 $_{\pm 0.1}$ | 40.6 |
| MLDG | 59.1 $_{\pm 0.2}$ | 19.1 $_{\pm 0.3}$ | 45.8 $_{\pm 0.7}$ | 13.4 $_{\pm 0.3}$ | 59.6 $_{\pm 0.2}$ | 50.2 $_{\pm 0.4}$ | 41.2 |
| CORAL | 59.2 $_{\pm 0.1}$ | 19.7 $_{\pm 0.2}$ | 46.6 $_{\pm 0.3}$ | 13.4 $_{\pm 0.4}$ | 59.8 $_{\pm 0.2}$ | 50.1 $_{\pm 0.6}$ | 41.5 |
| ERM | 63.0 $_{\pm 0.2}$ | 21.2 $_{\pm 0.2}$ | 50.1 $_{\pm 0.4}$ | 13.9 $_{\pm 0.5}$ | 63.7 $_{\pm 0.2}$ | 52.0 $_{\pm 0.5}$ | 43.8 |
| RSC | 55.0 $_{\pm 1.2}$ | 18.3 $_{\pm 0.5}$ | 44.4 $_{\pm 0.6}$ | 12.2 $_{\pm 0.2}$ | 55.7 $_{\pm 0.7}$ | 47.8 $_{\pm 0.9}$ | 38.9 |
| SAM | 64.5 $_{\pm 0.3}$ | 20.7 $_{\pm 0.2}$ | 50.2 $_{\pm 0.1}$ | **15.1** $_{\pm 0.3}$ | 62.6 $_{\pm 0.2}$ | 52.7 $_{\pm 0.3}$ | 44.3 |
| GSAM | 64.2 $_{\pm 0.3}$ | 20.8 $_{\pm 0.2}$ | 50.9 $_{\pm 0.0}$ | 14.4 $_{\pm 0.8}$ | 63.5 $_{\pm 0.2}$ | 53.9 $_{\pm 0.2}$ | 44.6 |
| SAGM | **64.9** $_{\pm 0.2}$ | 21.1 $_{\pm 0.3}$ | 51.5 $_{\pm 0.2}$ | 14.8 $_{\pm 0.2}$ | 64.1 $_{\pm 0.2}$ | 53.6 $_{\pm 0.2}$ | 45.0 |
| GGA | 64.0 $_{\pm 0.2}$ | 22.2 $_{\pm 0.3}$ | 51.7 $_{\pm 0.1}$ | 14.3 $_{\pm 0.2}$ | 64.1 $_{\pm 0.4}$ | 54.3 $_{\pm 0.3}$ | 45.2 |
| *CBD*-Gen | **64.9** $_{\pm 0.4}$ | **22.8** $_{\pm 0.3}$ | **52.5** $_{\pm 0.3}$ | **15.1** $_{\pm 0.6}$ | **64.7** $_{\pm 0.4}$ | **55.4** $_{\pm 0.7}$ | **45.9** |

