# OpenReview forum: "Rethinking Out-of-Distribution Detection and Generalization with Collective Behavior Dynamics"
_NeurIPS.cc/2025/Conference — NeurIPS 2025 poster_

### Official Review · Reviewer_Sdub · 2025-06-25

**Clarity:** 2
**Significance:** 3
**Originality:** 3
**Rating:** 5
**Confidence:** 3

**Summary:**

This paper introduces a novel, physics-inspired framework CBD to address Out-of-Distribution (OOD) detection and generalization. The authors conceptualize high-level features extracted from neural networks as interacting charged particles governed by the Vlasov-Poisson equations, analyzing their collective behavior within a dynamically evolving self-consistent electric field. They formally define a basin of attraction that distinguishes In-Distribution (InD) from OOD data and propose two distinct but complementary methods derived from this framework: (i) an OOD detection method based on identifying steady-state solutions to distinguish InD and OOD features, and (ii) an OOD generalization method leveraging a dispersion relation to stabilize perturbations and enlarge the basin boundary.

**Questions:**

Please refer to Weaknesses.

**Ethical Concerns:**

["NO or VERY MINOR ethics concerns only"]

**Final Justification:**

The authors' responses address the majority of my concerns. I am therefore willing to raise my overall evaluation score.

**Limitations:**

Yes

**Quality:**

3

**Strengths And Weaknesses:**

Strengths:
1. The paper uniquely leverages theoretical constructs from physics (Vlasov-Poisson dynamics, charged particle behavior) to fundamentally rethink the OOD detection and generalization problems, offering a refreshing theoretical perspective distinct from prevailing statistical and uncertainty-based methods.
2. The theoretical underpinning is compelling, outlining why the proposed dynamics-based approach naturally differentiates between InD and OOD data through steady-state solutions and basin boundaries.
3. The method can be integrated smoothly with other advanced OOD generalization techniques, adding to its practical utility and adaptability.

Weaknesses:
1. The method requires the simultaneous estimation of complex PDE solutions, potentially resulting in increased GPU memory usage and reduced computational efficiency, especially in very large-scale or resource-constrained environments.
2. While some hyperparameter analyses are presented, a deeper understanding of the sensitivity and stability related to PDE discretization parameters, regularization strengths, or solver accuracy would significantly enhance the method’s clarity and reproducibility.
3. The experimental evaluation is limited to relatively standard feature-extraction architectures, specifically ResNet and ViT, leaving the effectiveness of the approach on more diverse network architectures or modalities (e.g., text or multimodal data) unexplored.

---

> ### Author Rebuttal · Authors · 2025-07-31
>
> Dear reviewer Sdub, thank you for your thoughtful review and constructive feedback on our manuscript. We are pleased that you recognize the refreshing theoretical perspective our proposed method brings, which distinguishes it from conventional approaches in the field. We address your specific comments and suggestions in detail below:
>
> ---
>
> > Q1: **Weaknesses-1:** The method requires the simultaneous estimation of complex PDE solutions, potentially resulting in increased GPU memory usage and reduced computational efficiency, especially in very large-scale or resource-constrained environments.
>
> Thank you for your comment. A similar concern was raised by reviewer $\text{\textcolor{red}{jZnV}}$ in $\text{\textcolor{blue}{Q5}}$, and we have provided a detailed response there. Briefly, we acknowledge that our method introduces additional computational overhead during training due to the storage of intermediate computation graphs and automatic gradient calculations. To mitigate this, we adopt two strategies:
> - Directly parameterizing steady-state solution instead of the full time-dependent functions.
> - Using the Poisson-Boltzmann approximation to avoid solving the full integral form of the PDE.
>
> These design choices reduce GPU memory usage while preserving the core physics-informed structure. At inference time, the overhead is negligible, and the additional model parameters account for only a small fraction of the total. Please refer to our response to reviewer $\text{\textcolor{red}{jZnV}}$'s $\text{\textcolor{blue}{Q5}}$ for further details.
>
> ---
>
> > Q2: **Weaknesses-2:** While some hyperparameter analyses are presented, a deeper understanding of the sensitivity and stability related to PDE discretization parameters, regularization strengths, or solver accuracy would significantly enhance the method's clarity and reproducibility.
>
> We appreciate this insightful suggestion. However, we'd like to clarify a potential misunderstanding: Our method does **not** rely on an ODE/PDE solver to iteratively fit the particle distribution function $F(\mathbf{z}, \mathbf{v}, t)$ and electric potential $ \phi (\mathbf{z}, t)$ over time steps. Instead, we use MLPs to directly predict the steady-state \$F^* (\mathbf{z}, \mathbf{v})$ and \$\phi^* (\mathbf{z})$ in a single forward pass, completely bypassing numerical time integration. This fundamental distinction means that conventional ODE solver parameters (discretization step sizes, solver tolerance, numerical stability constraints, etc.) are not relevant to our method.
>
> That said, we recognize the value in analyzing the sensitivity of MLPs hyperparameters (e.g., layer depth, width), which could impact the stability of the steady-state approximation. We have examined these variations (hyperparameter tuning details are in our response to reviewer $\text{\textcolor{red}{MQNz}}$'s question $\text{\textcolor{blue}{Q12}}$). Below, we present the sensitivity analysis results for several relevant parameters:
>
> **Table 3: OOD Detection on CIFAR-100 with MLP Hidden Dimension of 512**
> |  |  |  |
> | :--- | :---: | :---: |
> | **Layer_depth** | **FPR95 ↓** | **AUROC ↑** |
> | **1** | 39.30 | 87.14 |
> | **2** | 34.81 | 90.90 |
> | | | |
>
>
>
> **Table 4: OOD Detection on CIFAR-100 with MLP Depth of 2 Layers**
> |  |  |  |
> | :--- | :---: | :---: |
> | **Hidden_dim** | **FPR95 ↓** | **AUROC ↑** |
> | **128** | 40.35 | 86.93 |
> | **256** | 38.52 | 88.20 |
> | **512** | 34.81 | 90.90 |
> | **1024** | 35.25 | 90.83 |
> |  |  |  |
>
> Table 3 and Table 4 show that performance scales with the number of MLP parameters, underscoring the key role of these hyperparameters in improving the model's ability to approximate steady state solutions. While increasing MLP architectural complexity delivers gains, returns diminish beyond a certain threshold (e.g., hidden dim=512 vs. 1024). This highlights the need to balance MLP intricacy with computational efficiency during tuning. We believe these insights will inform further optimizations of our approach for real-world applications.
>
> ---
>
> > Q3: **Weaknesses-3:** The experimental evaluation is limited to relatively standard feature-extraction architectures, specifically ResNet and ViT, leaving the effectiveness of the approach on more diverse network architectures or modalities (e.g., text or multimodal data) unexplored.
>
> We agree that more experimental evaluation would be useful to understand the details ofinteraction andenhancement. We would like to address this concern from several perspectives:
>
> - Our experimental design strictly **align with** established evaluation protocols for OOD detection and generalization in the visual domain. Specifically, we adopted the experimental **setups from the well-regarded benchmark frameworks OpenOOD [1] and DomainBed [2]**, including their datasets, evaluation metrics, and procedures. This ensures our results are comparable and consistent with recent existing works [3,4,5]. The datasets and backbone network models were carefully selected for their broad representativeness and ability to thoroughly validate the effectiveness of our approach.
>
> - Extending experimental evaluation from visual data to text, tabular data or multimodal data would require fundamental redesign of our method and development of specialized components for different modality characteristics. Text and multimodal OOD problems differ inherently from visual OOD problems and require distinct technical approaches, including but not limited to: prompt engineering, modality-specific fine-tuning strategies, cross-modal alignment techniques, and multimodal fusion mechanisms. **These technical challenges constitute separate research directions in their own right**.
>
> - Nevertheless, to address your suggestion and preliminarily explore the extensibility of our method, we conducted exploratory experiments on multimodal model CLIP. Specifically, we followed the ERM with CLIP method described in [6] and adopted the following experimental setup: we first initialized the classifier head weights using text embeddings extracted by the CLIP text encoder. Then, we fine-tuned the CLIP image encoder along with two additional MLP branches, with the overall experimental configuration following the standard implementation from [7]. Unlike [7] which only used standard ERM loss, we additionally integrated our three proposed loss components during training: the Vlasov loss $\mathcal{L}_ {\text{Vlasov}}$, the Poisson loss $\mathcal{L}_ {\text{Poisson}}$, and the Dispersion Relation loss $\mathcal{L}_ {\text{disp}}$. The results are shown in Table 5. This preliminary multimodal experiment validates the potential extensibility of our approach. We hope this additional multimodal experiment and the detailed explanation above effectively address your concerns while clearly indicating that our current research remains focused on image-based OOD problems and their theoretical foundations.
>
>
> **Table 5: Accuracy on the PACS and OfficeHome Datasets with Domain Shift.**
>
> |||||
> | :--- | :---: | :---: | :---: |
> | **Method** | **Backbone** | **PACS** | **OfficeHome**|
> | **Zero-shot** | CLIP |96.2 | 82.0 |
> | **ERM** | CLIP | 96.1 | 83.3|
> | **ERM (with our loss)**| CLIP | 96.4 (${\small \text{\textcolor{red}{+0.3}}}$) | 84.0 (${\small \text{\textcolor{red}{+0.7}}}$) |
> |||||
>
> We fully agree with your suggestion. For the specific cases you mentioned, our method will require further customization. The approach based on discrete electric fields is conceptually similar to addressing OOD and related issues, and we plan to further explore this direction in future work.
>
> ---
> [1] Openood: Benchmarking generalized out-of-distribution detection. NeurIPS 2022.
> [2] In Search of Lost Domain Generalization. ICLR 2021.
> [3] Gradient-Guided Annealing for Domain Generalization. CVPR 2025.
> [4] Extremely Simple Activation Shaping for Out-of-Distribution Detection. ICLR 2023.
> [5] Out-of-distribution detection based on in-distribution data patterns memorization with modern hopfield energy. ICLR 2023.
> [6] Clipood: Generalizing clip to out-of-distributions. ICML 2023.
> [7] Robust fine-tuning of zero-shot models. CVPR 2022.

---

> > ### Comment · Reviewer_Sdub · 2025-08-02
> >
> > Thank you for clarifying the distinction regarding your approach. Given that your method employs MLPs to directly predict the steady state distribution function and steady-state potential in a single forward pass, rather than solving the ODE/PDE through numerical time integration, could you provide an analysis or estimation of the gap in accuracy or fidelity between your direct prediction approach and conventional numerical solvers? Specifically, how closely do the steady-state solutions predicted by your MLP align with those obtained through numerical time integration methods? Addressing this gap explicitly would enhance the theoretical robustness and practical credibility of your proposed method.

---

> > > ### Author Response · Authors · 2025-08-03
> > > **Thank You!**
> > >
> > > Dear reviewer Sdub, thank you very much for raising this critical and insightful question, and we have taken it seriously.
> > >
> > > To address your concerns, we designed and conducted additional experiments to quantify the AUROC gap between MLP direct steady-state prediction and numerical solvers using time integration for PDE solutions. Using the DeepXDE library [1] (an ODE/PDE package), we configure the PDE solver with MLP architecture (2 layers, 512 hidden dimensions) and test both 10 and 30 time steps. Complete experimental details and results are presented in Table 6 and Table 7.
> > >
> > > **Table 6: OOD Detection AUROC Performance Comparison (Time Steps = 10)**
> > >
> > > ||||||||
> > > | :--- | :---: | :---: |:---:|:---:|:---:|:---:|
> > > | **Dataset** | **Sample_Size** | **Classes** | **Complexity** | &nbsp;&nbsp;&nbsp;**MLP**&nbsp;&nbsp;&nbsp; | &nbsp;**PDE_Slover**&nbsp; | **Forward_Pass_Time**|
> > > | CIFAR-10 | 50,000  |  10 |  Low | 96.39 | 96.41 (${\small \text{\textcolor{red}{+0.02}}}$) | $\text{\textcolor{red}{7\textsf{x}}}$ slower|
> > > |  CIFAR-100 |  50,000 |  100|  High |  90.90 | 91.20(${\small \text{\textcolor{red}{+0.30}}}$) | $\text{\textcolor{red}{7\textsf{x}}}$ slower|
> > > ||||||||
> > >
> > > **Table 7: OOD Detection AUROC Performance Comparison (Time Steps = 30)**
> > > ||||||||
> > > | :--- | :---: | :---: |:---:|:---:|:---:|:---:|
> > > | **Dataset** | **Sample_Size** | **Classes** | **Complexity** | &nbsp;&nbsp;&nbsp;**MLP**&nbsp;&nbsp;&nbsp; | &nbsp;**PDE_Slover**&nbsp; | **Forward_Pass_Time**|
> > > | CIFAR-10 | 50,000  |  10 |  Low | 96.39 | 96.45 (${\small \text{\textcolor{red}{+0.06}}}$) | $\text{\textcolor{red}{23\textsf{x}}}$ slower|
> > > |  CIFAR-100 |  50,000 |  100|  High |  90.90 | 91.32 (${\small \text{\textcolor{red}{+0.42}}}$) | $\text{\textcolor{red}{23\textsf{x}}}$ slower|
> > > ||||||||
> > >
> > > Table 6 and Table 7 reveals that the performance relationship between MLP and PDE solvers methods strongly correlates with dataset complexity. For relatively simple datasets, the direct prediction method achieves AUROC very close to that of PDE solvers. For instance, on CIFAR-10 (10 classes, simple distribution), the AUROC difference between MLP predictions and PDE numerical integration solutions is merely **0.02** and **0.06**. However, we must acknowledge that for the more complex CIFAR-100 dataset (100 classes, complex distribution), PDE solvers outperform the MLP method, with AUROC gaps reaching **0.30** and **0.42**.
> > >
> > > This performance difference can possibly be explained through error propagation analysis. Suppose a small perturbation $\epsilon$ is introduced to the test input. Comparing single-step prediction $y_1 = f(x + \epsilon)$ to multi-step prediction $y_n = f_n \circ \cdots \circ f_1(x + \epsilon)$, we analyze the propagated error. A first-order Taylor expansion shows the error in the single-step case is approximately $|f'(x)|\epsilon + O(\epsilon^2)$, while for the multi-step case, it becomes $\prod_{i=1}^n |f_i'(x_i)|\epsilon + O(\epsilon^2)$. If each sub-function is well-designed such that $|f_i'(x)| < 1$, then $\prod_{i=1}^n |f_i'(x_i)| < |f'(x)|$, indicating that the multi-step process accumulates less error. This stems from decomposing complex transformations into smoother intermediate steps, avoiding large gradients that amplify single-step errors. When class diversity is limited (as in CIFAR-10), the perturbation $\epsilon$ remains small and both methods perform similarly. But with increased class diversity (as in CIFAR-100), $\epsilon$ tends to be larger, making the multi-step integration approach more reliable.
> > >
> > > While we recognize PDE solvers' performance advantages for high-complexity problems, the computational costs pose significant practical challenges. Table 6 and Table 7 demonstrates that PDE solvers require several orders of magnitude more computation time than our direct MLP approach. For applications demanding real-time response or large-scale parameter studies, such computational overhead becomes prohibitive. Additionally, the substantial GPU memory requirements of PDE numerical integration methods create challenges in resource-constrained environments.
> > >
> > > Our choice of the MLP direct steady-state prediction approach reflects a careful trade-off between performance and efficiency. Importantly, **even when some accuracy compromises are present, our MLP direct steady-state prediction method consistently outperforms existing baselines across all tested datasets.**
> > >
> > > Thank you again for this important question. We hope our above response addresses your concerns and welcome any further guidance on our work.
> > >
> > > ---
> > >
> > > [1] Lu, Lu, et al. "DeepXDE: A deep learning library for solving differential equations." SIAM review 63.1 (2021): 208-228.

---

> > > > ### Comment · Reviewer_Sdub · 2025-08-06
> > > >
> > > > The authors' responses address the majority of my concerns. I am therefore willing to raise my overall evaluation score.

---

> > > > > ### Author Response · Authors · 2025-08-07
> > > > > **Thank you so much!**
> > > > >
> > > > > Dear reviewer Sdub, thank you very much for your thoughtful feedback and for increasing your score. I truly appreciate the time and effort you dedicated to reviewing my work, as well as your constructive comments. Your support means a great deal, and it has been a pleasure to revise the manuscript based on your insights.

---

### Official Review · Reviewer_jZnV · 2025-06-27

**Clarity:** 2
**Significance:** 2
**Originality:** 3
**Rating:** 4
**Confidence:** 3

**Summary:**

This paper introduces a novel physics-inspired framework called Collective Behavior Dynamics (CBD) to address out-of-distribution (OOD) detection and generalization. It conceptualizes high-level features as charged particles evolving under a self-consistent electric field, modeled via the Vlasov–Poisson system, and defines the “basin of attraction” as the region where in-distribution (InD) features converge to a steady state. The authors show that OOD samples can be identified as those falling outside this basin, while model generalization can be enhanced by enlarging the basin through dispersion relation regularization. According to evaluation results, CBD achieves competitive performance across multiple OOD detection and generalization benchmarks, and is compatible with existing methods for further performance gains. This approach bridges dynamical systems theory with deep learning, offering both theoretical insights and practical benefits for robust model behavior under distribution shifts.

**Questions:**

- What is the theoretical or empirical justification for modeling high-level features as charged particles governed by the Vlasov–Poisson system, as opposed to using conventional statistical methods? It is beneficial if the authors provide clearer rationale and comparative discussion that contrasts this physics-inspired approach with established statistical paradigms (e.g., energy-based models, density estimation, confidence scoring). What unique modeling capabilities or interpretability does the proposed framework offer?
- Are there specific cases where CBD fails to detect OOD or generalize well? What are the limitations in terms of types of shifts or datasets?
- Can the authors evaluate CBD in additional domains (e.g., time series, tabular data) or on more realistic OOD scenarios (e.g., long-tailed or adversarial settings)?

**Ethical Concerns:**

["NO or VERY MINOR ethics concerns only"]

**Final Justification:**

As most of my concerns have been addressed, I have decided to increase my score to 4.

**Limitations:**

- The experimental design is limited to imaging data. Generalization beyond vision tasks is not discussed, leaving unclear how well the method transfers to domains like healthcare, finance, or natural language, where OOD robustness is also critical.
- The proposed approach introduces multiple MLP branches (for learning potential and distribution functions), PDE-based losses, and Fourier-domain regularization, all of which significantly increase computational and memory overhead.
- Several derivations (e.g., use of plane wave decomposition, homogeneous steady state) rely on simplifying assumptions (e.g., uniform ion background, constant velocities) that may not capture the full complexity of real-world feature distributions.

**Quality:**

3

**Strengths And Weaknesses:**

Strengths:

- Treating high-level features as particles in a dynamical system offers a new paradigm for OOD detection/generalization grounded in physical systems rather than statistical heuristics.
- The method supports both OOD detection and generalization using the same underlying principles, and defines clear, interpretable loss functions (Poisson, Vlasov, and dispersion losses) tied directly to physical laws.
- Code is provided for reproducibility purposes.

Weaknesses:

- This paper lacks a comprehensive comparison between physics-based and statistical approaches for OOD detection and generalization. While the proposed method is rooted in a novel physics-inspired framework, the authors do not provide sufficient discussion on how it fundamentally differs from or improves upon traditional statistical methods. Specifically, it remains unclear why modeling high-level features as particles under self-consistent fields offers a more effective or principled solution compared to well-established statistical techniques such as energy-based models, density estimation, or confidence-based scoring. A deeper theoretical and empirical analysis comparing the strengths, limitations, and use-case suitability of both perspectives would significantly strengthen the motivation and positioning of the proposed approach.
- The framework assumes that high-level features behave like charged particles in a self-consistent field and reach a steady state. While this is an elegant abstraction, it lacks empirical or theoretical grounding to show that this analogy meaningfully holds for deep neural networks.
- The limitation of the proposed theoretical framework is not discussed. In particular, this framework doesn’t address conditions under which the Vlasov–Poisson-based formulation might fail or be unstable, especially in highly non-linear or high-dimensional feature spaces.
- The method is built around visual features extracted by CNNs or ViTs, and it's unclear whether the approach generalizes well to non-image modalities (e.g., text, graphs, time series).
- The writing is overly complex and lacks clear organization, which makes it challenging to fully grasp the core concepts of CBD, particularly its application to OOD detection and generalization, as well as the accompanying theoretical analysis.

---

> ### Author Rebuttal · Authors · 2025-07-29
>
> Thank you for your thoughtful suggestions and constructive feedback. We're delighted that you see our work as providing a fresh perspective for OOD detection and generalization. We have responded to your comments and questions below.
>
> ---
>
> > Q1: **Weaknesses-1:** This paper lacks a comprehensive comparison between... how it fundamentally differs from...
> > **Weakness-2:** The framework assumes... this analogy meaningfully holds for deep neural networks.
> > **Questions-1:** What is the theoretical or empirical justification for modeling high-level features as charged particles... What unique modeling capabilities or interpretability does the proposed framework offer?
>
> We appreciate your thoughtful question. Since the first and second point in Weaknesses and the first point in Questions address similar concerns, we respond to both together. The following we will explore the core aspects of this question from three aspects:
>
> **1. Why model features as charged particles under Vlasov–Poisson system?**
>
> Our adoption of the Vlasov-Poisson system is methodologically motivated by the need to explicitly model global interactions among high-level features in deep network latent spaces. The theoretical foundation is as follows:
>
> - Current some OOD detection and generalization approaches utilizing KNN methods (score computation based on sample-neighbor distances) [1] or local graph-based structures [2,3] (scores derived from graph isomorphism/connectivity) **implicitly** encode feature interactions. These interactions naturally parallel physical forces, where similar features exhibit mutual attraction and dissimilar features demonstrate repulsion, analogous to electrostatic interactions among charged particles. The Vlasov-Poisson system **explicitly** captures this phenomenon through a self-consistent field related formulation: each particle (representing high-level features) $\mathbf{z}$ evolves under the collective field $E(\mathbf{z}, t)$ generated by all particles (governed by equation $\frac{\partial F}{\partial t}+\mathbf{v} \cdot \nabla_\mathbf{z} F+\nabla_\mathbf{v} \cdot \mathbf{E}=0$), effectively aggregating global information. The resulting steady-state particle distribution function $F^* (\mathbf{z}, \mathbf{v})$ and steady-state potential $\phi^* (\mathbf{z})$ serve as natural scoring mechanisms.
> - Crucially, while traditional approaches such as KNN or graph-based methods rely on **localized feature interactions** (such as $p(x)= {\textstyle \prod_{i}} p(x_i) $), the Vlasov-Poisson system enables **globally-informed** particle dynamics. This fundamental distinction provides theoretical justification for expecting superior OOD detection and generalization performance.
> - As elaborated in our response to reviewer $\text{\textcolor{red}{KKbs}}$'s question $\text{\textcolor{blue}{Q1}}$, the physical quantities emerging in the Vlasov-Poisson system (such as particles, self-consistent electric forces, and mass) all have corresponding abstract interpretations derived from the properties of high-level features.
>
> **2. Comparison to traditional statistical methods.**
>
> |||||||
> |-|-|-|-|-|-|
> | **Paradigm**| **Modeling Assumption**| **OOD Mechanism**| **Sample Interactions**| **Key Formula/Concept**| **Interpretability**|
> | **Energy-based Models (EBMs)** | Modeling with energy functions (logits or feature space energy)| Low energy → InD sample, high energy → OOD | [✖] No inter-sample/features interactions| $\mathcal{E}(x)=−\log ⁡p_\theta (x)$, where $\mathcal{E}$ is the energy of sample $x$| Intuitive in low-dimensional spaces, but hard to interpret globally|
> | **Density Estimation**| Models the probability density of data| Low probability density → OOD| [⚠] **Local** inter-sample interactions (via density estimation) | $p(x)= {\textstyle \prod_{i}} p(x_i) $, density function of feature vectorx $x$| Can capture local trends but lacks interpretability for high-dimensional spaces|
> | **Confidence Scoring**| Confidence score based on model output (e.g., softmax or ensembles)  | Low confidence → OOD| [✖] No inter-sample interactions| $C(x) = \text{softmax}(f(x))$, confidence score from model output| Heuristic and intuitive, but limited by model's ability to generalize|
> | **Ours (Vlasov–Poisson System)** | Modeling feature embeddings as charged particles interacting via a self-consistent field | OOD samples cannot reach steady state | [✔] **Globally** explicit inter-features interactions| $\frac{\partial F}{\partial t}+\mathbf{v} \cdot \nabla_\mathbf{z} F+\nabla_\mathbf{v} \cdot \mathbf{E}=0$, evolution equation for particle distribution, $\mathbf{E}$ is the electric field| Physically grounded interpretation through particle interactions (e.g., attraction/repulsion/chaos) |
> |||||||
>
> A key distinction is that most conventional methods produce point-wise scores that fail to account for how samples relate structurally to one another, or only consider local neighborhood relationships. In contrast, the Vlasov-Poisson system inherently captures collective interactions, making it particularly well-suited for subtle OOD problems, as we demonstrate through the **sharp gradient** properties near decision boundaries established in Corollary 2.3 of our paper.
>
> **3. Unique modeling advantages & interpretability.**
>
> - From an interpretability perspective, the physical metaphor (e.g., attraction/repulsion/chaos) provides an **intuitive lens** for understanding why certain samples are classified as OOD—not simply due to "low confidence" or "low density" scores, but because they fundamentally violate the governing field structure learned from InD data.
> - A key methodological distinction is that our method constitutes a system governed by PDEs that models the "collective dynamics" of the entire distribution. Unlike conventional static approaches, each sample is assessed within its relational context to all other samples through self-consistent field interactions, enabling us to capture global distributional shifts and inter-sample dependencies critical for robust OOD problems.
>
> ---
>
> > Q2: **Weaknesses-3:** The limitation of the proposed theoretical framework is not discussed. In particular... **Questions-2:** Are there specific cases where CBD fails to detect OOD or generalize...
>
> You raise a valid concern. Our hyperparameter analysis (final subplot in Figure 6) demonstrates that insufficient batch sizes during training can indeed degrade performance, we provide additional details in our response to reviewer $\text{\textcolor{red}{KKbs}}$'s question $\text{\textcolor{blue}{Q6}}$. Regarding the dimensionality issues you mention, we employ several mitigation strategies, which we address comprehensively in our response to Reviewer $\text{\textcolor{red}{KKbs}}$'s question $\text{\textcolor{blue}{Q2}}$.
>
> ---
>
> > Q3: **Weaknesses-4:** The method is built around visual features... non-image modalities... **Questions-2:** Can the authors evaluate CBD in additional domains... **Limitations-1:** The experimental design is limited to imaging data...
>
> Thank you for raising this point. We respectfully encourage you to refer to $\text{\textcolor{blue}{Q3}}$ in our response to reviewer $\text{\textcolor{red}{Sdub}}$.
>
> ---
>
> > Q4: **Weaknesses-5:** The writing is overly complex and lacks clear organization...
>
> Thank you for highlighting the writing issues. Reviewer $\text{\textcolor{red}{MQNz}}$ raised similar concerns in $\text{\textcolor{blue}{Q1 $\sim$ Q7}}$ and $\text{\textcolor{blue}{Q9}}$, which we've addressed in detail. We encourage you to refer to those responses. We will integrate these changes into the revised manuscript to enhance readability.
>
> ---
>
> > Q5: **Limitations-2:** ... increase computational and memory overhead.
>
> We acknowledge that our physics-inspired approach incurs additional computational overhead. To mitigate this, we employed two techniques to reduce this overhead, including directly parameterizing $F^* (\mathbf{z}, \mathbf{v})$ and $\phi^* (\mathbf{z})$ rather than the full time-dependent $F(\mathbf{z}, \mathbf{v}, t)$ and $\phi(\mathbf{z}, t)$, and using the Poisson-Boltzmann approximation $\phi^* (\mathbf{z}) = - \frac{1}{\epsilon_0} (e^{-\phi^* (\mathbf{z})} - \rho_{\text {ion }}(\mathbf{z}))$ in place of integral $\nabla^{2} \phi^* (\mathbf{z}) = -\frac{1}{\epsilon_{0}} ( \int F^* (\mathbf{z}, \mathbf{v}) \mathrm{d} \mathbf{v} - \rho_ {\mathrm{ion}} (\mathbf{z}) )$. The primary computational cost arises during training, driven by GPU memory demands for storing intermediate computation graphs and performing automatic gradient calculations. During inference, however, these intermediate computations are unnecessary, resulting in minimal impact on inference speed. Moreover, the added MLP branches contribute only a small fraction of model parameters (2.7% for ViT-b16 and 4.7% for ResNet-18, as detailed in Appendix Table 5), ensuring negligible effects on model storage and inference costs.
>
> ---
>
> > Q6: **Limitations-3:** Several derivations... may not capture the full complexity of real-world feature distributions.
>
> Thank you for raising this point. We believe there may be a misunderstanding here.
>
> - For OOD detection, our post-hoc method CDB-De does not modify pre-trained parameters, so InD feature distributions remain unaffected by physical assumptions.
> - For OOD generalization, our physical loss serves as a strong structural constraint. Rather than merely acting as regularization, this constraint actively guides the feature learning process to discover representations that respect underlying physical laws, thereby improving robustness across domains.
>
> We acknowledge these potential limitations and plan to explore relaxed assumptions in future work.
>
> ---
>
> [1] Out-of-distribution detection with deep nearest neighbors. ICML 2022.
> [2] Environment-aware dynamic graph learning for out-of-distribution generalization. NeurIPS 2023.
> [3] Learning invariant graph representations for out-of-distribution generalization. NeurIPS 2022.

---

> > ### Comment · Reviewer_jZnV · 2025-08-06
> >
> > Thank you to the authors for the detailed clarification and additional experiments. This clarification highlights the motivation behind the proposed approach and helps differentiate it from existing methods in OOD detection and generalization, such as energy-based models, density estimation, and confidence scoring techniques.
> >
> > However, I still have several concerns that remain:
> >
> > * **Curse of Dimensionality**: While the authors outline several strategies to mitigate high-dimensional challenges, the empirical effectiveness of these techniques is not clearly demonstrated. A more informative evaluation would involve a direct comparison of performance with and without these mitigation techniques to quantify their impact.
> >
> > * **Limited Experimental Scope**: The current experimental design is primarily focused on image data, which restricts the generalizability of the findings. Although Table 5 in the rebuttal to Reviewer Sdub provides results on multi-modal data, it only reports the performance of the proposed method without comparisons to relevant baselines. This makes it difficult to assess relative performance. Furthermore, it appears non-trivial to adapt the proposed approach to other data modalities, and the manuscript would benefit from a more in-depth discussion or demonstration of how the method generalizes beyond vision tasks.

---

> > > ### Author Response · Authors · 2025-08-07
> > > **Response to Concern 2 (Part II)**
> > >
> > > > **Limited Experimental Scope**:
> > >
> > > We appreciate your concern regarding experimental scope. We would like to emphasize that our experimental design is **consistent with most baselines and strictly adheres to established evaluation protocols** for visual OOD detection and generalization. Our work addresses two distinct tasks, which require different experimental considerations:
> > >
> > > **OOD Generalization:** The comprehensive surveys by [1] and [2] establish text and multimodal OOD generalization as distinct research domains, precisely because these challenges cannot be addressed using vision-based methodologies. For instance, text encounters fundamentally unique challenges that do not arise in vision tasks:
> > >
> > > 1. *Compositional generalization failures*: Models cannot generalize to novel combinations of seen elements. For example, a model trained on "small red objects" and "large blue objects" fails when tested on "small blue objects." This failure mode does not occur in vision tasks because visual object properties remain consistent across contexts. **Visual OOD generalization methods address distributional shifts in visual features, not symbolic compositional failures.**
> > > 2. *Discrete semantic space gaps*: Text's discrete nature requires semantically-constrained perturbations, unlike visual data where interpolation is meaningful. This creates unique methodological requirements that render existing perturbation-based visual OOD generalization baselines inapplicable to text problems.
> > > 3. *Cross-domain semantic relationship breakdown*: Sentiment models trained on movie reviews catastrophically fail on medical reviews due to completely different expression conventions. This language-structure-based failure mode differs fundamentally from visual scene understanding.
> > >
> > > In summary, **text OOD generalization involves linguistic structures, semantic relationships, and causal factors absent in vision tasks**. Addressing these language-specific failures requires specialized methodologies that cannot be developed or validated through visual experiments. Our work focuses on the visual OOD generalization problem, and incorporating text or other modality experiments does not enhance understanding of OOD generalization challenges in either domain. Furthermore, **the baselines presented in the paper are also unsuitable for text or other modality tasks, making it unfair to expect our vision-based approach to apply to these areas.**
> > >
> > > ---
> > >
> > > **OOD Detection:** Unlike generalization tasks, our post-hoc OOD detection method operates on high-level features and is **modality-independent**. After careful investigation, we conducted challenging comparative experiments on text tasks, where the results of baselines are from [3]. As shown in Table 8, our post-hoc OOD detection method achieves competitive performance.
> > >
> > > **Table 8: AUROC Comparison on Text Datasets.**
> > > ||||||||||||
> > > |:-|:-:|:-:|:-:|:-:|:-:|:-:|:-:|:-:|:-:|:-:|
> > > | **ID** | **OOD** | **MSP** | **Energy** | **GradNorm** | **KLM** | **ReAct** | **DICE** | **KNN** | **ViM** | **Ours** |
> > > | SST-2 | IMDB |83.2${\small \pm 1.4}$ | 82.7${\small \pm 2.2}$| 70.3${\small \pm 2.3}$ | 55.0${\small \pm 2.7}$ | 83.3${\small \pm 2.4}$ | 34.5${\small \pm 10.7}$ | 87.2${\small \pm 1.7}$ | 83.9${\small \pm 3.3}$ | 89.0${\small \pm 2.2}$|
> > > | SST-2 | Yelp |75.7${\small \pm 2.2}$ |75.0${\small \pm 3.1}$| 61.3${\small \pm 2.7}$| 51.3${\small \pm 3.0}$| 75.7${\small \pm 3.4}$| 35.4${\small \pm 8.4}$| 87.8${\small \pm 0.4}$| 80.1${\small \pm 2.8}$|89.3${\small \pm 1.4 }$|
> > > | Yelp | IMDB |79.5${\small \pm 0.5 }$|79.2${\small \pm1.6 }$|71.7${\small \pm1.9 }$|38.6${\small \pm1.3 }$|79.5${\small \pm1.6 }$|26.8${\small \pm5.1 }$|84.7${\small \pm0.8 }$|88.6${\small \pm0.7}$| 90.2${\small \pm0.6}$|
> > > ||||||||||||
> > >
> > > We sincerely hope our explanation addresses your concerns and provides the clarification you need. We genuinely appreciate your thoughtful feedback, which has helped us improve our work. We are committed to resolving any remaining concerns before the discussion period concludes. Thank you for the time and expertise you have devoted to reviewing our work!
> > >
> > > ---
> > > [1] Out-of-distribution generalization in natural language processing: Past, present, and future. EMNLP 2023.
> > > [2] Advances in multimodal adaptation and generalization: From traditional approaches to foundation models. arXiv 2025.
> > > [3] Classical Out-of-Distribution Detection Methods Benchmark in Text Classification Tasks. ACL 2023.

---

> > > > ### Comment · Reviewer_jZnV · 2025-08-08
> > > >
> > > > Thank the authors for the additional clarifications and experiments. While I initially expected the proposed method to handle OOD generalization for non-image modalities, I understand that extending this work in that direction is non-trivial and would require significant effort. For my other concerns, I am satisfied with the authors’ clarifications. I encourage the authors to incorporate the materials provided during the rebuttal phase into the revised version. Overall, as most of my concerns have been addressed, I have decided to increase my score to 4.

---

> > > > > ### Author Response · Authors · 2025-08-08
> > > > > **Thank you for your support!**
> > > > >
> > > > > Thank you very much for your patience in reading our comprehensive response and for your dedication in adjusting the evaluation score. We will revise the paper accordingly based on your suggestions. We truly appreciate your increased support!

---

> ### Author Response · Authors · 2025-08-07
> **Response to Concern 1 (Part I)**
>
> Dear reviewer jZnV, we greatly appreciate your continued engagement. We have carefully considered both of your concerns and provide our response below:
>
> ---
>
> > **Regarding the curse of dimensionality and empirical validation.**
>
> We have conducted relevant comparison experiments to address this concern. As detailed in our response to reviewer Sdub, Tables 6 and 7 present the comparison between MLP direct steady-state prediction (with high-dimensional mitigation) and numerical solvers using time integration for PDE solutions (without mitigation strategies), providing the empirical effectiveness demonstration you requested. The results quantify the impact of our strategy to mitigate the curse of dimensionality: while numerical PDE solvers that solve the full time-dependent system achieve marginally higher AUROC on complex datasets (improvements of 0.30%-0.42%), they suffer from computational complexity explosion and substantial GPU memory consumption, which are classic manifestations of the curse of dimensionality. In contrast, the MLP direct steady-state prediction approach maintains competitive performance while dramatically reducing computational complexity (AUROC gaps of only 0.02%-0.06% on simpler datasets). The experimental evidence demonstrates that our dimensionality mitigation strategy achieves practical effectiveness with minimal performance trade-offs.

---

### Official Review · Reviewer_KKbs · 2025-07-02

**Clarity:** 2
**Significance:** 2
**Originality:** 2
**Rating:** 3
**Confidence:** 3

**Summary:**

This paper introduces a novel physics-inspired framework named Collective Behavior Dynamics (CBD) to tackle out-of-distribution (OOD) problems. The core idea is to treat high-level features from a neural network as charged particles interacting within a self-consistent electric field, and collective behavior of these particles is modeled using the Vlasov-Poisson system. In-distribution data is assumed to correspond to initial states that converge to a steady state within this system, and the set of all such initial states forms a "basin of attraction". OOD detection is then framed as determining whether a new input falls within this basin. For OOD generalization, the goal is to expand the basin of attraction, making the model robust to a wider range of perturbations. The authors achieve this by analyzing system stability through a dispersion relation and introducing a regularization loss to suppress unstable modes.

**Questions:**

Regarding In-Distribution Structure: The mathematical implementation of the paper (Equation 4-8) seems to build only one steady-state solution and one corresponding basin of attraction. This presupposes that the dynamic behavior of in-distribution (In-D) data is unimodal. However, real-world datasets are often highly multimodal. How to bridge this gap?

Regarding the Model Choice: How crucial is the specific choice of the Vlasov-Poisson system? Could simpler dynamical systems or energy-based models capture the essential dynamics of InD vs. OOD data?

Regarding steady state: The entire theory relies on the assumption that a steady state exists. However, during Stochastic Gradient Descent (SGD) training, the underlying electric field actually changes drastically at each step due to data being fed in mini-batches. It is questionable whether the concepts of steady state can still strictly hold in a constantly changing field.

Regarding assumption: The paper assumes that high-level features behave like charged particles with uniform initial velocities. How sensitive is the model's performance to this assumption? Have the authors explored alternative physical analogies or initial conditions?

**Ethical Concerns:**

["NO or VERY MINOR ethics concerns only"]

**Final Justification:**

Thanks a lot for the rebuttal. However, given the workload for modifying this paper, the reviewer decides to maintain the original score. For the next submission, if the writing clarity could be improved, the reviewer will give the acceptance score.

**Limitations:**

See weakness and questions

**Quality:**

3

**Strengths And Weaknesses:**

Strength:
Novelty: The paper's main strength lies in its novel perspective of bridging concepts from physics (the Vlasov-Poisson system) with deep learning. This provides an interpretable model for understanding OOD phenomena, moving beyond purely statistical approaches.

Strong Empirical Performance: The proposed CBD method achieves state-of-the-art results on multiple benchmarks for both OOD detection and OOD generalization. This demonstrates the practical effectiveness of the physics-inspired approach.

Unified Approach: The framework addresses both OOD detection and generalization within a single conceptual model. The basin of attraction serves as a unifying concept for identifying outliers and for quantifying and improving model robustness.
Compatibility: The proposed method is a modular addition to existing network architectures. The experiments show that it can be combined with other advanced OOD generalization techniques (like SAM and SAGM) to yield further performance improvements.

Weakness:
1.The paper's entire theoretical foundation is based on treating feature vectors as charged particles and describing their behavior with the Vlasov-Poisson system. However, the paper does not sufficiently justify how much deviation will this approximation induce and why using this specific model.

2.The Vlasov-Poisson system is difficult to handle in high-dimensional spaces. The paper applies this model to the high-dimensional feature space generated by deep networks, but it does not explicitly address the challenges posed by the "curse of dimensionality."

3.The framework introduces several new and non-standard hyperparameters, such as the loss weights (α, β), the wavevector scaling factor (σ), and the wavenumber (N). Hyperparameter analysis is required.

---

> ### Author Rebuttal · Authors · 2025-07-29
>
> We appreciate your constructive feedback and acknowledgment of the innovative aspects of our work. Our detailed responses are as follows.
>
> ---
>
> > Q1: **Weaknesses-1:** The paper... treating feature vectors as charged particles... how much deviation will this approximation...
>
> We appreciate your concerns about modeling feature vectors as charged particles under the Vlasov-Poisson system. You questioned whether mapping computational entities to physical quantities introduces bias. We identify 3 potential sources of bias from your feedback (please note if we missed any) and address each by demonstrating our model's alignment with neural network.
>
> - **Appropriateness of Analogizing High-Level Features to Charged Particles:** This analogy represents mathematical abstraction leveraging collective behavior of interacting entities, not literal physical equivalence. Neural network features share key properties with particle systems: distributed nature (features occupy high-dimensional spaces with continuous distributions, analogous to particle phase space), interaction patterns (features influence each other through attention mechanisms and nonlinear transformations), conservation properties (information flow follows conservation principles similar to physical laws) [1,2], and statistical behavior (both systems benefit from statistical mechanics when handling large ensembles).
>
> - **Reasonableness of Feature Evolution Driven by Self-Consistent Electric Fields:**
> The attention mechanism captures relationships between tokens, analogous to self-consistent electric fields where electrons (high-level features) move under Coulomb forces from all other electrons. This analogy reveals our approach as a novel form of interaction: (i) query-key interactions represent forces, with attention weights as force magnitudes; (ii) attention implements a mean field approximation where each feature interacts with the average effect of all others; (iii) queries function as test charges probing the electrostatic field generated by keys (source charges); (iv) self-consistency emerges as attention depends on features distribution. Coulomb forces thus provide a universal **interaction kernel** in high-dimensional feature space, unconstrained by the physical properties of real charged particles.
>
> - **Lack of Mass in High-Level Features Compared to Physical Electrons:**
> As we answered reviewer $\text{\textcolor{red}{MQNz}}$ in question $\text{\textcolor{blue}{Q10}}$, in neural networks, we can interpret "mass" as the "inertia" or "stability" of features, so features with higher mass resist deviation during learning or optimization processes, just as mass affects trajectories in physics.
>
> Moreover, selective borrowing from physics is standard practice in physics-inspired machine learning. For example, [2] omits specific force laws and particle properties, while [4] disregards charge characteristics. Our charged particle modeling does constitute a form of bias. **However, structured deviation are often beneficial in deep learning: CNNs assume spatial locality, Transformers assume positional ordering, and RNNs assume sequential dependencies.** Our physical constraints enhance transparency and interpretability compared to black-box alternatives. While we acknowledge the limitations of this modeling choice, we believe the resulting performance gains and interpretability benefits justify the trade-off in our application domain. We hope this addresses your concerns.
>
> ---
>
> > Q2: **Weaknesses-2:** "Curse of dimensionality" problem.
>
> We appreciate your insightful concern regarding the curse of dimensionality. You are absolutely correct that the Vlasov-Poisson system, being a system of partial differential equations, becomes computationally prohibitive in high-dimensional settings. We would like to address how our approach mitigates these challenges:
>
> - Rather than solving the full time-dependent system $F(\mathbf{z}, \mathbf{v}, t)$ and $\phi(\mathbf{z}, t)$, which would require computationally expensive ODE integration, we directly parameterize the steady-state distributions $F^* (\mathbf{z}, \mathbf{v})$ and potential $\phi^* (\mathbf{z})$ using neural networks (Eq. 4). While this introduces some approximation error, it dramatically reduces computational complexity.
> - Instead of computing the integral $\nabla^{2} \phi^* (\mathbf{z}) = -\frac{1}{\epsilon_{0}} ( \int F^* (\mathbf{z}, \mathbf{v}) \mathrm{d} \mathbf{v} - \rho_ {\mathrm{ion}} (\mathbf{z}) )$ required by the original Vlasov-Poisson formulation, we employ the Poisson-Boltzmann approximation: $\phi^* (\mathbf{z}) = - \frac{1}{\epsilon_0} (e^{-\phi^* (\mathbf{z})} - \rho_{\text {ion }}(\mathbf{z}))$, which eliminates expensive integral computations.
> - In traditional physics, solving the Vlasov-Poisson equation typically requires complete discretization of the phase space. In our neural network approach, we conveniently employ Monte Carlo methods for random sampling of the phase space. As shown in the last subplot of Figure 6, increasing batch size allows our stochastic sampling to gradually approximate full grid-based methods.
> - We operate directly on high-level features extracted by deep networks, which have been shown to naturally capture low-dimensional manifold structure within high-dimensional spaces [5], effectively reducing the intrinsic dimensionality of the problem.
>
> Though a full theoretical analysis of high-dimensional convergence is challenging, our experiments show that these designs effectively overcome the curse of dimensionality in practice.
>
> ---
>
> > Q3: **Weaknesses-3:** Hyperparameter analysis.
>
> We have already provided hyperparameter analysis for all mentioned parameters in the Experiments section: loss weights (α, β) are analyzed in the "Hyperparameter Analysis" subsection with results in Figure 6 (subplots 4-5), while the wavevector scaling factor (σ) and wavenumber (N) are analyzed in the "Plane Wave Analysis" subsection with results in Figure 7.
>
> ---
>
> > Q4: **Questions-1:** Regarding In-Distribution Structure.
>
> We believe there is a misunderstanding. Our method does **not** assume unimodal in-distribution data. The unimodal illustrations are purely for visualization clarity, just as commonly seen in other OOD detection or DA papers.
>
> Our theoretical framework imposes **no constraints** on the shape of the InD—it accommodates arbitrary multimodal distributions. The steady-state formulation characterizes the equilibrium of the underlying dynamical system, not the geometric structure of the data distribution itself.
>
> ---
>
> > Q5: **Questions-2:** Regarding the Model Choice.
>
> The Vlasov-Poisson system lies at the heart of our framework because it uniquely captures the collective interactions of high-level features as "charged particles": the **"clustering"** of features under InD data (i.e., convergence toward steady state) versus the **chaos** or **instability** of features under OOD data (see Figure 4). This arises from the system's kinetic-potential energy balance, enabling us to quantify distributional deviations through a physics-motivated lens, which providing novel theoretical insights for OOD problems.
>
> Alternative dynamical systems typically describe **individual** particle trajectories or employ low-order approximations, sacrificing the precise modeling of collective effects that is essential for comprehensive InD/OOD characterization. Energy-based models similarly rely on static energy minimization to identify InD stable states and OOD high-energy anomalies, but neglect the **collective** interactions that are critical for capturing temporal dependencies and nonlinear effects in phase space. We kindly refer you to our response to Reviewer $\text{\textcolor{red}{jZnV}}$'s $\text{\textcolor{blue}{Q1}}$, which provides detailed motivation for our modeling approach.
>
> ---
>
> > Q6: **Questions-3:** Regarding steady state.
>
> Thanks for raising concerns about the steady-state assumption, which lies at the heart of our theoretical framework. To address this, we offer the following clarifications:
>
> - With small batch sizes, the steady state is indeed more susceptible to disruption, as noted in the hyperparameter analysis within our Experiments section (see the final subplot in Figure 6). Our results demonstrate that larger batch sizes make the steady state increasingly robust, with stabilization occurring beyond a certain threshold.
> - In our experiments, the steady state solution exhibits residual fluctuations due to limited sampling from the batch sizes used, resulting in incomplete field descriptions per iteration. Through repeated sampling, the learning process gradually refines the field’s representation, requiring multiple iterations for an accurate steady-state solution.
>
> ---
>
> > Q7: **Questions-4:** Regarding assumption.
>
> We appreciate this question about our uniform initial velocity assumption. We conducted additional experiments with different initialization strategies: uniform vectors with values of 0.5 matching the feature dimension, and random vectors of matching dimension. Random initialization slightly underperforms uniform, but differences are minimal, showing robustness. The uniform assumption maintains effectiveness while simplifying computation.
>
> **Table 1: OOD Detection on CIFAR-100**
> ||||
> | - | :-: | :-: |
> | **Initial_Velocity** | **FPR95 ↓**  | **AUROC ↑** |
> | **1** | 34.81 | 90.90 |
> | **0.5** | 35.20 | 90.47 |
> | **Random** | 38.29 | 89.52 |
> ||||
>
>
> **Table 2: OOD Generalization on PACS**
> | | |
> | :- | :-: |
> | **Initial_Velocity**| **Accuracy ↑** |
> | **1** | 87.7 |
> | **0.5**  | 87.7 |
> |**Random**|87.4|
> ||||
>
> ---
>
> [1] Neural conservation laws: A divergence-free perspective. NeurIPS 2022.
> [2] Hamiltonian neural networks. NeurIPS 2019.
> [3] Mean-field and kinetic descriptions of neural differential equations. arXiv 2020.
> [4] Poisson flow generative models. NeurIPS 2022.
> [5] Representation learning: A review and new perspectives. TPAMI 2013.

---

> > ### Comment · Reviewer_KKbs · 2025-08-07
> >
> > Thank you very much for your detailed answer!  Parts of the comments are addressed. However, there are still some questions which are still remained to be resolved.
> >
> > 1.Main: Necessity of introducing the complex physical systems' concepts.
> > The authors list a few examples to explain that introducing the physical systems' concepts is necessary and good for the development of machine learning. For example, CNNs assume spatial locality, Transformers assume positional ordering, and RNNs assume sequential dependencies. These are just very good examples. These examples do not have to introduce extra recognition burdens and are crystally clear on how they achieve good performance. In the reviewer's point of view, this paper could have been a very good one without spending too much space on the physical background, which may be unfamiliar to a large group of readers in this venue. Instead, if the paper is written in a plain manner, how the methods are derived and maybe additionally how it is related to physics, instead of directly borrowing from physics, it would be more appealing.
> >
> > 2.Others: Convergence in the high-dimensional case.
> > This further reveals the additional complexity would increase the difficulty in terms of both theoretical analysis and large-scale  experiments.

---

> > > ### Author Response · Authors · 2025-08-07
> > > **Response to Concerns**
> > >
> > > Dear Reviewer KKbs,
> > >
> > > We sincerely appreciate your thoughtful consideration of our rebuttals. We are glad that our previous rebuttal have addressed some of your concerns, and we appreciate the opportunity to clarify the remaining points regarding the necessity of the physics concepts and the convergence analysis:
> > >
> > > ---
> > >
> > > > **Regarding the necessity of the physics systems' concepts.**
> > >
> > > We understand your concern that introducing concepts from physical systems might create a "recognition burden" for some readers in the machine learning community. However, we respectfully argue that the physical systems is not merely decorative, it is fundamental to our contribution.
> > >
> > > - **Physics provides principled design, not just analogy.** Our algorithm doesn't simply dress up machine learning heuristics in physical metaphors.  Instead, it directly leverages core concepts from established physical systems, giving us a rigorous conceptual foundation. Without this physical grounding, the mathematical operations would appear arbitrary and ad hoc, leaving readers questioning our design choices. The Vlasov-Poisson system we employ has been rigorously developed and validated in physics over half a century. This deep natural foundation provides a level of sophistication that cannot be achieved through heuristics alone. Stripping away the physics would hide the unifying logic that makes our approach coherent and principled.
> > >
> > > - **Physics enhances clarity and intuition.** By building on well-established physical concepts, we avoid lengthy derivations from first principles and present complex ideas efficiently. The physics background also provides universally understood definitions for terms like "basin of attraction" and "steady state," making these concepts more intuitive than abstract mathematical formulations alone.
> > >
> > > We fully agree that clarity is crucial and recognize the importance of making our work accessible to the broader community. To address your concerns, we plan to adjust the paper in the following ways:
> > >
> > > - Refine the narrative flow by revising the paper to first present a clear, high-level physical motivation (the "why"), followed immediately by the rigorous, self-contained mathematical formulation (the "how").
> > >
> > > - Appropriately reduce the introduction of physical concepts in the main text, moving more detailed explanations to the appendix for interested readers.
> > >
> > > - Strengthen the motivation in the main text by more clearly explaining how our physics-inspired method provides unique solutions.
> > >
> > > We believe these adjustments will ensure that readers who are uncomfortable with the physics can still follow the technical derivation, while others can benefit from the deeper intuition.
> > >
> > >
> > > ---
> > >
> > > > **Regarding convergence in high-dimensional cases.**
> > >
> > > We fully acknowledge your comment regarding the challenge of convergence in high-dimensional settings. Indeed, providing a complete and formal proof of convergence in arbitrary high-dimensional cases remains a significant challenge for the theoretical machine learning community. This is a well-recognized difficulty in the field.
> > >
> > > In the context of the Vlasov-Poisson system under the background charge assumption, it has been rigorously proven that the system converges to a steady state in the low-dimensional case, as established in the physics literature [1][2]. While the extension of this result to high-dimensional settings is more complex and still remains an open question. However, as demonstrated in our rebuttal to reviewer Sdub (Table 6 and Table 7), we performed numerical integration of the full PDE and observed convergence to the steady state in our experiments, which we will further highlight in the revised manuscript by including the loss curve plots to better illustrate this convergence.
> > >
> > > Our model's rich dynamics enable the modeling of sophisticated feature interactions that are may be beyond the representational capacity of simpler models. We believe that complexity when it serves to better model the underlying feature interactions is worthwhile.
> > >
> > > ---
> > >
> > > As the end of the discussion period approaches, we would greatly appreciate your input on the paper. Any further questions or suggestions are welcome, and we hope for the opportunity to respond to them. Thank you for your time and effort!
> > >
> > > ---
> > >
> > > [1] Stationary solutions of the Vlasov-Fokker-Planck equation. Mathematical methods in the applied sciences 1987.
> > > [2] On long time asymptotics of the Vlasov-Fokker-Planck equation and of the Vlasov-Poisson-Fokker-Planck system with Coulombic and Newtonian potentials. Differential and Integral Equations 1995.

---

### Official Review · Reviewer_MQNz · 2025-07-02

**Clarity:** 2
**Significance:** 3
**Originality:** 4
**Rating:** 4
**Confidence:** 3

**Summary:**

The paper introduces a novel physics perspective on out-of-distribution detection by treating high-level data features as particles in a field. These particles evolve under the influence of a learned potential field, with in-distribution samples converging toward a steady-state region or “basin of attraction.” The boundary of this basin is then used as a decision surface for detecting OOD inputs. The authors support their proposal with theoretical results involving partial differential equations and evaluate their approach on standard OOD benchmarks.

**Questions:**

Please help me address the weaknesses above. In particular, can you explain the full training and testing procedure for your models?
Additionally, what hyperparameter tuning was performed for your method? (e.g. how did you choose the MLP architectures etc.)

Note that I am willing to raise my score upon reaching more clarity on the above.

**Ethical Concerns:**

["NO or VERY MINOR ethics concerns only"]

**Final Justification:**

I found this paper to be interesting, although limited in empirical rigor and clarity. I am somewhat concerned that the paper is excessively "motivated", in the sense that the authors seem like they want the physics interpretation to be constructive even if the empirical evidence of its value is less clear. However, the authors engaged with me and addressed most of my clarity concerns during the rebuttal, and with the updates I think the paper is sufficiently interesting to justify my recommendation.

**Limitations:**

Yes

**Paper Formatting Concerns:**

No concerns

**Quality:**

3

**Strengths And Weaknesses:**

*Strengths*
- The idea of interpreting feature dynamics via a physical system governed by a PDE and identifying the basin of attraction as an OOD boundary is original and interesting
- The authors attempt to rigorously ground their model in physics-inspired PDE theory, including steady-state solutions and stability analysis
- The approach appears to be usable in a post-hoc setting, potentially making it broadly applicable to existing trained models (though this is not clearly communicated in the main text).
- The authors speculate that model robustness may correlate with the size of the basin of attraction, which, if substantiated, could offer a new direction for designing new methods.
- The approach achieves supposedly SotA performance for OOD detection and generalisation (however, I was not able to reach sufficient clarity on how this method produces scores for an individual input point, and how exactly the networks are trained to confidently interpret these results, see weakness below)

*Weaknesses*
- Key terms like "collective behavior," "self-consistent electric field," "macroscopic properties," and "perturbation" are not clearly defined early in the paper, making it very hard to appreciate clearly without several passes
- Many technical statements are vague or confusing (e.g., “Plane wave modes” as an x-axis label in a figure; unclear phrasing such as “measure its boundary” without formally clarifying what "its" refers to).
- The flow of theoretical principles (e.g., Principles 1 and 2) lacks sufficient intuitive grounding and logical coherence (i.e. "thus" is used despite it being unclear how the two principles are necessary or sufficient for the subsequent statement).
- Several figures (e.g., Figure 1 and Figure 2) are confusing or potentially mislabeled, weakening the explanatory power of the paper.
- There is an over-reliance on jargon without providing accessible interpretations early in the paper, making the work difficult to follow even for an expert audience familiar with OOD literature.
- It’s not transparent from the main body how the method is actually used to compute OOD scores. Only by reading the appendix does it become evident that the method is post-hoc ("post-hoc" is not even mentioned anywhere in the paper, nor the fact that the F and phi functions are learned with gradient descent using held out InD data; my experience of reading this paper was that I like the idea but I felt continuously unclear what the method itself actually is until reading the appendix--I should not need to read the appendix to appreciate the actual algorithm)
- The use of "batch" in the pseudocode is ambiguous -- it's not clear whether the method assumes access to a test-time batch or operates on individual points, it's not clear whether the batch is used just to fit the F and phi, but then at test time use only a singleton input point, or if the intended use is to use batches at test time.
- “we posit that network robustness improves as the basin’s range expands.” → “How can we enlarge basin of attraction’s boundary to enhance model generalization?” This is an interesting idea but I don’t like that they went from an idea to a question that presumes the idea is good and operationalises it. It is not clear that what they posit is indeed true, so it feels like a big leap to then start already seeking to maximize it.
- Notation in Theorem 2 (e.g., $L_1$, $L_\infty$, $n_i(z)$, $\delta(v−v_0)$) is introduced without definition.
- The physical analogy (particles, mass, steady state) is inconsistently applied and appears more illustrative than foundational e.g. ("mass" appears once in the paper, but no definition or intuition).
- Modulo my lack of clarity on the exact evaluation setting (e.g. what is the "batch" in the pseudo-code, how many particles lie in R^z for an individual input point (is this $N$?)) the results seem compelling, but confidence intervals are provided in the tables without any explanation for what they represent and how they are constructed.

*Small points*
- Figure 1 shows only a single particle, despite the paper’s emphasis on collective particle dynamics. Consider revising to reflect the collective behavior.
- The x-axis label of Figure 2 “Plane wave modes” is confusing and lacks explanation. Also, fix the typo “solusion.”
- In the sentence “as illustrated in Figure 6,” it seems you meant Figure 2 instead.

---

> ### Author Rebuttal · Authors · 2025-07-28
>
> Thank you for the positive feedback and insightful questions. We really appreciate your kind words about our work, and your suggestions will help us improve both the quality and clarity of our research. Below, we will address your concerns one by one.
>
> ---
>
> > Q1: Key terms like... are not clearly defined early in the paper, making it very hard to appreciate clearly without several passes.
>
> We acknowledge that key terms lack clear definitions early in the paper. In the revision, we will include the following definitions in the Introduction:
>
> - **Collective behavior:** Emergent system-wide phenomena resulting from multi-particle interactions. In our work, where high-level features are treated as charged particles, this represents the temporal evolution of feature distributions driven by inter-feature interactions.
> - **Self-consistent electric field:** Field where particles generate the electric field that governs their motion, creating iterative dynamics. Particles produce a field, which drives their movement, redistributing the particles and generating a new field until equilibrium is reached, leading to collective behavior.
> - **Macroscopic properties:** System-level characteristics, which in our work refer to the particle distribution function $F(\mathbf{z},\mathbf{v},t)$.
> - **Perturbation:** Minor deviations from the steady-state particle distribution function and electric potential (line 161).
>
> ---
>
> > Q2: Many technical statements are vague or confusing (e.g., "Plane wave modes"... what "its" refers to).
>
> We acknowledge that several technical statements require clarification:
>
> - **Regarding the x-axis label in the Figure 2:** It should be labeled 'time'. Plane wave modes decompose perturbations into constituent waves with distinct directions and wavelengths, similar to how Fourier transforms decompose images into frequency components, revealing each wave's contribution to the overall perturbation.
> - **Regarding unclear pronoun references:** In the expression 'measure its boundary', the pronoun 'its' specifically refers to the 'basin of attraction'.
>
> We will conduct a thorough review of the manuscript and revise any instances of ambiguous words in the revised submission.
>
> ---
>
> > Q3: The flow of theoretical principles... lacks sufficient intuitive grounding and logical coherence.
>
> We sincerely apologize for the unclear logical progression of our theoretical principles and thank you for this valuable feedback. We respectfully encourage you to refer to our response to Reviewer $\text{\textcolor{red}{jZnV}}$'s question $\text{\textcolor{blue}{Q1}}$ for the motivation behind our modeling approach. Building upon that foundation, the relationships among our theoretical principles follow this logical sequence:
>
> - Theorem 2.1 establishes that steady-state solutions of Vlasov-Poisson systems provide a theoretical foundation for OOD detection.
> - Since directly solving the steady-state electric field $E^* (\mathbf{z})$ (Eq 4) involves computationally challenging integrals, Definition 2.2 introduces a tractable condition via the Poisson-Boltzmann equation for the steady-state electric potential $\phi^* (\mathbf{z})$, simplifying the computation of $E^*(\mathbf{z})$.
> - Corollary 2.3 connects the basin of attraction $\mathcal{B}$, its boundary $\partial \mathcal{B}$, and the steady-state potential $\phi^*(\mathbf{z})$, revealing sharp gradients near $\partial \mathcal{B}$ that further justify the system's applicability to OOD detection.
> - Theorem 2.4 proves that our proposed dispersion loss expands the basin of attraction $\mathcal{B}$.
> - Theorem 2.5 and Corollary 2.6 demonstrate that this expansion enhances neural network robustness.
>
> We will clarify these logical connections with appropriate transitional explanations in the revision.
>
> ---
>
> > Q4: Figures errors (with all small points).
>
> Thank you for the detailed observations! We will revise all relevant figures in the revision
> to correct labeling errors, improve axis descriptions, fix spelling errors, and ensure the figures accurately reflect the collective dynamics discussed in the text.
>
> ---
>
> > Q5: There is an over-reliance on jargon.
>
> We apologize for not adequately considering the accessibility of our mathematical and physics concepts for readers with diverse backgrounds. To improve clarity, we will add accessible explanations in the introduction footnotes, and provide intuitive descriptions of technical terms in the appendix. We hope these revisions address your concerns.
>
> ---
>
> > Q6: It's not transparent from the main body how the method is actually used... the method is post-hoc... I like the idea but I felt unclear what the method itself actually is until reading the appendix...
>
> We thank you for appreciating the core idea. We acknowledge that the main text inadequately explained the post-hoc nature of our method and the OOD implementation process. In the revised manuscript, we will:
>
> - Explicitly present the OOD detection loss $\mathcal{L}_ {\text{De}} = \mathcal{L}_ {\text{Vlasov}} + \mathcal{L}_ {\text{Poisson}}$ and OOD generalization loss $\mathcal{L}_ {\text{Gen}} = \mathcal{L}_ {\text{Classify}} + \alpha(\mathcal{L}_ {\text{Vlasov}} + \mathcal{L}_ {\text{Poisson}}) + \beta \mathcal{L}_ {\text{disp}}$ in the methodology section.
> - Clarify in the Introduction that *CBD*-De is a post-hoc detection method that does not require modifying pre-trained network parameters.
> - Label the input as InD data in the left subplot of Figure 3 to clarify that $F$ and $\phi$ are learned on the InD data.
>
> ---
>
> > Q7: The use of "batch" in the pseudocode is ambiguous...
>
> The batches in our pseudocode are used exclusively during training to fit $F^* (\mathbf{z}, \mathbf{v})$ and $\phi^* (\mathbf{z})$, while at test time we process **singleton** inputs (or **arbitrary** batch sizes). We will clarify this distinction in the revised manuscript.
>
> ---
>
> > Q8: ... from 'we posit...' to 'how can we enlarge...' may unintentionally suggest that the hypothesis is taken as established fact.
>
> We acknowledge this logical leap and thank you for this important observation. The transition from positing our hypothesis ('robustness improves as the basin's range expands') to immediately asking how to enlarge the basin was premature, as it assumed the hypothesis was valid. Our intent was to explore a promising direction, not assert a fact. While **Theorem 2.5 and Corollary 2.6 provide theoretical support for this hypothesis**, we will revise the manuscript to explicitly frame basin enlargement as a hypothesis-driven investigation.
>
> ---
>
> > Q9: ... $L_1$, $L_\infty$, $n_i(z)$, $\delta(v−v_0)$ is introduced without definition.
>
> Thank you for highlighting the missing definitions in Theorem 2.1. This was an oversight on our part. In the revision, we've added the following explanations:
>
> - $L^1(\mathbb{R})$ refers to the space of integrable functions on $\mathbb{R}$, while $L^\infty(\mathbb{R})$ denotes the space of essentially bounded functions.
> - $n_i(z)$ represents the spatial density for the $i$-th system.
> - $\delta(v - v_0)$ is the Dirac delta distribution centered at $v = v_0$, indicating a velocity distribution focused at that point.
>
> ---
>
> > Q10: ... (particles, mass, steady state) is inconsistently applied and appears more illustrative than foundational...
>
> That's a great question. It gives us a chance to elaborate on our thinking about neural networks, especially the concept of "mass".
>
> - In physics, particles have position. It's reasonable to use "particles" for advanced features because they behave like **discrete entities** in a high-dimensional feature space, akin to charged particles in plasma driving dynamics similar to network training.
> - "Mass" relates to inertia—the property that allows an object to maintain its state of motion (rest or constant velocity), with greater mass requiring more force to induce change. In neural networks, **we can interpret "mass" as a feature's "inertia" or "stability"**, so features with higher mass resist shifts during learning or optimization, much like mass influences trajectories in physics. This could link to the feature's impact on network output.
> - As for "steady state," it corresponds to the condition where the partial derivative of the distribution function $ F(\mathbf{z}, \mathbf{v}, t) $ with respect to time equals zero, indicating equilibrium.
>
> We’ll include dedicated sentences in the revision to expand on these ideas.
>
> ---
>
> > Q11: Modulo my lack of clarity on the exact evaluation setting...
>
> Thanks for the feedback. The batch definition is addressed in our response to $\text{\textcolor{blue}{Q7}}$. Confidence intervals in the tables reflect mean ± standard error over three independent test runs. We'll clarify this in the table captions.
>
> ---
>
> > Q12: Full training and testing procedure and hyperparameter tuning.
>
> - **For OOD detection:** We freeze a pre-trained model and add two MLP branches after the feature extractor (Figure 3), training only these branches on InD data with loss $\mathcal{L}_{\text{De}}$ and batch size 128. At test time, scores are computed from MLP outputs (Eq.9) with flexible batch sizing.
> - **For OOD generalization:** We train all model parameters and  MLP branches with loss $\mathcal{L}_{\text{Gen}}$ using batch size 32 (24 for DomainNet). Figure 6 provides batch size analysis. Testing uses classifier outputs only, with arbitrary batch sizes supported.
> - **Hyperparameter tuning:** We employed Optuna with systematic optimization based on GPU memory constraints and domain knowledge from related OOD literature. Using TPE sampling across 30 trials, each configuration was evaluated with 3 random seeds for stability. For example, the search space on CBD-De included:
>
> ```python
> {
>     'batch_size': [32, 64, 128],
>     'learning_rate': [1e-3, 5e-4, 1e-4, 5e-5],
>     'hidden_dim': [128, 256, 512, 1024],
>     'MLP_layers': [1, 2],
>     ...
> }
> ```
>
> All reported results use the optimal configurations identified through this process.

---

> > ### Comment · Reviewer_MQNz · 2025-08-08
> >
> > Thank you for your thoughtful, in-depth responses. This significantly improves the clarity of the paper. I have updated my original review.

---

> > > ### Author Response · Authors · 2025-08-08
> > > **Thank you so much!**
> > >
> > > Thank you for your supporting our work! Your valuable suggestions and insights have significantly helped us to improve our manuscript. Once again, we sincerely appreciate your time and efforts!

---

> ### Author Response · Authors · 2025-08-07
> **Thank you for your review!**
>
> Dear reviewer MQNz,
>
> We greatly appreciate your time and effort in reviewing our work, and understand that you may have a busy schedule. We are eager to ensure that we have adequately addressed your concerns and are prepared to offer further clarifications or address any additional questions you may have.
>
> Since the discussion period will end in around 40 hours, we will be available and happy to respond to any feedback or questions you may have regarding our rebuttal. We would be grateful if you could share your thoughts with us.
>
> Best regards,
>
> Authors

---

### Comment · Area_Chair_zfhS · 2025-08-07

Dear Reviewers,

Thank you for your efforts in reviewing the submission.

The authors have submitted their rebuttals in response to your comments. Please ensure that you engage with the rebuttal and provide a response before selecting “Mandatory Acknowledgement.”

Please note “Mandatory Acknowledgement” button is to be submitted only when reviewers fulfill all conditions below (conditions in the acknowledgment form):
1. read the author rebuttal
2. engage in discussions (reviewers must talk to authors, and optionally to other reviewers and AC - ask questions, listen to answers, and respond to authors)
3. fill in "Final Justification" text box and update “Rating” accordingly (this can be done upon convergence - reviewer must communicate with authors first)

---

### Decision · Program_Chairs · 2025-09-17

**Decision:**

Accept (poster)

**Comment:**

The paper models features as charged particles under a self-consistent field to unify OOD detection and generalization. It achieves strong benchmark results and complements existing methods, and the authors’ thorough rebuttal further demonstrates its value. Therefore, I recommend acceptance.